# Triggers of the 2022 Larsen B multi-year landfast sea ice break-out and initial glacier response

**Naomi E. Ochwat[1,2]\*, Ted A. Scambos[1], Alison F. Banwell[1], Robert S. Anderson[2], Michelle L. Maclennan[3], Ghislain Picard[4], Julia A. Shates[5], Sebastian Marinsek[6], Liliana Margonari[6], Martin Truffer[7,8], and Erin C. Pettit[9]**

[1]Earth Science Observation Center (ESOC), Cooperative Institute for Research in Environmental Sciences (CIRES), University of Colorado Boulder, Boulder, CO, USA

[2]Department of Geology, University of Colorado Boulder, Boulder, CO, USA,

[3]Department of Atmospheric and Oceanic Sciences, University of Colorado Boulder, Boulder, CO, USA,

[4]Univ. Grenoble Alpes, CNRS, Institut des Géosciences de l'Environnement (IGE), UMR 5001, Grenoble, France

[5]Department of Atmospheric and Oceanic Sciences, University of Wisconsin–Madison, Madison, WI, USA

[6]Instituto Antártico Argentino, Buenos Aires, Argentina

[7]Geophysical Institute, University of Alaska Fairbanks, Fairbanks, AK, USA,

[8]Department of Physics, University of Alaska Fairbanks, Fairbanks, AK, USA

[9]College of Earth, Ocean, and Atmospheric Sciences, Oregon State University, Corvallis, OR, USA

*correspondence to: N. Ochwat (naomi.ochwat@colorado.edu)*

**Abstract**

In late March 2011, landfast sea ice (hereafter, 'fast ice') formed in the northern Larsen B embayment and persisted continuously as multi-year fast ice until January 2022. In the 11 years of fast ice presence, the northern Larsen B glaciers slowed significantly, thickened in their lower reaches, and developed extensive mélange areas leading to the formation of ice tongues that extended up to 16 km from the 2011 ice fronts. In situ measurements of ice speed on adjacent ice shelf areas spanning 2011 to 2017 show that the fast ice provided significant resistive stress to ice flow. Fast ice breakout began in late January 2022, and was closely followed by retreat and break-up of both the fast ice mélange and the glacier ice tongues. We investigate the probable triggers for the loss of fast ice and document the initial upstream glacier responses. The fast ice break-up is linked to the arrival of a strong ocean swell event (>1.5 m amplitude; wave period waves >5 s) originating from the northeast. Wave propagation to the ice front was facilitated by a 12-year low in sea ice concentration in the northwestern

Weddell Sea, creating a near-ice-free corridor to the open ocean. Remote sensing data in the months following the fast ice break-out reveals an initial ice flow speed increase (> 2-fold), elevation loss (9 to 11 m), and rapid calving of floating and grounded ice for the three main embayment glaciers Crane (11 km), Hektoria (25 km), and Green (18 km).

## 1 Introduction

As the climate warms, ice shelves in Antarctica are predicted to become more susceptible to collapse (Mercer, 1978; Gilbert and Kittel, 2021). In the late 1980s and mid 1990s several ice shelves along the Antarctic Peninsula (AP) coast retreated and eventually disintegrated, including the Wordie, Prince Gustav, Larsen Inlet, Larsen A ice shelves, and in March 2002, the northern two-thirds of the Larsen B Ice Shelf (Rott et al., 1996; Glasser and Scambos, 2008; Cook and Vaughan, 2010). In 2008 and 2009, several smaller break-up events occurred on the Wilkins Ice Shelf (Braun et al., 2009; Scambos et al., 2009). There has been significant research elucidating the causes of these collapses, focusing on both ice-shelf thinning due to basal and surface melting (Smith et al., 2020), as well as lake drainage mechanisms related to surface meltwater-induced ice-shelf flexure and hydrofracture (Doake and Vaughan 1991; Scambos et al., 2000; Scambos et al., 2003; Banwell et al., 2013; Banwell and MacAyeal, 2015) partly attributed to warmer climate conditions (Rott et al., 1998), plate-bending stresses on the ice-shelf front (Scambos et al., 2009), and ocean swell flexure (Massom et al., 2018). Massom et al. (2018) further implicate loss of fast ice and ocean swell in the Wilkins Ice Shelf breakup events, following loss of a protective pack ice buffer offshore – due to the vulnerability of fast ice to ocean swells (Crocker and Wadhams, 1989; Langhorne et al., 2001). While fast ice is consolidated sea ice that remains stationary attached to the coast and can be annual or perennial (Fraser et al., 2021), pack ice refers to sea ice that is comprised of separate floes and is under the influence of winds and ocean currents. The loose structure of pack ice has a strong damping effect on ocean swell (Squire, 2007).

Intense surface melt events on the eastern Antarctic Peninsula have been linked to atmospheric rivers (ARs; Wille et al., 2019; Wille et al., 2022) and foehn winds (Cape et al., 2015; Datta et al., 2019; Laffin et al., 2022). ARs are long narrow bands of warm and moist air that can cause extreme warm temperatures, increase surface melting, advect sea ice away from the ice edge, reduce sea ice concentrations, and generate foehn events (Bozkurt et al., 2018; Wille et al., 2022; Liang et al., 2023). Foehn events occur when a moist air mass ascends on the windward side of a mountain range or ridge and cools at the (lower) wet-adiabatic rate, while losing moisture to precipitation. It then descends over the lee side, adiabatically warming at the higher dry-air rate, resulting in an increase in temperature. The loss of ice shelves can substantially reduce the stability of tributary outlet glaciers, leading to acceleration, increased calving, thinning, and ultimately, sea level rise. For example, when the Larsen B collapsed in 2002, Crane Glacier thinned by 25 m yr$^{-1}$ over much of its length (Needell and Holschuh, 2023) and immediately sped up by roughly 3-fold (Rignot et al., 2004) and the Hektoria-Green-Evans (hereafter, HGE) Glacier system ice flow speed increased by up to 8-fold (Rignot et al., 2004). After the collapse of the Larsen B in 2002, the embayment was frequently filled by seasonal sea ice (landfast and pack ice). However, in late March 2011, landfast sea ice (hereafter 'fast ice') formed in the

Larsen B embayment, and the interior two-thirds of the embayment was continuously covered by multi-year fast ice until January 2022. On 19 January 2022, this fast ice cover was suddenly fractured and began to drift out, leading within days to retreat and break-up of the tributary glacier mélange and floating ice tongue areas (Fig. 1). Fast ice has been shown to stabilize outlet glaciers by reducing calving (Amundson et al, 2010; Robel 2017) and suppressing wave action against the outlet glacier (Murty, 1985; Langhorne et al., 2001) causing the glacier terminus to advance (Reeh et al., 2001). When fast ice or mélange breaks up, ice-shelf calving resumes, sometimes releasing several decades of accumulated ice flux, and exposing the new terminus to ocean dynamics (Reeh et al., 2001; Cassotto et al., 2015).

Several studies have suggested that break-up of fast ice can reduce the structural integrity of ice shelves and ultimately lead to their collapse (Khazendar et al., 2007; Massom et al., 2010; Borstad et al., 2013; Banwell et al., 2017; Massom et al., 2018). There have been many examples of tributary glacier acceleration and significant ice front retreats following the removal of fast ice or pack ice in Greenland and Antarctica (Miles et al., 2017, Miles et al., 2018, Gomez-Fell et al., 2022). Others (Sun et al., 2023; Surawy-Stepney et al., 2023) suggest that fast ice does not provide sufficient buttressing (resistive stress to impact the system dynamics.

Here we investigate the climatic and oceanic drivers that led to the rapid break-out of the decade-old Larsen B fast ice in January 2022, while also drawing parallels to previous fast ice and ice shelf collapses. We then assess the initial glacier dynamic response to the loss of the buttressing fast ice by evaluating changes in velocity, terminus position, and elevation of the Crane, Jorum, Punchbowl, and HGE glaciers. A preliminary assessment of the cause of the fast ice break-up was discussed as a sidebar in the NOAA State of the Climate 2022 report (Ochwat et al., 2023b). However, the current study evaluates the events in much greater detail and includes a quantitative look at the glacier response.

**2 Study area**

The Larsen B embayment (65.24° S, and 61.00° W; Fig. 1) is located on the eastern side of the AP, between Graham Land and the northwestern Weddell Sea and is ~7000 km$^2$ in area. North of the embayment is the Seal Nunataks Ice Shelf (Shuman et al., 2016) and to the south is a remnant of the Larsen B Ice Shelf; the Scar Inlet Ice Shelf. Prior to 1995, the eastern coast of Graham Land was almost entirely flanked by ice shelves (e.g., Skvarca et al., 1999; Cook and Vaughan, 2010), but after a series of disintegrations, only the Scar Inlet and Larsen C Ice Shelf remain.

Due to the elevated narrow ridge of the northern AP (Graham Land) and the prevailing westerly wind, the climate of the Larsen B embayment region differs greatly between the western and eastern flanks. The ridge obstructs the Southern Hemisphere westerlies and induces strong orographic lifting and precipitation on the western side, while the eastern side is much drier and cooler (King et al., 2003; Van Wessem et al., 2015). The climate is heavily influenced by the phase of the Southern Annual Mode (SAM; Leeson et al., 2017; Fogt and Marshall, 2020). When the SAM index is positive, warming events occur more

 frequently on the eastern side of the Peninsula due to an increase in westerly flow across the Peninsula (Orr et al., 2008; Van

Lipzig et al., 2008).

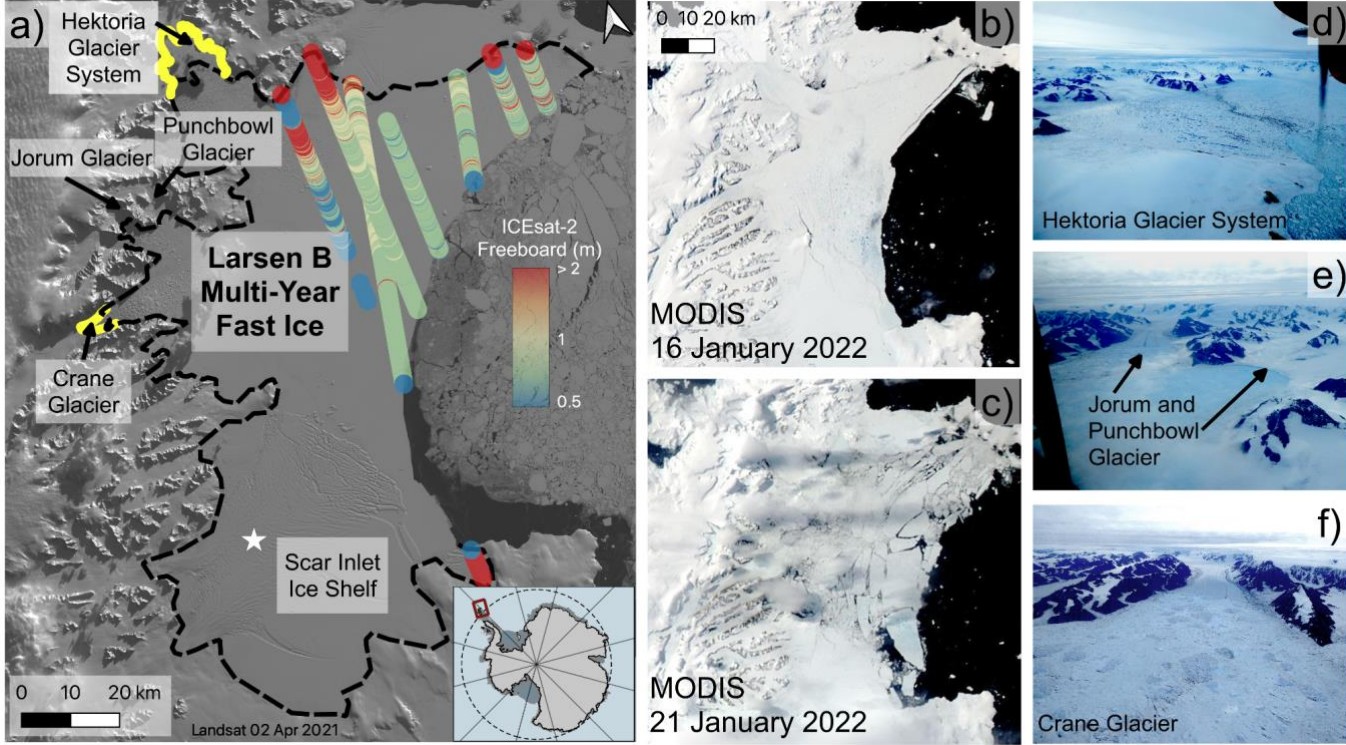


*Figure 1: a) Freeboard thickness from ICESat-2 data from 1 January 2021 to 1 January 2022. The yellow dashes show the*
*TanDEM-X determined 2016 grounding zone (Rott et al., 2018) and black dashes are a slope change and calving-morphology*
*inferred 2021 grounding zone (this study). An AMIGOS GPS installation on Scar Inlet Ice Shelf is indicated by the white star.*
*The background image is from Landsat 8 2 April 2021. b) MODIS image from 16 January 2022. c) MODIS image from 21*
*January 2022, two days after initial rifts in the fast ice formed. d, e, f) Images captured by a British Antarctic Survey overflight*
*on 31 January, 11 days after the fast ice break-out event.*
**3 Data and Methods**
The following datasets are used in various capacities to evaluate the triggers of the fast ice break-out as well as the initial
glacier response. Reanalysis data is used to evaluate both potential atmospheric and oceanic triggers. Passive microwave data
is used to determine sea ice extent and surface melt conditions. Optical satellite imagery from a number of satellite systems,
and synthetic aperture radar data, are used for assessing glacier ice, fast ice, elevation changes, and determining glacier speeds.
Laser altimetry data is also used for assessing initial glacier and fast ice elevation. Lastly, GNSS data is used to look at Scar
Inlet Ice Shelf speeds.

## 3.1 Reanalysis Data

We used ERA-5 Reanalysis data (Hersbach et al., 2020) at both monthly and hourly temporal resolution to assess temperature and precipitation anomalies in 2017 to 2022, as well as foehn wind occurrence in the months prior to and including January 2022. To investigate the presence of foehn winds, we followed Laffin et al. (2022), who determined that foehn winds that produce surface melt require a temperature $> 0°C$, a wind speed of $> 2.85$ m s$^{-1}$, humidity $< 79\%$, and a wind direction from the north or northwest. These thresholds agree with Cape et al. (2015)'s determination of the onset of "foehn days" (foehn conditions for greater than 6 hours).

To identify ARs during the last two weeks of January 2022, we use hourly ERA-5 to examine vertically integrated water vapor transport (IVT) bands that extend from the extra-tropics towards the Antarctic ice sheet (Bozkurt et al., 2018; Wille et al., 2019). IVT is calculated as the vector magnitude of eastward integrated water vapor transport (uIVT) and northward integrated water vapor transport (vIVT). We identify an AR event during the breakout as a continuous, extended region of locally high IVT that reaches a peak intensity of almost 300 kg m$^{-1}$ s$^{-1}$, consistent with Wille et al. (2022).

Following Massom et al. (2018) and Teder et al. (2022), we investigate the occurrence of open-ocean corridors across the sea ice zone, using ERA-5 and WaveWatch III wave data. We used significant wave height as a proxy for wave energy (Teder et al., 2022), calculated to be four times the square root of the zeroth moment of the energy density spectrum (Massom et al., 2018). We used peak wave period as an indication of longer swell wavelengths, which can transmit more energy into the fast ice plate (Robinson and Haskell, 1992; Massom et al., 2018). Mean wave direction is used to assess alignment with the corridor axis and propagation toward the ice front. We examined the hourly time series of ocean wave variables for January 2022 at two different locations, within the corridor and near the Larsen B fast ice front.

## 3.2 Satellite Data

### 3.2.1 Passive Microwave Data

We combined passive microwave data from two successive sensors, namely the Advanced Microwave Scanning Radiometer for the Earth Observing System (AMSR-E) on the Aqua Satellite, and the Advanced Microwave Scanning Radiometer 2 (AMSR-2) on the 'Shizuku' (GCOM-W1) satellite. Together, these passive microwave sensors provide nearly continuous daily data from 2002 until present (apart from a gap from October 2011 to June 2012 between the two satellites' operation). The daily overall sea ice concentration data product (Spreen et al., 2008) was used to assess overall sea ice extent and concentration on the same day (January 19th) for the 12-year period.

We also used AMSR-E/2 data to investigate fast ice melt extent for each melt season (October 1 to March 31) from 2011 to 2022. To do this, we followed the algorithm from Torinesi et al. (2003) and methods of Picard et al. (2007). For each 12.5 km

grid cell and each day, the liquid water is detected as present if the 19 GHz horizontally-polarized brightness temperature is higher than a threshold that is empirically determined in each cell and for each year by using the brightness temperatures during the winter (dry snow) season. 'Melt days' are defined as days when meltwater is present on or near the ice surface, but active melting is not necessarily taking place. Finally, we calculated the total number of melt days for each melt season. We used the same fully automatic algorithm as used recently in Banwell et al. (2021, 2023) for Antarctic ice shelves. However, for the current study, a careful visual evaluation of the passive microwave data (brightness temperature timeseries) was done for all the pixels in proximity to the open ocean. This was necessary because the real footprint of the measurements acquired by the radiometer is larger than the pixel size and of elliptical shape (14 x 22 km) (Meier et al., 2018). The ellipse's position and orientation changes from track to track with respect to the pixels. As a consequence, the fast ice pixel near the shore may be contaminated by signal coming from the nearby open ocean, hence potentially perturbing the detection of the melt. Our manual selection prevents this effect.

### 3.2.2 Optical Imagery and Synthetic Aperture Radar

We used MODIS (Moderate-Resolution Spectroradiometer), Landsat 8 and 9, Worldview (WV) -1, 2, and 3, and Synthetic Aperture Radar (SAR; Sentinel 1) to investigate changes in glacier characteristics and dynamics. The MODIS sensor, on the Aqua and Terra satellites, has a data archive from 2002 to present and was used to determine the dates of fast ice formation, seasonal area changes, break-up timing and extent of retreat. The Landsat 8 and 9 Operational Land Imager product was used to assess melt patterns during the 2021/2022 austral season and to determine ice flow speeds using a Python-based image cross-correlation software, PyCorr (Fahnestock et al., 2016). PyCorr measures ice displacement between two images by finding the peaks in normalized cross-correlation surfaces between image chips extracted in a grid pattern over both images. For images separated by one year, using Landsat 8 and 9 panchromatic images (15 m spatial resolution) error is ~±7.5 m yr$^{-1}$, however shorter time intervals result in higher errors (Fahnestock et al., 2016). WV-1, 2, and 3 satellite images have very high resolution (< 0.5 m) and were used for investigating the morphology of icebergs and the creation of digital elevation models (DEMs). Worldview in-track stereo-image DEMs (Table S1) were obtained from the Polar Geospatial Center (PGC). The DEMs have a spatial resolution of 2 m and absolute accuracy of ~4 m in horizontal and vertical dimensions (from PGC documentation). We corrected for the geoid using EGM 2008 and then assessed the mean elevation difference (i.e., bias) for six bedrock regions in each of the WV DEMs relative to the REMA DEM (Howat et al., 2022) and applied the mean offset to the WV DEMs, similar to the method used with the ArcticDEM for the Hunt Fjord Ice Shelf in Greenland (Ochwat et al., 2023a).

We assessed calving styles and approximate grounding zone positions using the imagery and DEM data. In Figure 1a, the yellow dashes show Rott et al.'s (2018) 2016 grounding line, determined by surface elevation changes from TanDEM-X differencing. The black dashes show a slope change and calving-morphology inferred 2021 grounding line (this study). Our grounding zone is estimated from a break in slope in the DEMs and morphological changes, such as the appearance of broad

surface undulations suggestive of bottom crevassing and changes in calving style at the glacier front. We do not suggest the
grounding line advanced between the two estimates, but arise from the two different determination methods. Our grounding
zone position is similar to the partial grounding zone proposed by Sun et al. (2023) and Tuckett et al. (2020); where a presence
of an ice plain can explain why there are multiple determinations of a grounding zone (Friedl et al., 2019). Calving styles of
grounded ice often show surface slumping or tilting prior to separation, indicative of listric faulting (Parizek et al., 2019),
super-buoyancy (Murray et al., 2015) or ice-cliff stresses (Bassis et al., 2021; Crawford et al., 2021). Further analysis of the
evolution of the grounding zone position and the evolution of calving styles for the lower HGE Glaciers will be assessed in a
later study.

We used Sentinel-1A and -1B SAR data to estimate ice flow speeds. The Alaska Satellite Facility HyP3 Pipeline uses speckle
tracking to create velocity rasters using SAR image pairs. The HyP3 pipeline utilizes GAMMA and auto-RIFT algorithms
through the Vertex On-Demand Processing Tool (Gardner et al., 2018; Lei et al., 2021). The autoRIFT code includes an
iterative process for determining the flow velocity, with varying relative errors that have an average of 4% for both X and Y
direction velocity (Lei et al., 2021). Sentinel-1A and -B have a repeat time of 6 days when used in combination, and 12 days
if only 1A or 1B pairs are used. Sentinel-1B malfunctioned in December 2021, leaving only Sentinel-1A data available after
that date.

We extracted ice speed profiles in a band centered on the Airborne Thematic Mapper (ATM) profiles from Operation IceBridge
for Crane, Jorum, Green, and Hektoria glaciers, generating five profiles that span the central 1 km near the approximate glacier
centerlines. To approximate the mean monthly speed, we averaged the speed profile of two 12-day Sentinel-1 cycles.
**3.2.3 Laser Altimetry**
To study changes in surface ice elevation, we combined the WV image-derived DEMs with ICESat-2 altimetry data. We used
the ICESat-2 ATL06 version 5 product, which provides a linear surface approximation of 40 m overlapping segments along
each ground track (Smith et al., 2021) with a 91-day repeat cycle (clouds permitting). We correct for the geoid prior to
estimating the initial thickness of the fast ice, glacier tongues, and elevation of the glaciers. We used ICESat-2 data for the
period January 2021 to December 2021 (Fig. 1) to determine the initial fast ice and glacier tongue freeboard. To account for
tidal variations, we only used tracks that crossed open water (as assessed in MODIS or Sentinel 1 imagery). For Fig. 1, we
applied offsets to each track according to the reported elevation of the open sea surface at the time of acquisition. Assuming
the proportion of snow relative to ice thickness is low, we calculated fast ice thicknesses from the freeboard using the standard
hydrostatic equilibrium floating ice relationship using a density of 1028 kg m$^{-3}$ for sea water and 900 kg m$^{-3}$ for ice. For
analyzing glacier elevation changes, we extracted ICESat-2 data from where tracks cross < 200 m of near-centerline tracks
flown by Operation IceBridge using the ATM sensor, and averaged the data. Standard error analysis was performed on the
individual WV and ICESat-2 elevations. We use the square-root of the sum of the squares of the error, where the errors are the
standard error of the mean and the instrument error of WV or ICESat-2.

### 3.2.4 GNSS data from AMIGOS station on Scar Inlet

An Automated Meteorology-Ice-Geophysics Observing System (AMIGOS) unit with a dual-channel GPS receiver was placed
on the Scar Inlet Ice Shelf in February of 2010 (Scambos et al., 2013) as part of the Larsen Ice Shelf System, Antarctica project
(LARISSA; Wellner et al., 2019). The system provided hourly position data spanning February 2010 through August 2017,
with several data gaps due to power and system malfunctions, that were periodically repaired during re-visits. Precision of the
hourly position data was approximately ±20 cm due to wind on the tower mounting of the GPS antenna. We used daily, weekly,
and monthly averaged data to evaluate ice flow of the Scar Inlet Ice Shelf over the formation and thickening period of the
adjacent fast ice.

### 3.2.5 Aerial photography

To evaluate how the fast ice break-up occurred and the potential calving styles of the outlet glaciers, we also analysed airborne
photography. On 31 January 2022, the British Antarctic Survey flew a Twin Otter over the study area with a digital camera
(Panasonic DMC-TZ80e) and a series of photos of the glacier fronts and ice tongue areas were taken along with approximate
geolocation.

### 4 Results

### 4.1 Multi-year fast ice in the Larsen B embayment

### 4.1.1 Formation and evolution of Larsen B multi-year fast ice

The fast ice that formed in March 2011 had a few partial retreats during the 11 years that the interior embayment region was
continuously covered. Portions of the fast ice broke out and reformed in May 2011 and March 2012. From March 2012
onwards, the fast ice maintained a minimum area of ~3975 km$^2$ (based on MODIS imagery). From 2012 to 2016 the fast ice
cover was relatively stable with a maximum area of ~6280 km$^2$ in 2016. After 2016, the eastern portion of the fast ice, ~1200
km$^2$ in area, seasonally re-formed and broke out. The lowest extent in the MODIS record was in February 2019, when a slightly
larger area (2000 km$^2$) broke out. In late 2021 the outer portion broke out again, returning to the 2019 areal extent, and by
early 2022 the extent was similar to the 2019 minimum (Supplemental Video. 1). Lateral rifts appeared in ~2016 near the
confluence of Scar Inlet Ice Shelf and Crane Glacier extending north and south from Cape Disappointment. Early assessments
of the fast ice thickness in the Larsen B embayment near the oceanward ice front were 2.5 to 4 m (Scambos et al., 2017). In
the inner embayment, altimetry data indicate a thickness of tens to hundreds of meters in areas of mélange containing fast ice
and glacier tongue ice (Fig. 1; Fig. S1 and S2).

### 4.1.2 Upstream glacier response to fast ice formation

During the 2011-2022 period of fast ice presence in the embayment, changes in the glacier extents and GNSS data suggests that the fast ice stabilized the Larsen B tributary glaciers and buttressed the Scar Inlet Ice Shelf, relative to the state prior to the fast ice occupation. This corroborates the findings of Christie et al. (2022). MODIS and Landsat images show that the glacier tongues readvanced into the embayment, and the fjords became a floating composite of glacier ice, large icebergs, and fast ice (hereafter, 'mèlange'). The ice reached thicknesses of up to 320 m near the glacier tongue termini (inferred from freeboard estimates in Fig. 1). Crane Glacier's terminus and associated mélange advanced at ~1 km yr$^{-1}$ (11 km total) and the main trunk of the glacier thickened (Rott et al., 2018; Needell and Holschuh, 2023). The HGE floating tongue and mélange reformed into an ice-shelf-like feature with a central freeboard exceeding 40 m. HGE advanced approximately 20 km from February 2011 to January 2022, with the new floating mixed-ice-type area covering ~250 km$^2$. Jorum Glacier advanced ~4.5 km over the same period, while Punchbowl Glacier only readvanced ~0.5 km and did not create an extensive mélange or glacier tongue.

GPS data from Scar Inlet Ice Shelf (Fig. 2) show an acceleration of ice shelf flow speed from installation of the GPS in early 2010 to late 2012, followed by a cyclical variation in flow speed that varied by season. This indicates that the ice shelf was accelerating prior to the formation and thickening of the multi-year fast ice. From late 2012 onwards, the acceleration of the ice shelf was halted, and an annual cycle with faster flow during late summer-early autumn and a springtime minimum flow speed was observed. We infer that significant buttressing of the ice shelf by the fast ice mitigated the acceleration of the shelf. Moreover, the seasonal cycle of flow speed, with highest flow speed during mid-summer, is interpreted as an effect of seasonal weakening of the fast ice plate due to summer warmth (Pettit et al., in prep).

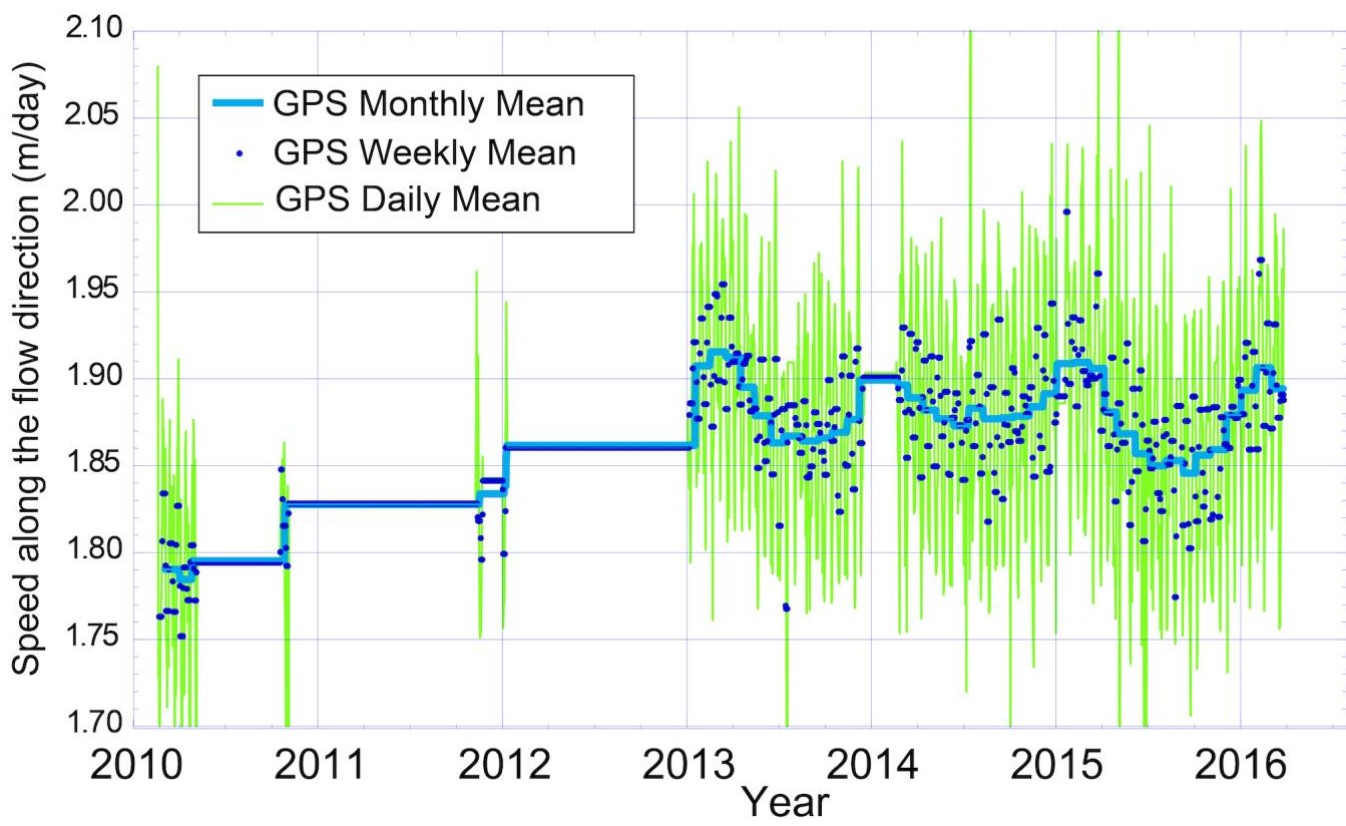


*Figure 2: Scar Inlet Ice Shelf ice flow speeds from AMIGOS GPS from 2010 to 2017. Blue line is the monthly mean, blue dots*
*are the weekly means, and green vertical lines are the daily means.*
**4.1.3 Multi-year fast ice break-up**
MODIS imagery shows that new narrow fractures started to form in the fast ice between 18 and 19 January 2022, widening
thereafter, and by 20 January the fast ice area was densely fractured and no longer coherent. By 21 January, floes derived from
the fast ice plate had drifted 9 to 16 km northeast into the Weddell Sea, exposing the tributary glacier fronts to open water (Fig.
1c). The fast ice floes continued to drift away, fully clearing the embayment by 8 February.

Pack ice began to reappear in the embayment in March 2022, but overall sea ice cover was not persistent through the next 12
months. Over the course of the late austral summer into the autumn and winter, MODIS images indicate overall sea ice cover
in the embayment varied in extent and apparent coherency. Open water conditions in the embayment and the area adjacent to
the AP and James Ross Island persisted through March 2022. Landfast ice did not form in the embayment during the southern
hemisphere autumn and winter 2022. In October 2022 the sea ice in the embayment varied in spatial extent, and began to
decrease significantly in November 2022, and by December 2022 there were minimal floating bergs or pack ice floes. From
January to March 2023 the embayment was devoid of floating ice and remained open ocean. However, by the end of March
2023, pack ice and fast ice started to reform in the embayment.

**4.2 Potential Attributions of the 2021-2022 Fast Ice Breakout**

**4.2.1 Seasonal meteorological conditions**

For November 2021 to January 2022, there is no substantial precipitation anomaly in our study area (Fig. S3A). The wind
speed anomaly composites indicate a slightly higher than average wind speed during the 2021/2022 melt season, with
December having the largest anomaly, primarily in the Bellingshausen Sea (Fig. S3B). The temperature anomaly over this
period indicates the Bellingshausen Sea was slightly warmer (~2°C) than the 1979 to 2022 climatological average, whereas
the Larsen B embayment was up to 4°C warmer (Fig. 3).

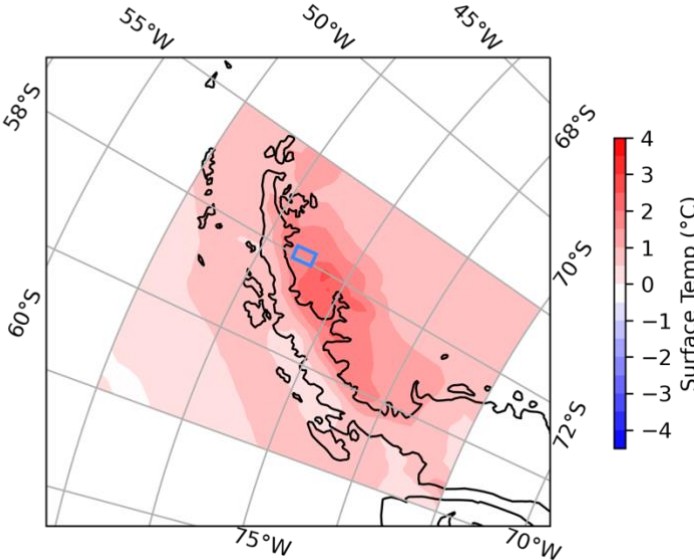

*Figure 3: ERA-5 surface air temperature anomalies around the Antarctic Peninsula. The blue box is the area of grid cells used for the foehn wind analysis (Fig. S4).*

We also looked at the January 2022 mean hourly values of several meteorological variables that indicate foehn wind events:
temperature, windspeed, wind direction, relative humidity and net ablation for the Larsen B region (blue box, Fig. 3), (Fig.
S4). Five identified foehn events occurred from 17 to 21 January 2022, two prior to the fast ice break-out, one during that
event, and two after it. These events likely enhanced surface melting on the fast ice, potentially augmenting the break-up of
the ice in the post-break-up days (19 and 21 January) and dispersing the fast ice floes northeastward in the following weeks.
Since ARs can be linked to foehn events and therefore to increased surface melting (Bozkurt et al., 2018), we also investigated
AR occurrence in the period of the fast ice break-out event. A time series of IVT in the Larsen B region indicates that IVT
associated with an AR event from the northwest begins to increase on 19 January and peaks on 20 January 11:00 UTC (Fig.
4a and b; Wille et al., 2022). IVT remained high until 22 January, when the AR weakened and dissipated. This event occurred
simultaneously as the series of foehn events from 19 to 22 January, suggesting the AR led to the foehn winds that occurred
just after the initiation of the break-out.

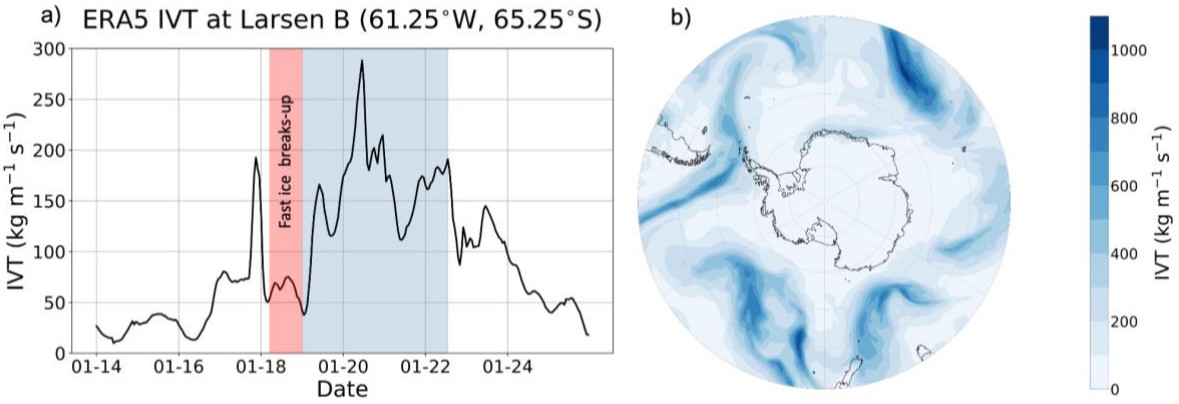


*Figure 4: a) time series of IVT for January 2022 at 65.25°S, 61.25°W. b) map of ERA-5 IVT in the southern hemisphere at 11:00 UTC on 20 January 2022, during the peak IVT at Larsen B. The AR is identified as a long filament of high IVT that extends from the eastern Pacific across the Antarctic Peninsula and into the Atlantic Ocean. Red shading indicates the arrival of the swell and fast ice break-up. Blue shading indicates the duration of the AR event over Larsen B.*

### 4.2.2 Surface melt

Figure 5 shows cumulative melt days for each melt season from 2012/2013 to 2020/2021 over the Larsen B multi-year fast ice
and Scar Inlet Ice Shelf, derived from AMSR-E/2 passive microwave data. Fig. 5a shows a map of the grid cells used in the
analysis, as well as cumulative melt days for the 2019/2020 season, 2020/2021 season (i.e. the two melt seasons preceding the
break-up event), and the mean cumulative melt days for each season from 2012/2013 to 2020/2021. We do not include the
2021/2022 melt day data in Fig. 5a because of the mid-season break out of the fast ice in that summer. Maps of cumulative
melt days for all melt seasons are available in Fig. S5. Fig. 5b shows the spatially-averaged melt days over the study area, as
well as the cumulative days when the melt area was 100% of the study area, for nine melt seasons leading up to the break-up
event, as well as the melt season with the fast ice break-out (2021/2022).
The 2021/2022 season did not have a particularly long or spatially more extensive melt season relative to the previous nine
melt seasons. Of the years studied, 2019/2020 had both the longest melt season and the one with the highest number of days
with 100% melt area; nonetheless the fast ice survived this season, as well as the preceding high-melt years.
In addition to our analysis of passive microwave data (above), which may indicate the presence of surface meltwater ponding
(e.g. Picard et al., 2022), we also analyzed optical satellite images for evidence of surface meltwater ponding. Landsat 8 images
in November and December 2021 show the surface of the fast ice was extensively covered with melt ponds (Fig. S6). However,
by January 2022, the surface melt ponds on the fast ice appeared to have refrozen, and melt pond extent was reduced (Fig. S6).

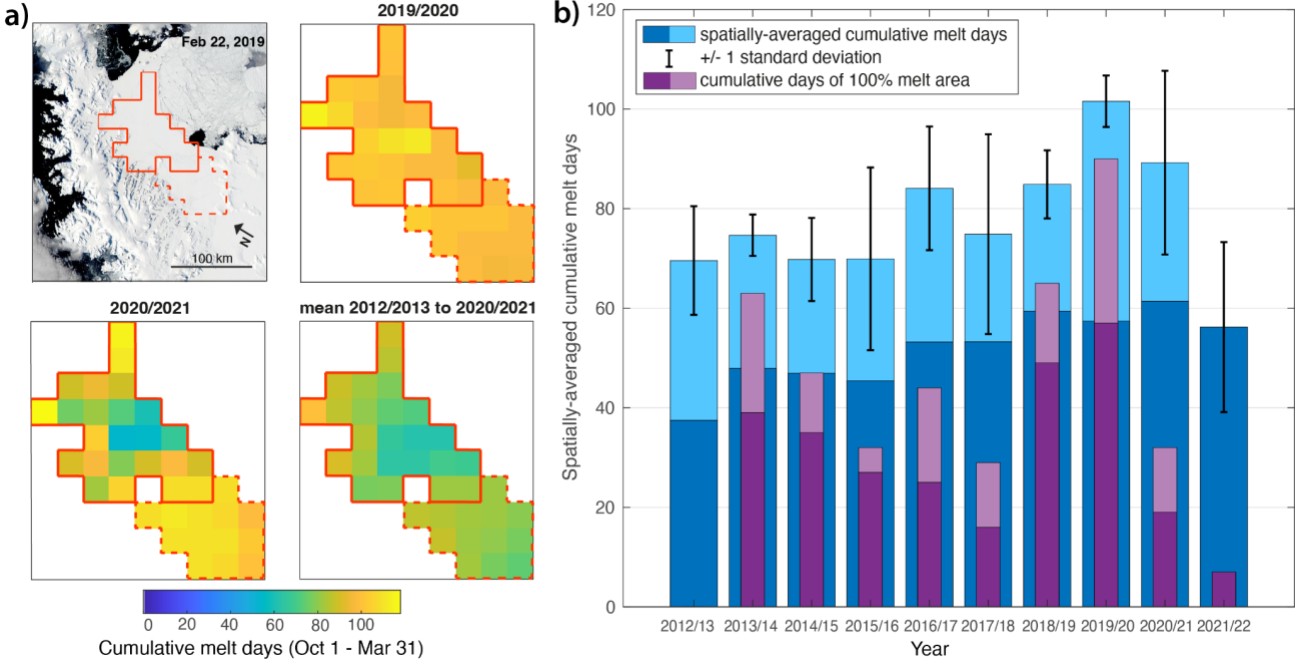

*Figure 5: Cumulative melt days derived from AMSR-E/2 passive microwave melt data. a) Cumulative melt days over the fast*
*ice area in the Larsen B embayment (area within solid red lines) and over the Scar Inlet Ice Shelf (area within dashed red*
*line) for the 2019/2020 and 2020/2021 melt seasons, and the mean from 2012/2013 to 2020/2021. b) Spatially-averaged melt*
*days (blue shades) and cumulative days of 100% melt area (purple shades) over just the Larsen B embayment fast ice (solid*
*red lines in panel a) from 2012/2013 to 2021/2022. The dark purple and dark blue bars show cumulative melt days from just*
*1 October through 18 January (i.e. the data available for the 2021/2022 season), and the light purple and light blue bars show*
*cumulative melt days from 1 October through 31 March.*
**4.2.3 Regional Sea ice Cover**
Figure 6a displays a mapping of sea ice concentration from AMSR-E/2 data in the Weddell Sea on 19 January 2022. Fig. 6b
shows a time series of overall sea ice area (concentration that is greater than 15% multiplied by area of pixel) for the date of
19 January for each year from 2010 to 2022 in a selected region (gray box in Fig. 6a; 2011/2012 did not have AMSR-E/2
sensor data on this date; MODIS imagery shows extensive sea ice cover in the Larsen B fast ice front area through this time).
The selected region represents a potential ocean swell corridor leading to the Larsen B embayment from 2010 to 2022 (see
Section 5.2; also Teder et al., 2022). For the 8-year period (2013 to 2020 inclusive) the overall sea ice area in this region of
the northwest Weddell Sea was over 125,000 km$^2$ (>50% of the box area). In 2011, sea ice area was just 100,000 km$^2$ on 19
January; however, we note that the fast ice formed later in this year (March). The overall sea ice area dropped in 2021 to 75,000
km$^2$, and in 2022 its area was just below 40,000 km$^2$. As Fig. 6a shows, a corridor is present along the eastern side of the
Peninsula in January 2022, which opened on ~8 January 2022 according to the MODIS and AMSR-E/2 record. This pathway,
which allows for wave action to access the front of the Larsen B fast ice, had not been present since the fast ice's formation in
2011 (Figs. 6b and S7).

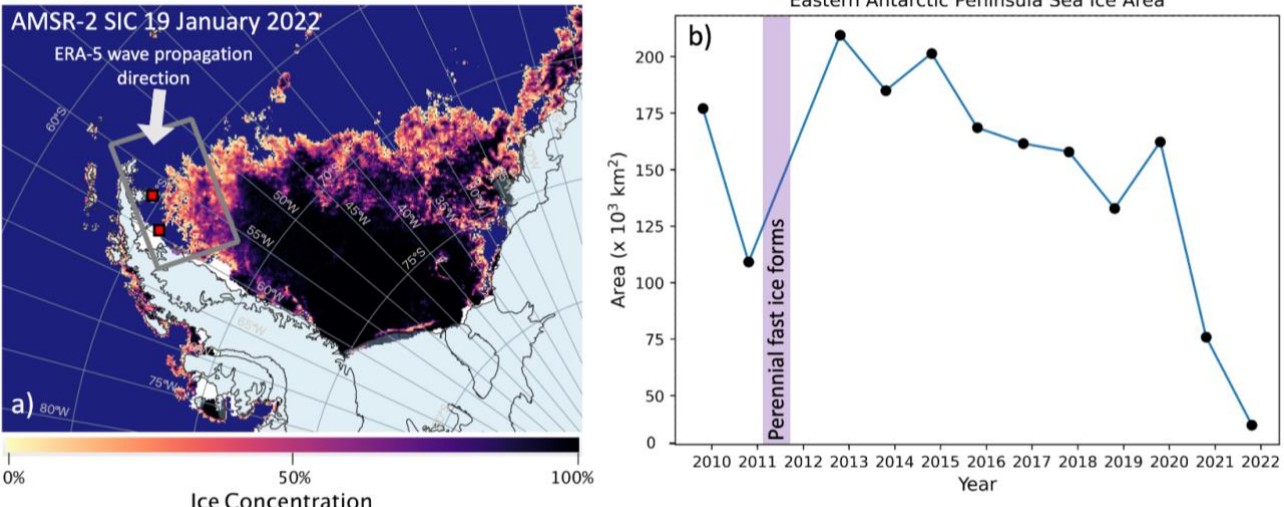

*Figure 6: a) Pack sea ice concentration and distribution map on 19 January 2022 from AMSR-2 data (Spreen et al., 2008).*
*Small red squares show the location of the ERA-5 wave height grid cells (Fig. 7). The gray box is the region selected for the*
*overall sea ice area in 6b. The white arrow denotes the wave propagation direction on 19 January ERA-5 data. b) Overall sea*
*ice area (concentration in each grid cell multiplied by the grid cell area) from AMSR-E and AMSR-2 data in the corridor*
*region of the NW Weddell Sea for 2010 to 2022. Error in sea ice concentration according to Spreen et al. (2008) is ~7%.*
*Purple vertical shading indicates the time period of fast ice formation.*

### 4.2.4 Wave action

Examining both ERA-5 and WaveWatch-III wave data, the first large swell able to pass through the open-ocean (sea ice-free) corridor and reach the Larsen B fast ice edge occurred on 18 and 19 January (Figs. 7 and S8). In the early hours (UTC) of 18 January 2022, the significant wave height averaged ~0.1 m in the selected grid cell region. By the afternoon on 18 January the average wave height rose steeply to a maximum of 1.75 m near Larsen B and to over 2 m near James Ross Island ~150 km to the northeast (red boxes, Fig. 6). Simultaneously, the peak wave period increased to ~5 s, indicating a wavelength equivalent to ~40 m. The wave propagation direction was bearing ~250° ± 25° through this period, similar to the orientation of the open corridor in the pack ice. There were no events in November or December 2021 that included both a long peak period and a high significant wave height (and in any case, pack ice damped wave propagation in the region until ~8 January). Both months have peak periods consistently less than 6 s and wave heights below 1.4 m (Fig. S8 and S9). Furthermore, there were no other times during January 2022 when the wave swell had both a long peak period and high significant wave height (Fig. 7). Abrupt shifts in peak period and significant wave height (see Methods) are evident when the wave corridor opens near James Ross Island (gray band 8 Jan 2022) and when the event occurs (gray band 18 Jan 2022), as well as when the wind direction changes (Fig. S4).

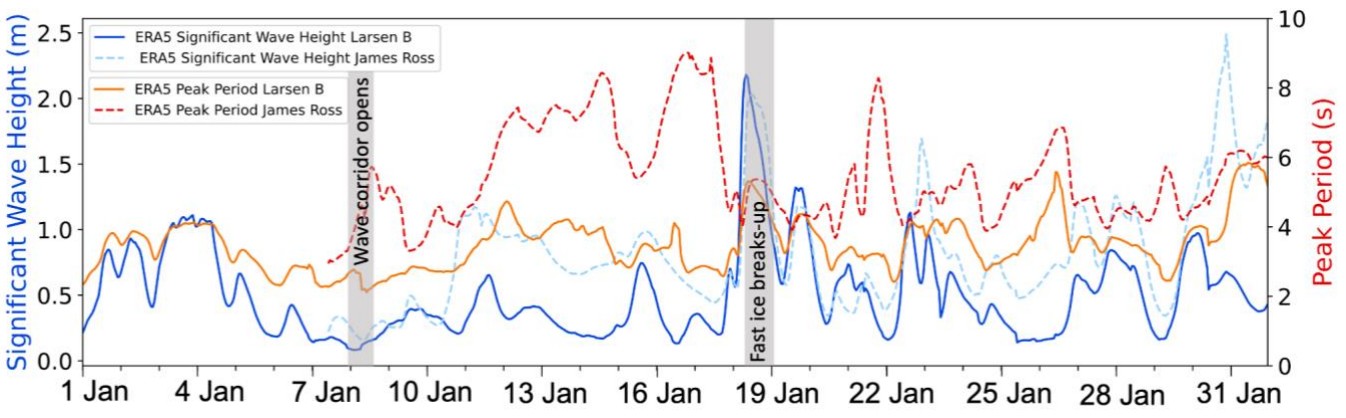

*Figure 7: ERA-5 significant wave height (blue) and peak period (red) for both the Larsen B area (solid lines) and near James Ross Island (dashed lines) during January 2022. The red and dark red lines and the blue and light blue lines correspond to the peak period and significant wave height, respectively. The opening of the wave corridor and fast ice break-up are denoted by the gray vertical bands.*

**4.3 Initial glacier response to fast ice break-out**

**4.3.1 Retreat of glacier fronts**

Four glaciers along the Larsen B embayment coast responded almost immediately to the fast ice break-out. Crane and Jorum Glaciers exhibited similar responses, losing most of their floating ice tongues within days of the fast ice breakout (Fig. 8a) and calving a number of large (several km$^2$) full-thickness tabular icebergs. Once the floating tongue portion was removed, both glaciers underwent buoyancy-driven calving and a tidewater-style retreat at their grounding zones, indicated by the presence of toppled icebergs in optical images and high-backscatter iceberg surfaces in Sentinel-1 data. Scattering intensity is related to surface roughness as well as how much melt has affected the surface of the berg; freshly toppled cold bergs will have a brighter surface, whereas tabular bergs that have been exposed to surface melt will display a decreased backscatter intensity (Young et al., 1998). Punchbowl Glacier began calving in a style that appears to be buoyant full thickness calving (Murray et al., 2015), indicated by toppled dark blue icebergs. Unlike Crane, Jorum, and HGE, Punchbowl did not readvance into the embayment during the fast-ice occupation. Hektoria and Green Glacier retained a 13 km extended thick (greater than 300 m) floating tongue after the immediate break-out, until March 2022. However, at this point their floating ice areas also underwent full-thickness tabular calving with occasional toppled icebergs (Fig. 8). From April to October 2022 the ice fronts were relatively stable, but rapid retreat reinitiated in November 2022. The calving style resembled tidewater glacier retreat for grounded ice with buoyant calving, similar to the Röhss Glacier response from the loss of the Prince Gustav Ice Shelf (Glasser et al., 2011) or calving regimes at Helheim Glacier, Greenland (Murray et al., 2015).

In the weeks and months following the fast ice break-up, Crane, Jorum, and Punchbowl glaciers continued to retreat. By 8 February 2022 the Crane Glacier floating front (defined here as the limit of contiguous ice > 100 m in thickness; consistent with Needell and Holschuh, 2023) had retreated more than 6.5 km and was still calving large tabular bergs (several km$^2$ and > 300 m thick, based on WV DEMs; Fig. 8a). From 8 February until 11 March 2022 only 400 to 800 m of retreat occurred. Crane continued its episodic periods of retreat of several hundred meters at a time throughout the 2022-2023 summer season (Fig. 8a). Its retreat totalled ~11 km, of which possibly 1 to 2 km was grounded ice using this study's grounding zone or no grounded ice using Rott et al. (2018)'s, 2016 grounding line (Fig. 8b). Similar to Crane in calving style, the Jorum Glacier main trunk lost ~5 km of floating ice and its (former) tributary branch glacier lost ~6 km. Punchbowl Glacier, in contrast, has only lost a few hundred meters of its ice front as of May 1 2023.

Hektoria and Green Glacier responded to the fast ice break-out in later months. Hektoria Glacier had an extended thick (> 300 m) floating tongue that persisted until 12 to 17 March 2022, when it retreated ~7 km (Fig. 8c). From 26 to 30 March, Hektoria's floating tongue retreated another ~6 km, exposing an arcuate ice front. From April 2022 until August, Hektoria's ice front retreated ~1 km. This retreat is inferred to be the start of the grounded ice retreat based on a change in calving style and surface morphology of the upstream ice. For all of September and October Hektoria's front did not change. Hektoria retreated ~3 km

by 14 November and another 1.2 km by 30 November 2022. In December 2022, Hektoria underwent another series of retreats totaling ~4 km. From 17 January to 15 March 2023, another ~1.5 km retreat into the fjord occurred, and Hektoria is still actively retreating as of April 2023 and has retreated a total of ~25 km, of which ~10 km may have been grounded ice using this study's grounding zone or 1 to 5 km of grounded ice using Rott et al., (2018)'s 2016 grounding line (Fig. 8d). Green Glacier has also retreated substantially but not as far into its fjord. Following a similar timeline to Hektoria, Green has retreated ~18 km total.

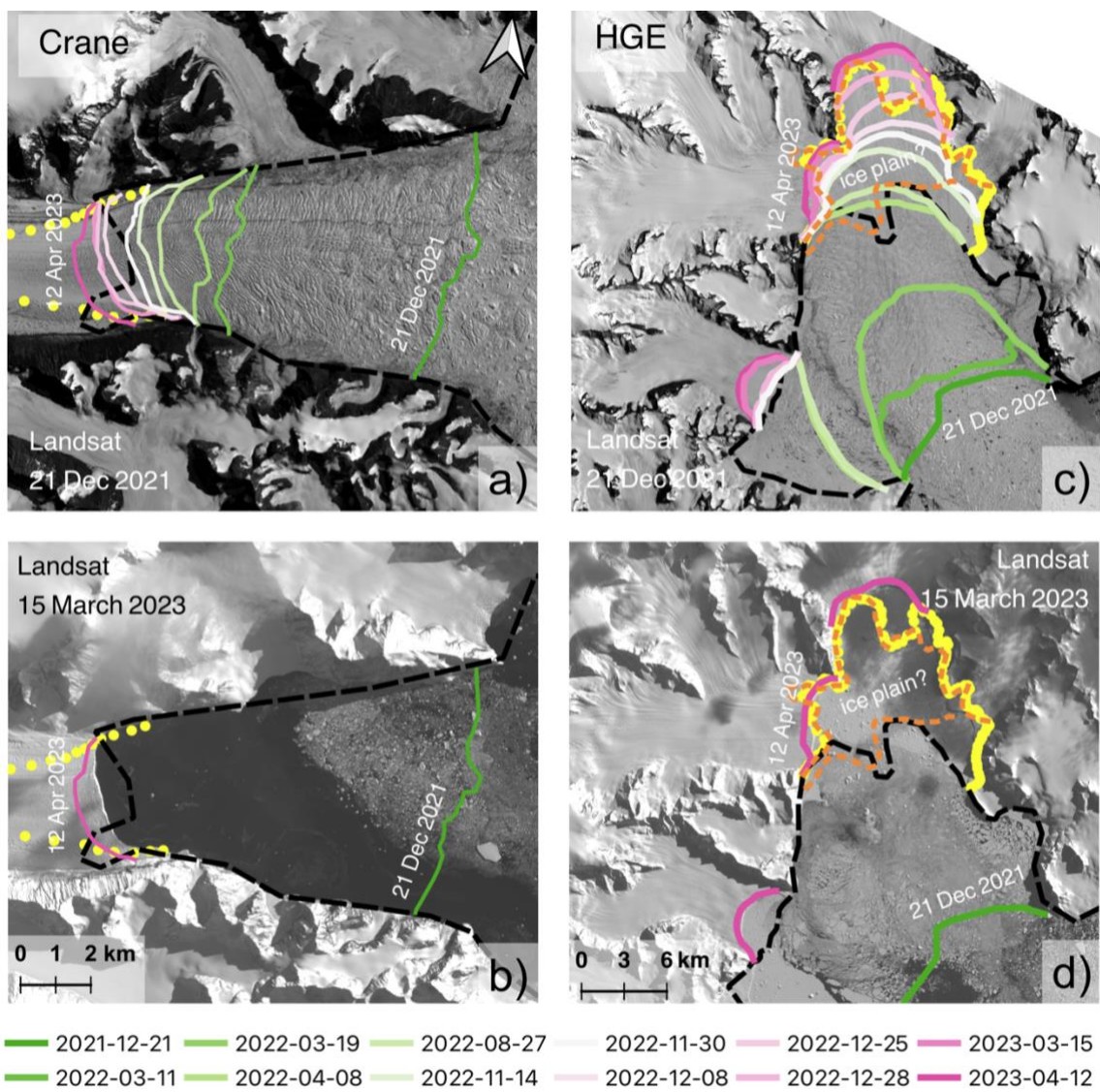

| | | | | | |
|---|---|---|---|---|---|
| — 2021-12-21 | — 2022-03-19 | — 2022-08-27 | — 2022-11-30 | — 2022-12-25 | — 2023-03-15 |
| — 2022-03-11 | — 2022-04-08 | — 2022-11-14 | — 2022-12-08 | — 2022-12-28 | — 2023-04-12 |

*Figure 8: Yellow points are Rott et al., 2018's grounding zone, black dashed lines are this study's inferred grounding zone, orange dotted lines are Tuckett et al., 2020's grounding zone and ice plain location; a) Crane Glacier retreat fronts from*

*November 2021 to April 2023, length scale is the same as panel b; b) Crane Glacier with pre-break out terminus position and*

*April 2023 terminus position; c) HGE system retreat fronts from November 2021 to April 2023, length scale is the same as*

*panel d; d) HGE system with pre-break out terminus position and April 2023 terminus position. For a) and c) the background*

*is a Landsat 8 image from 21 November 2021, and for b) and d) the background is a Landsat 9 image from 15 March 2023.*

### 4.3.2 Glacier centerline speed changes

Initial ice flow speed profiles along near-centerline tracks of Crane, Jorum, Green and Hektoria glaciers all show an increase in speed of various magnitudes since the fast ice break-out event. For all the glaciers besides Punchbowl, the floating portions increased in speed dramatically immediately after the break-out event while the grounded portion of the glaciers took many months to be affected, according to this study's grounding zone estimation (Fig. 9a-c; gray shaded bands on profiles and dashed white lines on insets). Additionally, the observed speed profiles in the 26-month period (January 2021 to March 2023) show far less local variability upstream of our inferred grounding line.

The Crane Glacier tongue accelerated and extended along-flow immediately after the event leading to an increase of speed from 1000 m yr$^{-1}$ to 1300 m yr$^{-1}$ within the first two months (Fig. 9a; light blue to yellow-green solid lines). The grounded portion of Crane Glacier responded in the months following. By November 2022 the grounded ice speed increased from 800 to 900 m yr$^{-1}$ (Fig. 9a; yellow to dark-yellow solid lines) and by March 2023 the speed was 1200 m yr$^{-1}$ (Fig. 9a; red solid lines). Crane Glacier is still undergoing retreat and acceleration as of March 2023.

Jorum Glacier did not experience as dramatic a change in speed after the event. Jorum Glacier has three flow speed sections: the upper glacier was slow-moving at 100 to 200 m yr$^{-1}$, the glacier's steep portion accelerated to 500 m yr$^{-1}$ over a distance of 1.5 km, and the lower glacier flowed at ~475 m yr$^{-1}$ prior to the break-out (Fig. S10). By November 2022, this lower section increased in speed by ~75 to 100 m yr$^{-1}$, and has remained at ~550 m yr$^{-1}$ as of March 2023 (Fig. S10). Jorum Glacier's floating tongue quickly calved away after the event so the floating icebergs and loose mélange were not tracked for speed.

The HGE system experienced significant changes after the break-up of the fast ice. While the floating portion of the system did not experience speed changes immediately after the fast ice loss (Fig. 9a-c; January to March 2022; light blue to yellow-green solid lines). The floating tongue was removed by April 2022 leaving only grounded ice, according to this study's grounding zone estimation. The mélange speed is tracked when the mélange is cohesive (solid lines downstream of this study's grounding line from April 2022 onwards), reaching ~1500 (Hektoria) and 1700 m yr$^{-1}$ (Green) by October 2022.

Speed changes occurred in the lower trunk areas of both Green and Hektoria glaciers. Green Glacier increased from ~500 m yr$^{-1}$ prior to fast ice break-up to 1150 m yr$^{-1}$ by January 2023. Green Glacier's SAR-derived and Landsat-derived ice speeds for December 2022 and January 2023 agree with the general trend (Fig. 9b; light brown and brown solid lines). Hektoria

Glacier's Landsat-derived ice speeds show a velocity increase 10 km upstream of the grounding line from September 2022 to January 2023 from 400 to 900 m yr$^{-1}$ (Fig. 9c; light brown and brown solid lines). Both Green and Hektoria are still undergoing retreat and acceleration as of March 2023.

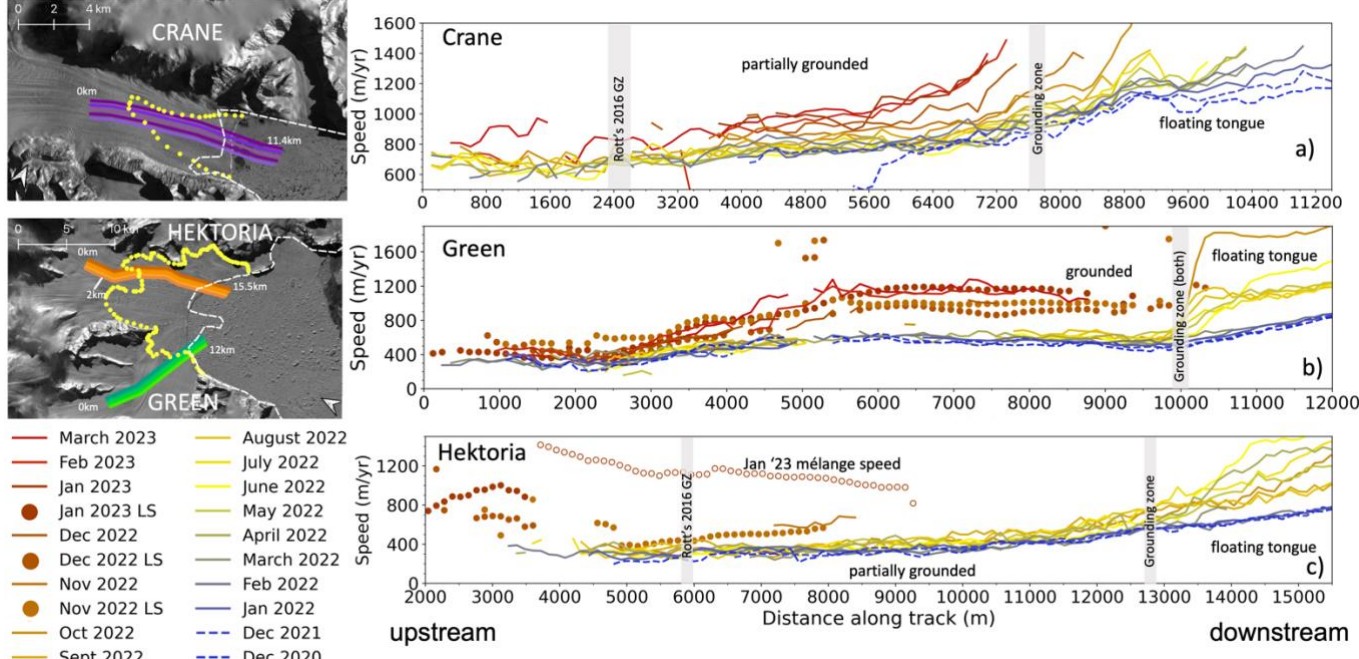

*Figure 9: Monthly averaged ice flow speeds along the IceBridge flight centerlines, derived from Sentinel-1 speckle tracking from Alaska Satellite Facility HYP3-pipeline, solid-colored lines. Image pair flow speeds from Landsat imagery (using PyCorr) are indicated by colored solid dots. Gray bands on profiles and dashed white line on image insets show inferred grounding zones, with the Rott et al. (2018) grounding zone of 2016 as yellow points on insets. a) Crane Glacier velocity profile. b) Green Glacier velocity profile. c) Hektoria Glacier velocity profile. The open circles represent the melange speed from the January 2023 Landsat velocity data. The along-track distances are set at an arbitrary point well upstream of each glacier's grounding zone. Blue dashed lines are reference years Dec 2021 and Dec 2020, prior to break-out. Background image is a Landsat 9 image from 06 October 2022.*

### 4.3.3 Elevation changes

We used ICESat-2 altimetry and WV-1, -2, and -3 stereo-image DEMs to assess elevation changes of the Larsen B embayment glaciers from 2017 to present. For each glacier, we evaluated three reference points along the near-centerline to these changes. Lower Crane Glacier (red box, Fig. 10) may have thinned by up to 16 m immediately after the fast ice break-out, however the trend is incomplete due to the glacier's rapid retreat and calving. This abrupt thinning may have been a consequence of a change in calving style near the front, e.g., listric faulting in the ice (e.g., Parizek et al., 2019). Our trend for the middle and upper section of Crane (orange and blue box, Fig. 10) shows thickening from 2017 to 2022, consistent with Needell and

Holschuh's (2023) findings. Thinning may have now begun in those regions, however the data are inconclusive, as the thinning is only 1 to 3 m as of February 2023, which is within the measurement error and surface roughness variations on the glacier. Jorum and Punchbowl glaciers show very variable results as well (Fig. S11).

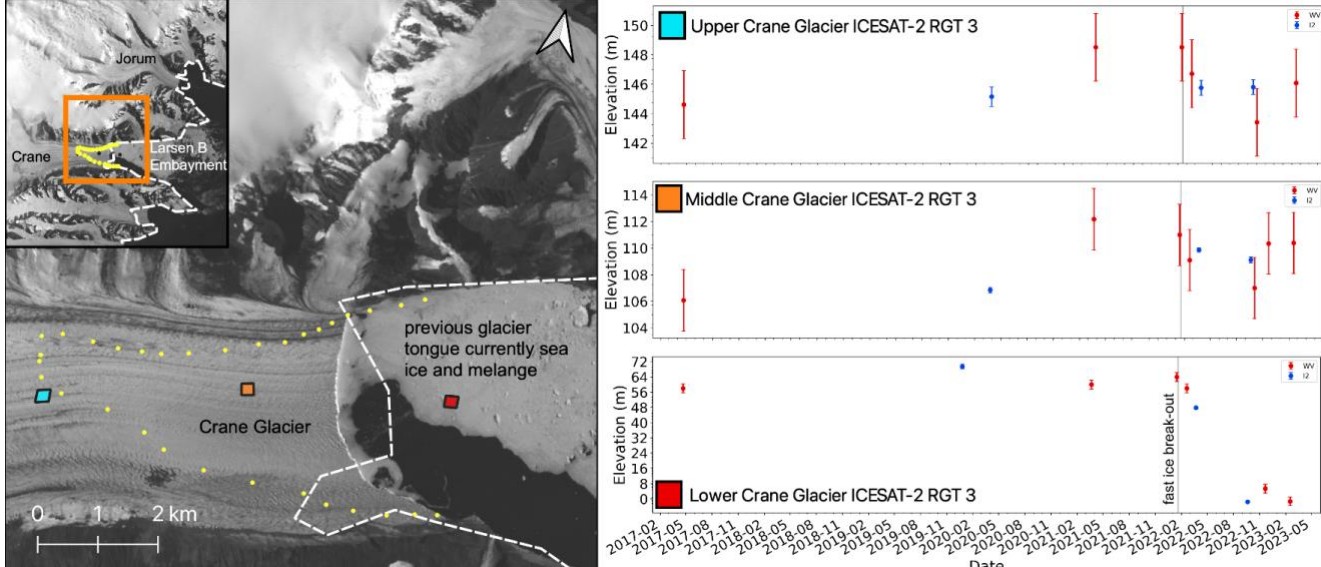

*Figure 10: Crane Glacier near-centerline elevation changes through time. Background image is from Landsat 9 17 January 2023 Landsat. The time series plot corresponds to the area of the box of the same color. The gray band indicates the date of the fast ice break-out event. Note the different vertical scales on the figure.*

The HGE system shows thinning in various regions and rapid calving and retreat in the elevation data.. Figure 11 (yellow box) shows the HGE system floating tongue freeboard as 40 m, which is consistent with a ~320 m glacier thickness, assuming hydrostatic equilibrium. After the break-out event, icebergs (which we define as ice with >5 m freeboard) are present in the fjord until the open ocean period (December 2022). Lower Hektoria Glacier lacks elevation change data points from the start of the collapse, simply because the downstream-most regions calved and drifted away before a repeat elevation measurement could be acquired (blue and green boxes; Fig. 11). The upper portion of Hektoria (dark orange box; Fig. 11) appears to have thinned since early 2018 (or prior). Minimal thinning occurred from April 2022 to late December 2022. As of 6 April 2023, this portion of the glacier is just 400 m upstream from the rapidly retreating glacier terminus and is unlikely to remain intact for further measurements. Both the lower and upper Green Glacier (orange and blue boxes, respectively; Fig. 11) show strong thinning above the level of surface variability from 2017-present. Lower Green Glacier thinned ~11 m between March 2022 to late December 2022, going from 79 m to 68 ±2.3 m. Upper Green Glacier thinned ~9 m between January 2021 to late December 2022, going from 162 ±0.5 m to 153 ±2.3 m. There are no data available between January 2021 and July 2022, so

the initiation of thinning of the glacier is uncertain. However, from July 2022 until late December 2022, ~5 m of the total 9 m of thinning occurred.

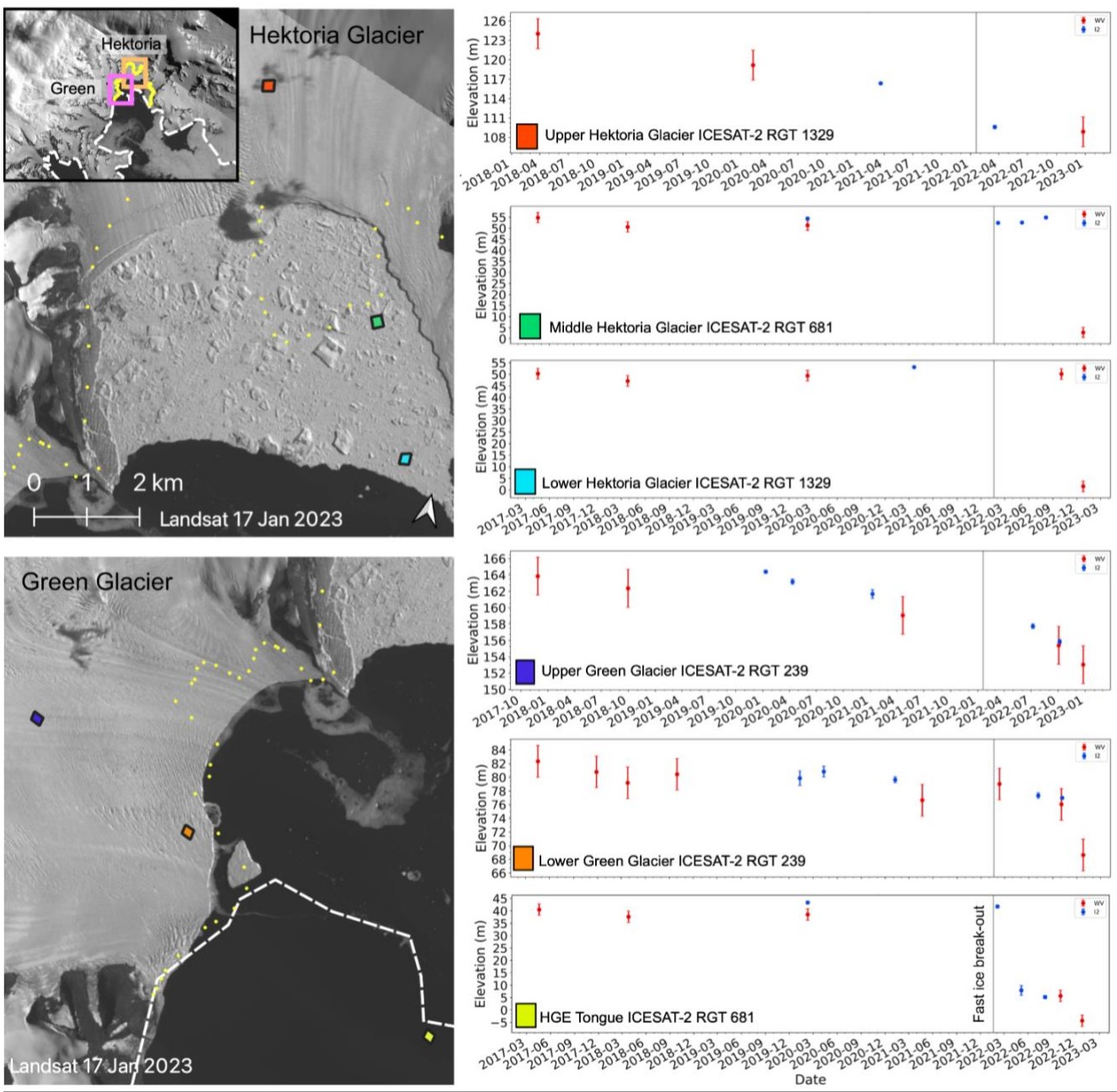

*Figure 11: Hektoria and Green Glacier system elevation changes through time. Background image is from Landsat 9 17 January 2023 Landsat, yellow points indicate the Rott et al. (2018) 2016 grounding zone and white dashed line is the grounding*

*zone inferred in this study. The pink box in the study area inset is the area depicted for Green Glacier and the orange box is Hektoria Glacier. The time series plot corresponds to the area of the box of the same color. The gray band indicates the date of the fast ice break-out event.*

**5 Discussion**

Figure 12 summarizes the chronology of events in the 22-year period from 2001-2023 of the Larsen B embayment and the tributary glaciers. We document the changes that have occurred during the two break-out events and how the glaciers and the Scar Inlet Ice Shelf have responded to those events. Below we discuss the conditions and aftermath of the 2022 event in light of our findings and related literature.

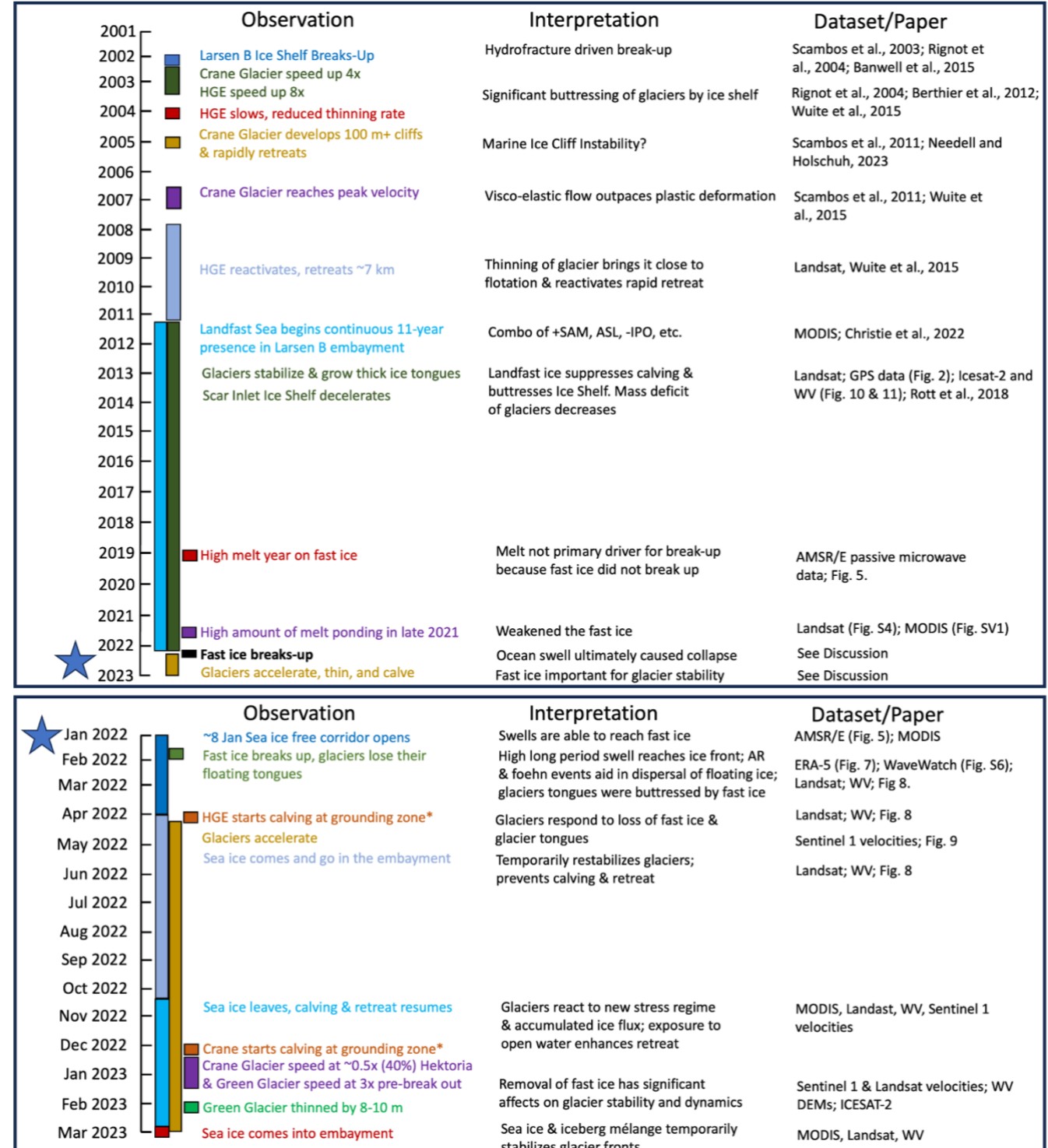

497
498     *Figure 12: Schematic of the chronology of events from 2001-2023 of the Larsen B embayment region discussed in the text.*

## 5.1 Meteorological Conditions and Modes of Atmospheric Variability

The AP climate is influenced by several large-scale modes of atmospheric variability. These patterns are drivers of the formation and demise of pack ice, fast ice, the mass balance and stability of the glaciers and ice shelves. Climate patterns and variability are driven by several modes with a variety of time scales, e.g., the Interdecadal Pacific Oscillation (IPO) at 10 to 30 years and SAM phase oscillations, changing on the scale of weeks to months. The IPO was in a negative phase from 2000 to 2014, favoring an increase of overall sea ice extent at ~$0.57 \pm 0.33 \times 10^6$ km$^2$ per decade (Meehl et al., 2016). Additionally, slightly cooler conditions around the AP in the 2010s limited the area of melt ponding on the Scar Inlet Ice Shelf and the northern Larsen C (Cape et al., 2015; Bevan et al., 2018). This situation paired with intensified cyclonic circulation in the Weddell Sea (Christie et al., 2022), which may be broadly favorable for the formation of the Larsen B embayment fast ice and advancement of the glacier tongues (Fig. 12), yet due to local variability it can be difficult to pinpoint its exact drivers in a specific season. It appears the IPO reversed in 2015/2016 but that remains to be confirmed (Li et al., 2021). The SAM index has been trending toward more frequent periods of positive phase for many decades (Kwon et al., 2020; Li et al., 2021). A positive SAM is generally associated with a deepening of the Amundsen Sea Low, which subsequently enhances northwesterly flow across the AP, bringing warm air masses and an increase in foehn events into this region (Li et al., 2021; Turner et al., 2022). A positive SAM is also correlated with AR events, due to enhanced moisture fluxes towards the Antarctic Peninsula and a more easterly storm track, which can increase warming on the eastern (lee) side of the Antarctic Peninsula during AR-driven foehn events (Wille et al., 2021; Shields et al., 2022; Wille et al., 2022). Strong westerly winds increase pack ice drift eastward and northward, exposing the AP's eastern coast and ice fronts to open ocean. This may have caused the low sea ice cover in the Weddell Sea in summer 2021/2022 and lack of pack ice in the corridor region in January 2022 (Turner et al., 2022).

We found that the climate of the Larsen B region was anomalously warm from November 2021 to January 2022. However, despite the climate being warmer, the number of melt days, and mean areal extent of melt, over the fast ice derived from the passive microwave data in the 2021/2022 season were not a record (Fig. 5). According to the optical imagery, melt ponds were evident in Landsat 8 satellite images mainly in November and December 2022, however the areal extent of melt ponds in the Larsen B region declined prior to the fast ice break-out event in late January 2022 (Fig. S6). These observations suggest that neither surface melting, nor related hydrofracturing of pre-existing melt ponds in the thick glacier tongues, were a direct cause of the 19 January fast ice fracturing or the subsequent break-up, although they do point to a warmer (and likely weaker) fast ice cover at mid-summer of 2021/2022.

The low overall sea ice concentration and westerly winds from a deep Amundsen Sea Low led to an open water corridor between the pack ice and the eastern coast of the AP (Turner et al., 2022). This led to the region in front of the Larsen B having the lowest total overall sea ice area for the date in 13 years (2010-2022; Fig. 6). In February 2021, there was a smaller open corridor, however, the area was not as wide as in early January 2022 and did not create an unobstructed ice-free lane to Southern

Ocean swell. With low sea ice damping, ocean swell events could impact the eastern fast ice and coastal areas; these have been
shown to destabilize fast ice and ice shelves (Crocker and Wadhams 1989; Langhorne et al. 2001; Banwell et al., 2017; Massom
et al., 2018; Teder et al., 2022). However, swell events in February 2021 were low amplitude or proceeded from the coast to
the northeast, i.e. likely driven by foehn events.

**5.2 The fast ice break-out event**

Despite high temperatures and surface melt and meltwater ponding (Scambos et al., 2003; Banwell et al., 2013), alongside
thinning due to both basal melting and surface melting (Adusumili et al., 2018; Smith et al., 2020), known to be primary drivers
of ice shelf collapse, our analysis shows that the Larsen B multi-year fast ice persisted through warm and high melt years (e.g.,
2019/2020; Bevan et al 2020; Banwell et at., 2021) without breaking up. Until the 2021/2022 melt season, the absence of a
large sea ice-free corridor prevented high ocean swells from the northeast from reaching the fast ice (Fig. 12). Long period
ocean swells, such that the wavelength is substantially greater than the ice thickness, can expose ice shelves and fast ice to
flexural strains (Crocker and Wadhams, 1989;  Langhorne et al. 2001; Banwell et al., 2017; Massom et al., 2018). In our case,
the fast ice was several meters thick, wave height was nearly 1.75 m, and the wave period at the time of the event was 5 to 6
s, corresponding to wavelengths of order of 40 m (Fig. 7). The resulting strains can weaken the outer margins of the fast ice or
ice shelf through plate-bending and fracturing. As the outer margin breaks, the stress is redistributed within the fast ice, possibly
initiating further fractures within the ice (Massom et al., 2018). The fast ice was potentially preconditioned to break-up by the
rifts that were open near the confluence of the Crane Glacier tongue and Scar Inlet Ice Shelf, however, contrary to Sun et al.
(2023) we attribute the break-up to the long period, high-amplitude swell that came from the northeast through the open sea
ice corridor. This order of events and ultimate cause of break-up is similar to what was proposed in Gomez-Fall et al. (2022)
for the collapse of the Parker Ice Tongue.

The rapid removal of the fast ice fragments from the embayment coincides with the presence of foehn winds that were likely
caused by an AR event. Recently, ARs and AR-triggered foehn events have been linked to the collapse of ice shelves due to
their ability to cause extreme surface melting and subsequent hydrofracture (Laffin et al., 2022; Wille et al., 2022). Here, we
found that foehn events happened prior to, during, and after the January 2022 Larsen B wave event. We cannot rule out the
possibility that these foehn events may have sufficiently increased melting on the thick glacier tongues to cause
hydrofracturing, which may have acted to further fracture the fast ice and facilitate break-up. As a secondary driver, the winds
likely hastened the removal of the floating ice or helped create the open corridor for wave entry to reach the ice fronts. These
findings parallel that of Massom et al. (2018) who attributes the Larsen B Ice Shelf break-up in 2002 to the removal of sea ice
due to westerly/north westerly winds that not only enabled an ocean swell to reach the ice shelf but also helped to quickly
evacuate large icebergs and mélange out of the embayment. Additionally, the AR and foehn events in January 2022 may have
affected sea surface slope. It is possible that a sea surface sloping oceanward gravitationally promoted calving and removal of
the floating ice, similar to the calving of large icebergs off the Amery Ice Shelf in 2019 (Francis et al., 2021), the Brunt Ice
Shelf in 2021 (Francis et al., 2022), and Larsen D in 2020 (Christie et al., 2022). We suggest further research to investigate the
presence of cyclone(s) and the sea surface slope during the time of the break-up event.

**5.3 The initial glacier response**

After the break-up of the ice shelf in 2002, the presence of fast ice significantly affected the tributary glaciers' dynamics,
providing sufficient backstress that suppressed calving and permitted the tributary glaciers to form thick ice tongues and
readvance into the embayment during 2010 to 2022 after the initial break-up in 2002 (Fig. 2; Needell and Holschuh, 2023),
corresponding to an Eastern Antarctic Peninsula-wide ice front advance discussed in Christie et al. (2022). Rigid mélange
and/or fast or pack ice can stabilize rifts (Larour et al., 2021), and can allow ice shelves, tidewater glaciers, or floating glacier
tongues to advance, as has been seen in Greenland (Moon et al., 2015), the Parker Ice Tongue (Gomez-Fell et al., 2022), the
Cook West Ice Shelf (Miles et al., 2018), and other areas (Reeh et al., 2001; Massom et al., 2010; Cassotto et al., 2015; Banwell
et al., 2017). During the 2010 to 2022 period of the occupation of the Larsen B multi-year fast ice, the tributary glaciers Crane,
Hektoria and Green readvanced and decelerated (Rott et al., 2018) and the Scar Inlet Ice Shelf decelerated with clear seasonal
variability associated with the presence of the fast ice (Fig 2). The loss of fast ice can affect the seasonal variability of velocity
and calving dynamics of ice shelves, as seen for the Totten Ice Shelf (Greene et al., 2018) and Parker Ice Tongue (Gomez-Fell
et al., 2022). Due to the fast ice moving as a cohesive unit coupled with the lack of iceberg rotation embedded in the fast ice,
suggests a degree of mechanical coupling of the glacier tongues, and Scar Inlet Ice Shelf, to the fast ice, similar to the
relationship of mélange and multiyear fast ice for the Voyeykov Ice Shelf prior to disaggregation (Arthur et al., 2021). The
Larsen B fast ice played an integral role in the growth, deceleration, and stability of the Larsen B outlet glaciers, their floating
tongues, and the Scar Inlet Ice Shelf. Given that the fast ice was 5-10 m thick, it likely provided backstress greater than the $10^7$
N m$^{-1}$ threshold required to suppress calving (Robel, 2017). Contrary to recent modelling results (Sun et al., 2023; Surawy-
Stepney et al., 2023), in absence of any other plausible cause, our observations show the loss of the fast ice led directly to
dramatic dynamical changes in the aforementioned tributary glaciers tongues and thereafter the subsequent grounded glaciers.

According to de Rydt et al. (2015), an "immediate" glacier response is one that occurs < 2 years after an initial event. Here we
see an immediate disaggregation of the glacier tongues after the fast ice broke out. These responses mimic Voyeykov Ice Shelf
where over the course of several months the ice shelf lost stabilizing land fast ice, then mélange, followed by partial loss of
the ice shelf (Arthur et al., 2021). The immediate complete disaggregation of multiple floating ice tongues after the loss of the
fast ice suggests that the fast ice was not only preventing calving but also supplied sufficient backstress that essentially held
the tongues together. The glacier speeds began to gradually increase after the loss of their thick floating tongues (Fig. 9).

The calving regimes and dynamical changes of the Larsen B tributary glaciers are similar to their response after the 2002
Larsen B ice shelf disintegration, suggesting that calving is an immediate response to stress perturbations (Hulbe et al. 2008).
At first glance, the two events were quite different; for example, the tributary glaciers were stable prior to the 2002 event and

though they were readvancing and stabilizing prior to the 2022 event they were still in an imbalanced state (Seehaus et al., 2023), additionally the ice shelf was old and thick whereas the fast ice was much younger and an order of magnitude thinner. However, despite these differences, the similarities in the tributary glacier response to the two events are important to identify.

Crane Glacier experienced significant changes after the Larsen B Ice Shelf disintegration and fast ice break-out. In the three years following the Larsen B Ice Shelf disintegration event (2002 to 2005), the Crane Glacier ice front and grounding zone retreated 18 km into the fjord, and the ice front height increased from 60 m to just over 100 m (Scambos et al., 2011, De Rydt et al., 2015). Simultaneously, the glacier trunk upstream of the ice front lost elevation at a rate of 35 m yr$^{-1}$ (Shuman et al., 2011). As of March 2023, the 2022 event has caused Crane Glacier to retreat ~11 km in 14 months (Fig. 8). However, significant thinning in the upstream areas has yet to occur (Fig. 10). Between 2002 and 2003, Crane Glacier ice flow increased rapidly, roughly 3-fold from ~500 m yr$^{-1}$ to ~1500 m yr$^{-1}$ (Rignot et al., 2004). In 2007, Crane's terminus had speeds up to ~3500 m yr$^{-1}$ with a steady deceleration the following years (Wuite et al., 2015). By 2017 terminus speeds were ~1000 m yr$^{-1}$ (Rott et al., 2020) and remained that way until the break-out. In what is likely grounded ice (relative to both the grounding zone locations in this study and in Rott et al., 2018), speeds prior to the break-out were ~750 m yr$^{-1}$ and subsequently increased to ~1050 m yr$^{-1}$ by early 2023 (Fig. 9). Crane Glacier has responded similarly to the two episodes of buttressing loss in the last 20 years, although the magnitude of change was greater in the immediate aftermath of the 2002 event (Fig. 12).

Hektoria's calving in 2022 is similar to the 2002 event where initially floating tabular bergs calved and then an arcuate calving front formed with large rifts and slumping. In 2022/2023, Hektoria retreated ~25 km, with ~10 km of that retreat (or 1 to 5 km referencing the Rott et al.'s 2016 grounding line (Rott et al., 2018); Fig. 8) likely to be partially grounded ice on an ice plain (Tuckett et al., 2020). This is greater than the 2002 event in which Hektoria lost ~15 km of floating ice in the first year and it was not until a year after the loss of the ice shelf in 2003 that Hektoria began calving at its grounded terminus (Rack and Rott, 2004). This could possibly be explained with the lower Hektoria glacier being much closer to floatation in 2022 than it was in 2002 and possibly with a higher amount of accumulated damage. In that case, acceleration and thinning would have first been needed to bring the Hektoria Glacier to a height near floatation before significant retreat of grounded ice could occur after the 2022 event. In 2002 to 2003 Hektoria's ice flow speeds increased 8-fold from ~250 m yr$^{-1}$ to over ~2000 m yr$^{-1}$. Hektoria's terminus retreat paused from 2007 to 2009 and then reactivated in 2009 until the long-term fast ice formed in the embayment and stabilized the front in 2011 (Fig. 12). Following the fast ice breakout several kilometers upstream of the grounding line, Hektoria's ice flow speeds increased from 400 m yr$^{-1}$ to ~900 m yr$^{-1}$ (Fig. 9). Although the magnitude is not as great as the 2002 event, the glaciers have had a dramatic acceleration. Hektoria's thinning from 2002 to 2003 was between 5 to 38 m yr$^{-1}$ (Scambos et al., 2004), whereas the 2022 event resulted in thinning of between 8 to 11 m on Green Glacier from March 2022 to January 2023 (Fig. 11). Again, this is a similar and only slightly subdued response to the loss of embayment ice.

Both Crane and Hektoria experienced rapid changes after both the 2002 and 2022 events. The speed up of both glaciers was
immediate, yet gradual, as the evolution of the system adjusted to new geometry, particularly the glacier bed. This gradual,
slightly delayed increase in velocity may be why Sun et al. (2023) did not capture the acceleration as their velocity data ended
in July/August 2022 and the majority of the acceleration took place after that (Fig. 9). However, that is along the same timeline
of changes experienced in the 2002 event as the first velocity data was only available December 2002, nine months after the
ice shelf break-up (Wuite et al., 2015). Comparison of the speeds, thinning, and retreat rates, reveals that the 2002 event had a
greater impact on the glacier dynamics within the first year of the loss of ice shelf/multi-year fast ice buttressing. This is an
expected response, as the loss of the Larsen B Ice Shelf should result in a higher de-buttressing effect than the more recent loss
of the much thinner fast ice and thick glacier tongues.

## 6 Conclusions

The climate of the AP has been warming over the past several decades (Vaughan et al., 2003; Zagorodnov et al., 2012),
interrupted by a decade-scale cooling that coincided with the formation of the fast ice in 2011 (Turner et al., 2016). During the
2021/2022 summer, the Larsen B region of the AP experienced anomalously high temperatures, and strong westerly winds
contributed to an ice-free corridor to open along the eastern coast of the Peninsula that in turn allowed long-period high
amplitude ocean swell to reach the fast ice. In the 2021/2022 summer the Antarctic sea ice extent was at its lowest sea ice
extent in the satellite record (prior to 2023) with the Weddell Sea contributing 26% to that negative anomaly (Turner et al.,
2022). The sea ice extent in the Weddell Sea in 2022 was the 12[th] lowest in the satellite record (Turner et al., 2022) and the
pack ice area immediately near the Larsen B embayment was at its lowest since 2010.

The large-amplitude wave event with a long period swell that occurred 18 to 19 January reached the fast ice front via the pack
ice-free corridor. We infer that this flexed the fast ice, causing it to fracture and redistribute the stresses within the thin ice
plate, which would be seasonally at its weakest due to the recent warm air temperature. We note, however, that surface
meltwater-induced flexure and hydrofracture of the glacier tongues do not appear to play a direct role in this case. An AR and
foehn wind event occurred during and after the fast ice break-out, contributing to the quick removal of the fast ice from the
embayment.

All of the glacier responses following the Larsen B embayment fast ice break-out are reminiscent of the effects on glacier flow
and decreased surface elevation after the Larsen B ice shelf removal (i.e., extreme and varied; Rignot et al., 2004; Scambos et
al., 2004), despite the fast ice being substantially thinner than the ice shelf (5 to 10 m compared to ~250 m; Fig. 1) and the
glaciers being in different states prior to the break-outs (Seehaus et al., 2023). We conclude that the fast ice slab was acting to
significantly buttress the glaciers' floating tongues, and its removal led to the disaggregation of the tongues and destabilization
and dynamical changes of the grounded glaciers.

Antarctica's coastline is fringed with multi-year fast ice (Fraser et al., 2021) that is likely buttressing large glaciers around the continent. As the climate continues to change (Gilbert and Kittel, 2021), Antarctica's fast ice may become more susceptible to breakup due to increasing exposure to ocean swells via open-ocean corridors through pack ice (Reid and Massom, 2022; Teder et al., 2022). As Antarctic overall sea ice concentrations are projected to decrease over the current century (Holmes et al., 2022), this risk is inherently higher. Antarctic-wide fast-ice-buttressed glaciers will likely be subject to substantial dynamical changes and potential retreat if pack ice decline leads to multi-year fast ice break up.

This case study affirms the importance of examining the impacts of large-scale circulation patterns on foehn conditions, overall sea ice area, and ocean swells on the AP and other vulnerable ice shelf and outlet glacier areas. It is important to continue monitoring not only the glaciers feeding into the Larsen B embayment in terms of their response to changing fast ice conditions, but also other key glacier-ice shelf-fast ice interactive systems around Antarctica and their response to increased coastal exposure (Reid and Massom, 2022; Teder et al., 2022)

## Supplemental Information

Supplemental information can be found in tc-2023-88-supplement-version2.pdf.

## Acknowledgements

We would like to thank Chris Shuman and Mark Fahnestock for their help in monitoring the break-out and suggesting processing tools, as well as Bertie Miles for the suggestion on how to extract velocity profiles. We also thank the British Antarctic Survey pilots who captured images of the initial fast ice break-up on 31 January 2022, and the NSIDC for data access and technical support. NEO and TAS received support from NASA award 80NSSC22K0386 and USGS award 140G0118. AFB received support from the U.S. National Science Foundation (NSF) under award no. 1841607. GPS data was collected using the LARISSA award NSF OPP 0732602 and NSF OPP 0732921.

## Author Contributions:

NEO led the study, processed and analyzed the data, and led the writing of the manuscript. TAS initiated the idea and outlined the direction of the study with early data. GP processed the raw AMSR data to detect liquid water and AFB analyzed these data and produced Figure 5. RSA, SM, and LM contributed to the plan of the research. MLM analyzed the AR events. JAS contributed to the climate analysis methods. MT and ECP processed and analyzed the GPS data. All authors contributed to the writing of the manuscript and discussion of results.

**Competing interests:**
The authors declare no conflict of interest.
**Data Availability**
Sea ice concentration and extent data are available on the University of Bremen sea ice webpage (https://seaice.uni-
bremen.de/sea-ice-concentration/amsre-amsr2). Operation IceBridge data is available at NSIDC
(https://nsidc.org/data/icebridge) as well as the ICESat-2 data (https://nsidc.org/data/atl06/versions/6). Velocity data are
available from the Alaska Satellite Facility Sentinel-2 AutoRIFT pipeline (https://search.asf.alaska.edu). MODIS imagery can
be viewed and downloaded on the Worldview interface (https://worldview.earthdata.nasa.gov). ERA-5 data are available at
the Copernicus data store (https://cds.climate.copernicus.eu/cdsapp#!/home). WaveWatch III data are available on CSIRO
(https://data.csiro.au/collection/csiro:39819). The AMSR-E/2 data are available to download here: https://perscido.univ-
grenoble-alpes.fr/datasets/DS391. The Reference Elevation Model of Antarctica is available via the Polar Geospatial Center
(Howat et al., 2022, https://www.pgc.umn.edu/data/rema/). The Worldview DEMs are available from Polar Geospatial Center
upon request. The overview panel of Antarctica in Figure 1 is from Quantarctica (Matsuoka et al., 2021;
https://www.npolar.no/quantarctica/)

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
