# Peer review of "Triggers of the 2022 Larsen B multi-year landfast sea ice break-out"

_The Cryosphere, 2023_

## Referee Comment (RC1)

**Comments on "Triggers of the 2022 Larsen B multi-year landfast sea ice break-out and initial glacier response"**

by N.E. Ochwat et al.

This paper reports on the break-up of land-fast sea ice in the Larsen B embayment and the initial response of glaciers after the buttressing sea ice had drifted away. Furthermore, the authors studied various potential triggers leading to this process, including changes in atmospheric and circulation and sea ice cover during preceding and concurrent periods, checking numerical meteorological re-analysis data and changes in satellite-based sea ice concentration over an extended area. Patterns of fast ice break-up and the retreat of glacier fronts are documented, using optical satellite imagery. The analysis of glacier response is based on satellite observations, focusing on several Antarctic Peninsula outlet glaciers draining into the Larsen B embayment. Repeat observations of surface elevation on specific points and flow velocities along central flowlines are shown for a period spanning the break-up event.

By and large, this is a well written article, presenting interesting material on the land-fast sea ice break-up event, its potential causes and the response of glaciers. The work confirms the influence of land-fast sea ice on glacier flow dynamics, reflecting - with opposite sign - the slowdown, reduced downwasting and frontal advance of Larsen B glaciers during the previous period when fast ice was built-up. Still, there is need for major checks and improvement, as there are various issues lacking traceability or being inconsistent with previously published data.

*Main issues:*

This section includes a short summary on main issues to be clarified and improved. Details on the concerns and suggestions for improvements are put forth in the next section.

*Location of grounding zone*: The grounding zone positions are outdated. The 2016 grounding zone position of Hektoria Glacier was 12 km inland of the position shown in Fig. 9, and of Crane Glacier 5 km inland. This mismatch has major implications on various issues such as the interpretation of glacier flow dynamics, the estimation of the thickness of floating ice and the assessment of driving mechanisms for frontal retreat.

*Extent and thickness of floating sections of glacier tongues*: The floating section of the HGE terminus, extending between the glacier fronts of 2011 and of 21 November 2021 (shown in Fig. 8), covers more than 200 km$^2$ in area. In the manuscript ice thicknesses of the floating terminus up to some hundred metres are mentioned. This is not an agreement with mass continuity, as it would exceed the ice volume delivered across the flux gate close the 2011 ice front.

*Processes of fast ice break-up and drift:* The discussion on these processes refers to changes in the backstress of the fast sea ice as main factor, but estimates on the stress magnitude are not provided. It is questionable if changes in the backstress due to depletion of fast ice in a widening bay (as in front of HGE glacier) can be the main trigger or if other factors play an important role as well.

*Processes of glacier flow acceleration and frontal retreat:* Similarity of the glacier response in the wake of fast ice removal to the response after the 2002 ice shelf disintegration is claimed. However, the prerequisites and course of the events are quite different. The 2022/2023 case refers to recently formed, thin floating glacier tongues of structural weakness, in contrast to the compact, grounded ice bodies of the pre-2002 period. The patterns of frontal retreat and the ice fracture processes are also rather different.

*Details:*

L53-L56: It would be of interest mentioning in this context also the 10-fold ice speed increase in 2009 at the Hektoria Glacier front compared to the years before 2002 (Wuite et al., 2015) and the declining mass losses during the period when fast ice was present. The total HGE losses of grounded ice amounted to 4.26 Gt/yr in 2011 to 2013 and 1.75 Gt/yr in 2013 to 2016 (Rott et al., 2018).

L59-L60: Taking into account the ice volume discharged from grounded ice to floating sections of the glacier termini and assuming mass continuity, an ice thickness of hundreds of metres can refer only to a very small portion of the floating glacier tongue at its full extent. Rott et al. (2018) derived the mass fluxes across gates close to the 2011 glacier fronts of Larsen B glaciers for the period 2011 to 2016. Based on this number and accounting for further slowdown after 2016, the estimate of the total ice volume supplied to the newly formed section of the HGE tongue amounts to 39 km$^3$ between 2011 and 2021.

L93-L98; Fig 1: The scale of the figures is too small for providing clear information on features of interest in the glacier bays and Larsen B embayment (e.g. structural properties, deformation patterns, rifts, glacier fronts). The displayed ICESat freeboard map shows blocks and streaks, with discontinuities along straight lines. Furthermore, the grounding zone positions (though not well traceable at that scale) seem to be outdated (see comment on L343ff).

L108-109: Relative humidity less than 79% is not a suitable criterion for determining surface melt conditions. If the vapor pressure on the snow surface is higher than in the atmosphere sublimation may take place also at air temperatures above 0°C, depending on the net incoming energy.

L167-L171: Please provide information on the procedure for deriving flow velocities (matching window size, sampling steps, cross-correlation threshold) and on the uncertainty of the velocity products for the different sensors and time spans.

L215- L216: Considering the ice export across the 2011 glacier front, ice with thickness on the order of 300 m can stretch out only over rather small areas. Please provide details on the ice thickness information (dates, extent of area with freeboard > 40 m). Due to the coarse resolution the Fig.1 is not a suitable source for this information.

L217-L218: The maps of surface elevation change (SEC) 2013 to 2016 by Rott et al. (2018) show increase in surface elevation on the lower 10 km of Crane Glacier and frontal advance, coinciding with major surface lowering in the upper reaches of the main glacier trunk. This contradicts the statement on dominating ice input from tributary glaciers in the paper of Needell and Holschuh (2022) who do not cite the observations of Rott et al.

L218-L219: Referring to the glacier fronts on 25 June 2011 (TanDEM-X image, shown in Fig. 2 in Supplement of Rott et al., 2020) and on 21 November 2021 (shown in Fig. 8 of Ochwat et al.) the Hektoria Glacier front advanced during this period by about 20 km and the new floating area covered more than 200 km$^2$.

L232: Fig. 1b and 1c show MODIS images of 16 and 21 January 2002, not 18, 19 and 20 January.

L261-L264: Please explain the decision rule for identifying and for shutting on and off foehn events. According to Fig. S1 a threshold of 3 m/sec is used for Foehn detection. This number corresponds to a light breeze, not suitable for facilitating the breakthrough of the flow at the leeside of a mountain range.

L274, Fig. 4: The label of Fig 4a abscissa is date (not time).

L295: The surfaces may have been frozen on the date of the Landsat image (16 January 2022), but the temperature record (Fig. S2) shows several days of high temperature in January 2022, an indication for with surface melt that is also apparent in SAR images.

L343ff, Section 4.3.1: In the context of various issues addressed in this section, an outdated version of the grounding zone positions is used. Estimates for grounding line locations of Larsen A and Larsen B glaciers in 2013 and 2016 were obtained by Rott et al. (2018), based on changes in surface elevation of TanDEM-X data. The transition from grounded to floating ice is associated with a strong drop in surface elevation change (dh/dt). The 2016 grounding lines of Hektoria Glacier and Crane Glacier (see zoom images in Rott et al., 2020) are about 12 km (Hektoria) and 5 km (Crane) inland of the position

shown in Fig. 9 of Ochwat et al. The 2016 Crane location refers to the centre of the canyon. Further retreat during recent years has to be expected.

L346-L350: Please provide evidence on the iceberg properties and calving features described, as well as on the dates of the images to which the described features refer. Checking the pre-frontal embayment area in the Sentinel-1 time series of 2022 and 2023, there are no features that can be uniquely allocated to a toppled iceberg. In the SAR images of January and February 2022 and December 2023 to February 2023 icebergs in the pre-frontal areas show low reflectivity, evidence for the presence of melting firn. In subsequent colder periods the firn freezes, causing a transition to high backscatter intensity.

L373: An extended thick (>300 m) floating tongue in front of HGE glaciers does not match the mass continuity for input across the 2011 glacier front (see related comments above).

L389-L428, Section 4.3.2: Also for this section an update on the grounding zone positions is needed (see comment L343ff). The main parts of the velocity profiles shown in Fig. 9 are located either on floating ice and or on pre-frontal ice mélange. This questions the argumentation of issues referring to the grounding zone and to velocities on grounded ice.

L420, Figure 9: Please mark the updated location of the grounding line and the location of the ice front on different dates. The profiles of Hektoria and Green glacier deviate from the course of the central flowline. The Landsat-based Hektoria Glacier velocities of Nov. 2022 to Jan. 2023 deviate significantly from the Sentinel-1 based velocities of June to Oct. 2022. How reliable are the Landsat velocities?

L446: The statement "Hektoria Glacier lacks long-term elevation change data points" is not valid, at least for TanDEM-X data 2011 to 2016 (Rott et al., 2018), but also regarding further data from this mission and from other satellite sensors (e.g. CryoSat).

L523-L525: Quantitative estimates on the fast ice backstress magnitude and its spatial pattern are needed for affirming this conclusion, as well as considerations regarding possible changes of other driving factors, such as winds and ocean currents.

L531-L554: The argumentation, referring to proposed governing processes for glacier response and claiming similarity to the 2002 break-up event, are not well founded. There are various clues indicating major differences regarding the calving regimes and dynamic response of the Larsen B tributary glaciers in 2002 versus 2022/2023. Up to the 2002 event the glaciers were close to a balanced state, the tongues were several hundred metres thick and grounded. In 2022 the newly formed sections of the glacier tongues were composed of comparatively thin floating ice. Furthermore, there is evidence of structural weakness. On the floating sections high resolution elevation data show rugged surface structure and wave-like surface features of different wavelengths. There are also indications for major strain-rate weakening, as velocity data show. For example, on the central flowline of Hektoria Glacier a threefold increase of velocity between the grounding line and the glacier front is evident in 2017 data, and on Crane Glacier a twofold increase (Rott et al., 2020).

L537-L538: On Crane Glacier close to the calving front a velocity of 9.6 m/day was observed in June 2007 data (Wuite et al., 2015), 2.3 times the 2003 value cited in L538.

L542: Based on the updated grounding line position, the number for retreat of grounded ice should be corrected.

L543-L544: The floating ice of 15 km length in front of Hektoria Glacier was in fact a remnant section of the Larsen B ice shelf that remained in the pro-glacial bay for several month after disintegration of the main ice shelf. The boundary of the ice shelf is clearly evident in the tidal deformation pattern of ERS-tandem (1-day repeat) interferometric data. The retreat of grounded ice started in March 2003. The changes between February 2002 and April 2003 are documented by Rack et al. (2004) by means of several SAR images.

L555ff, Conclusions: Major revisions are needed, taking into account the comments above.

*References cited in the manuscript, but not included in the list:*

Crawford et al., 2022 (cited in L159)

Hersbach et al., 2020 (cited in L104)

Kwon et al., 2020; (cited in L473)

Ochwat et al., 2022 (cited in L159)

Robel et al., 2017 (cited in L527)

Smith et al., 2020 (cited in L38 and L 501)

**References**

Needell, C., and Holschuh, N.: Evaluating the retreat, arrest, and regrowth of Crane Glacier against marine ice cliff process models. Geophys. Res. Lett., 50, e2022GL102400. https://doi.org/10.1029/2022GL102400, 2023.

Rack, W., and Rott, H.: Pattern of retreat and disintegration of Larsen B ice shelf, Antarctic Peninsula, Ann. Glaciol., 39, 505-510, 2004.

Rott, H. Abdel Jaber, W., Wuite, J., Scheiblauer, S., Floricioiu, D., van Wessem, J.M., Nagler, T., Miranda, N., and van den Broeke, M.R.: Changing pattern of ice flow and mass balance for glaciers discharging into the Larsen A and B embayments, Antarctic Peninsula, 2011 to 2016, Cryosphere, 12, 1273–1291, https://doi.org/10.5194/tc-12-1273-2018, 2018.

Rott, H., Waite, J., De Rydt, J., Gudmundsson, G.H., Floricioiu, D., and Rack, W.; Impact of marine processes on flow dynamics of northern Antarctic Peninsula outlet glaciers, Nature Communications, 11:2969,| https://doi.org/10.1038/s41467-020-16658-y, 2020.

Wuite, J., Rott, H., Hetzenecker, M., Floricioiu, D., De Rydt, J., Gudmundsson, G. H., Nagler, T., and Kern, M.: Evolution of surface velocities and ice discharge of Larsen B outlet glaciers from 1995 to 2013, Cryosphere, 9, 957–969, https://doi.org/10.5194/tc-9-957-2015, 2015.

---

## Referee Comment (RC2)

**Comments on "Triggers of the 2022 Larsen B multi-year landfast sea ice break-out and initial glacier response"** by N.E. Ochwat et al., tc-2023-88

This paper presents an important case-study analysis of the region of perennial fast ice that formed in the embayment previously occupied by the former Larsen B Ice Shelf following its disintegration in 2002. Specifically it uses a combination of observational, reanalysis and remote-sensing data to examine the causes ("triggers") of a major fast-ice break-out event in 2002, and shows the effects of the latter on the speed, elevation and calving behaviour of various outlet glaciers feeding into the embayment. In so doing, it confirms the findings of a number of recent studies that highlight the important role of fast ice in mechanically buttressing adjacent glaciers and ice shelves, and the linkage between fast ice loss and changes glacier/ice shelf dynamics and calving behaviour - with this effect being modulated by surrounding pack ice that damps the impact of waves in breaking up the fast ice. An additional important finding of this study is that substantial decreases in glacier elevation occurred in response to loss of the fast ice buttress, in concert with major floe speed increases – in much the same way as occurred following the Larsen B disintegration.

In summary, this paper makes a valuable contribution to a growing corpus of work that highlights previously overlooked and poorly-quantified though crucially-important linkages between sea ice (change) – both in the form of stationary coastal fast ice and moving pack ice – and (change in) glacier and ice-shelf dynamics. This is particularly timely, given the current state of Antarctic sea ice and the increasing concern over Antarctica's contribution to sea-level rise. Having said this, there are a number of issues that I feel need to be addressed in order to improve the paper.

The science questions addressed are well within the scope of TC, and I recommend publication subject to substantial revisions, as laid out below.

Please find below my overall comments, followed by a more detailed listing of suggestions. I hope these are useful and help improve the paper.

**Overview Comments**

1. The paper itself is generally well written, apart from minor issues relating to inconsistent use of tense and minor grammatical errors. However, the text seems overly long, and could benefit from being substantially shorter and more concise. This would make it more readable while telling the story more clearly – leading to greater impact in this journal.

2. The terms "sea ice" and "fast ice" are used interchangeably. Explain the difference between fast ice and pack ice upfront in the introduction, then refer specifically to fast ice and pack ice as appropriate (or overall sea ice – which is what the passive microwave dataset gives). See my Comment against Line 41 below (in Specific Comments).

3. May I suggest that the Data and Methods Section be shortened and restructured around the variables and phenomena being investigated – with these being grouped accordingly – rather than listing (working through) the individual datasets themselves. This would also help focus the paper more fully on the story being presented, which is certainly a good one, while also minimising repetition and improving the "readability" of the paper. For example, an introductory sentence or two/three (preface) could be added immediately after the Section 3 heading (between lines 99 and 100), along the lines of – "The linkages between fast ice……………… and glacier events were detected and analysed using a combination of observational, reanalysis and remote-sensing data. (NB then briefly adding the high-level information about the individual datasets i.e., what they are/names, where they are obtained from)." Then, subsequent sub-sections could consolidate information currently scattered across the sub-sections by focussing on the different sub-topics - in a more logical sequence than is currently the case, and with

emphasis on the techniques used.  Section 3.1. could/would then become "Sea Ice Change and Variability" – pulling together information relating to how the fast ice breakout event was detected and monitored; detection of open-ocean corridors in the adjacent pack ice; and fast ice surface melt was determined and mapped. Then Section 3.2 could be "Glaciological Characteristics"; and Section 3 "Atmospheric and Oceanic Factors".

4. The Results (Section 4) is very long and contains detailed information about the timings of the different events (corridor formation, fast ice breakup, calving, glacier acceleration, elevation change etc. for each glacier) that is difficult to follow.  This section could be shortened substantially by (1) condensing the results into a Table (or two), and by (2) adding a timeline schematic along the lines of Figure 11 in https://agupubs.onlinelibrary.wiley.com/doi/full/10.1002/2014JF003223.  This would again substantially improve the readability of the findings and increase their impact.  It's hard to follow the different timings and events – and how they line up – in the current text.

5. The Discussion (Section 5) is also very long in the way that it works through all of the topics in sequence.  This could be substantially shortened – and repetition of Results avoided – by focussing on synthesising the main findings into a coherent story around why the fast ice breakout did not occur until 2002, and what the effects were then on the glacier systems – referring back to the suggested timeline schematic figure and associated Tables.

6. The claim in Lines 315-316 and elsewhere (e.g., Lines 495-496, 503-504) that in January 2022, a relatively ice-free corridor connected the fast ice front area to the open Southern Ocean for the first time since persistent fast ice formed in 2011 needs backing up with further evidence.  This assertion is based on Figure 6b, which gives sea ice area in an offshore box for January 19 only for the years 2010 to 2022 (I'm sorry but I don't have access to Figure S5 which is also referred to).  I did a quick search through past satellite sea-ice concentration images and also found the persistent occurrence of a corridor in February 2021 (for example).  This begs the question – why did the fast ice breakout only occur in January 2022 and not earlier?  It also suggests that the fast ice breakout in 2022 may be due to a combination of factors, and not only exposure to ocean swells (as stated in Lines 27-29 of the Abstract and elsewhere).

7. Section 5.3 should specifically refer back to, and compare the new findings with, other studies from elsewhere around Antarctica by Miles et al. (2018), Arthur et al. (2021), Greene et al. (2018) and Gomez-Fell et al. (2022) regarding relationships between fast ice presence/absence and both (1) glacier calving and (2) speedup (i.e., buttressing).

8.  Re the Figures – may I recommend marking key phenomena/events referred to in the text directly on the figures (e.g., X, Y or the like), such that pointers can then be given in the text e.g., "….this event is marked X in Fig. XXa".  This will greatly help the reader.

9.  General comment re the Figures – the colours are challenging to differentiate (at least for me) – e.g., Figures 8 and 9.  Also, Figure 9 is too complicated – too many lines.  This could be substantially simplified by reducing the number of lines (while including the results in a table).  The satellite image in Figure 10 is indistinct and difficult to interpret – this could be improved by adding boundaries and marking key features.

10. The paper is generally well referenced, but I've made suggestions regarding adding a few key references that are missing.  Also, the order of referencing is neither chronological nor alphabetical e.g., Lines 527-528

Just one other thing – I note that the authors (or rather the lead author and 2 others) have also submitted a shorter version of this topic for publication as a "sidebar" in the annual State of the Climate Report for 2022 (in press in the Bulletin of the American Met Soc) – and with a similar title. It may be best if the authors refer to this other publication upfront in this new paper.  May I suggest that this information be added in a sentence at the very end of the Introduction (onto Line 72) –

stating that a shortened version is in press in BAMS (and referencing that). However, this will also need to briefly state how this paper differs from that sidebar i.e., why this new paper is necessary.

**Specific Comments and Suggestions**

Line36 – 2008 and 2009 (add Braun, M., Humbert, A. & Moll, A. Changes of Wilkins Ice Shelf over the past 15 years and inferences on its stability. *Cryosphere* **3**, 41–56, 2009)

Line 41 – "….and outer-margin calving due to ocean swell-induced flexure (Massom et al., 2018). Massom et al. (2018) further implicate loss of attached landfast sea ice (fast ice) in the Wilkins Ice Shelf breakup events, following loss of a protective pack ice buffer offshore – due to the vulnerability of fast ice to ocean swells (Crocker and Wadhams, 1989). While fast ice is consolidated sea ice that remains stationary attached to the coast (Fraser et al., 2021), pack ice refers to sea ice that is constantly in motion under the influence of winds and ocean currents."  REFERENCE - G.B. Crocker, P. Wadhams, Breakup of Antarctic fast ice, Cold Regions Science and Technology, 17 (1), 61-76, https://doi.org/10.1016/S0165-232X(89)80016-3, 1989.

Line 45 – what is meant by "increase ocean swell"?

Line 50 – replace "catastrophically" with "substantially".

Lines 56-58 ("Fast ice……") – remove.

Lines 58-60 – inappropriate to have this Result in the Introduction – move this to the appropriate place.  It's also not clear how these thicknesses were derived.  ALSO – line 60 – "containing both fast ice and glacial ice".

Lines 65-68 – this should also refer to other studies relating fast ice to glacier calving and advance/speed e.g., Miles et al. (2018), Arthur et al. (2021), Greene et al. (2018) and Gomez-Fell et al. (2022).

There is a need to introduce the concept of the damping of waves by pack ice, with references. This is central to the ocean corridor concept proposed by Massom et al. (2018).

Lines 68-72 – define buttressing.

Line 67 – add Massom et al. (2010) after "collapse".

Line 76 – "south".

Line 78-79 – To the east, the northwestern Weddell Sea is generally covered by pack ice.

Line 83 – NB there's more to the Larsen B breakup than this (hydrofracture) alone – refer back to Lines 37-41, and my Comment on Line 41 above.  RE THIS, Lines 80-84 could probably be merged into Lines 33-36.

Line 102-103 – remove "a climate….ECMWF)".

In Section 3 and in the appropriate place, add – "Following Massom et al. (2018) and Teder et al. (2022), we investigate the occurrence of open-ocean corridors across the sea ice zone, enabling ocean swells to interact in an unobstructed fashion with the Larsen B embayment fast ice".

Lines 139-141 – not clear what this means.

Line 147 "several…images"

Line 152 – image cross-correlation

Line 161 – images

Line 162 "estimated from the location of a break in slope"

Line 164 – what is listric faulting?

Lines 179-180 – different tenses. Be consistent throughout the paper.

Line 181-182 – what is meant by "Assuming snow is negligible"? Also, what is this assumption based on?

Line 183 – why were these density values chosen (based upon what)?

Lines 189-194 – Did AMIGOS also provide meteorological information?

Line 202-210 – how was fast ice area determined. Were there any difficulties in distinguishing the boundaries?

Line 205 unclear – how does an edge reform?

Line 209 – not clear what "the edge broke out" means.

Lines 2002-210 – this needs a figure to show the sequence of events discussed, as a series of outer margin lines.

Line 212 – "occupation of"

Line 218 – "reformed into"

Line 219 – "advanced 16 km from February 2011 to XXXX"

Line 220 – "while Punchbowl"

Lines 223-224 – why is there a seasonal cycle in the Scar Inlet Ice Shelf flow speed? And is this a feature of all of the glaciers investigated? Please add this information.

Line 232 – it's hard to see the fractures in Figure 1b and 1b – the images are very small. Also – it's not clear what Figures 1d-f show – maybe consider leaving these out.

Line 238 – not clear what re-enter means here - is this floes from outside moving into, or the formation of new ice within?

Line 238 – "sea ice coverage"

Line 239 – "winter 2022"

Line 240 – what is meant by "apparent coherency"?

Lines 242-243 – this sentence needs rewriting. Also, change plates to floes. Regarding "sea ice concentration varied" – over what area, and does this refer to pack ice or fast ice (noting that a feature of fast ice is its consolidated nature i.e., 100% concentration)?

Lines 245 and 246 – should "climate" be "meteorological" here?

Line 245 onwards – need to refer to Crocker and Wadhams (1989) and Langhorne et al. (2001) here, regarding the fact that fast ice is particularly vulnerable to breakup by ocean waves. REFS: Langhorne, P., Squire, V., Fox, C., and Haskell, T. (2001). Lifetime estimation for a land-fast ice sheet subjected to ocean swell. *Annals of Glaciology, 33*, 333-338. doi:10.3189/172756401781818419

G.B. Crocker, and P. Wadhams (1989). Breakup of Antarctic fast ice. Cold Regions Science and Technology, 17(1), 61-76, https://doi.org/10.1016/S0165-232X(89)80016-3.

Figure 3 and Line 261 – make the blue box more prominent. Also, why was this location chosen, and why is 4 grid cells the size chosen?

ALSO – is "surface temperature" surface air temperature?

Line 262-264 – unclear. Occurring when?

Lines 271-273 – not clear as written.

Lines 292-295 – ungrammatical – rewrite as 2 sentences.

Figure 5 – make the solid and dashed lines thicker.

Line 307 etc. – is sea ice extent based on the 15% ice concentration threshold? (add this information to the appropriate Data and Methods sub-section).

Line 307 – why is January 19 chosen? This is the date of initial fast ice breakout, but what were sea-ice conditions like in the previous and subsequent days?

Figure 6b – the text above and y axis state "sea ice area", but the text talks about sea ice extent only. In Line 308 – should "time series of sea ice extent (concentration multiplied by area of pixel)" be "time series of sea ice area (concentration in each pixel multiplied by the number of ice-covered pixels)"?

Figure 6b – also, why is the value given only for January 19 in all of the years? This could be misleading to interpretation of when and how long open-ocean corridors occurred. Also in lines 315-316 – Figure 6b does not back up the statement that no other corridors occurred over the period from 2011, as it shows January 19 only. For example, I had a quick look at the satellite data and this shows the persistent occurrence of a corridor in February 2021 (for example). This leads to the question – why did the fast ice breakout only occur in January 2022 and not earlier? Therefore, the claim in Lines 315-316 that "This pathway, which allows for wave action to access the front of the Larsen B fast ice, had not been present since the fast ice's formation in 2011" needs backing up with further evidence. This comment also applies to Lines 495-496 – "Therefore, for the first time since the formation of the persistent fast ice cover in 2011, a relatively ice-free corridor connected the fast ice front area to the open Southern Ocean." (I'm sorry but I don't have access to Figure S5 which is also referred to). Also Lines 503-504 etc.

Line 325 – open-ocean (sea ice-free) corridor

Line 329 – equivalent to a wavelength

Line 343 – change "4.3.1 Initial retreats of landfast ice and glacier fronts" to "4.3.1 Retreat of glacier fronts"

Line 347 evidenced by

Line 356-358 – what does this mean, and why is it important?

Line 372 – what is meant by "Hektoria and Green Glacier responded to the collapse in later months following the fast ice break-out"? What collapse?

Lines 374-376 – unclear as written.

Figure 8a and 8c – length scales are missing.

Section 5 Discussion. This section seems overly long, and may repeats much of what has been stated before. Much of this information could be captured more concisely in a well-formulated schematic along with Tables – see my General Comment 6 above.

Lines 390-391 and Section 4.3.2 – Did Evans Glacier also show a speed change?

Line 396 – what is meant by noise levels in the data? What are they?

Figure 10 – the satellite images are indistinct and difficult to distinguish. Please mark of features and important boundaries.

Line 462 onwards - As stated above in General Comments, the Discussion (Section 5) is also very long in the way that it works through all of the topics in sequence. This could be substantially shortened – and repetition of Results avoided/minimised – by synthesising the main findings into a coherent story around why the fast ice breakout did not occur until 2002, and what the effects were then on

the glacier systems. This would then naturally refer back to the suggested new timeline schematic figure and associated Tables.

Line 463 – "Synoptic scale climate patterns" may be confusing, as synoptic is a meteorological term referring to the approximate horizontal scale of cyclones.  Maybe replace with "Meteorological conditions. This comments also applies to other places where "climate" is used e.g., Line 464.

Line 477 – "eastern (lee) side"

Line 480 – should low concentration be zero concentration?

Line 496 - damping

Lines 498 and 507-509 – need to add the references to Langhorne et al. (2001) and Crocker and Wadhams (1989) here.

Lines 507-509 – not clear whether this is referring to fast ice or glacier ice.

Line 511 – 'the broken-out fast ice had drifted 9-16 km"

Lines 515-517 – again, it is not clear whether this is referring to fast ice or glacier ice, or both. Hydrofracturing a process associated with crevasses on ice shelves/glacier, and has yet to be observed on fast ice.

Lines 517-519 – This is similar to the finding of Massom et al. (2018). They found that strong and persistent offshore westerly/northwesterly winds in late 2001 through early 2002 both (1) created a persistent sea-ice free corridor offshore from the Larsen B Ice Shelf to enable swell penetration that contributed to the ice-shelf breakup, then (2) blew the resultant icebergs and melange out of the Larsen B embayment.  Please refer to this parallel here.

Line 520 onwards (Section 5.3) – please specifically refer back to, and compare the new findings with, other studies from elsewhere around Antarctica by Miles et al. (2018), Arthur et al. (2021), Greene et al. (2018) and Gomez-Fell et al. (2022) regarding relationships between fast ice presence/absence and both (1) glacier calving and (2) speedup (i.e., buttressing).

Line 524 – replace "despite" with "contrary to".

Lines 525-529 – I didn't quite understand these 2 sentences, and how these factors relate to the findings of this paper.

Line 569 – effects on glacier flow and decreased surface elevation.  Also reference Rignot et al. (2004) and Scambos et al. (2004) here.

Line 560 – that the sea ice concentration in the Weddell Sea in 2022 was the lowest recorded is somewhat ambiguous.  When did this occur (in the year)?  Also, does this refer to the entire Weddell Sea?

ALSO – there was a large sea-ice free corridor prior to and during the Larsen B disintegration event in 2002 (see Massom et al., 2018)

Line 555 onwards (Conclusions) – again, please place the current findings more in the context of previous studies.

Lines 557-559 – where is it shown that high temperatures (alone) caused the ice-free corridor?  Is it more likely to be wind-driven?

Lines 559-561 – this may not be the case – refer to Massom et al. (2018) regarding the extraordinary opening in late 2001 through early 2002.  Also see my comments above and in the Overall Comments.

Lines 563-564 "pack ice-free corridor"

Lines 564-565 – fast ice flexure would not be confined to the outer margins – see Langhorne and Crocker and Wadhams papers.

Line 565 – hydrofracture is not a process that has been associated with fast ice.

Line 571 – replace "The fast ice was clearly buttressing…" to "This suggests that the fast ice slab was acting to buttress…"

Line 571-572 – this is a place to reference previous studies i.e., "confirming the findings of previous studies e.g., Massom et al. (2018), Miles et al. (2018), Arthur et al. (2021), Greene et al. (2018) and Gomez-Fell et al. (2022).

Lines 576-586 – Suggest combining the 2 paragraphs into one coherent paragraph.

Line 576 – move the Fraser reference to "…fringed with multi-year fast ice (Fraser et al., 2021)…"

Line 577-578 – this is not a new trigger mechanism. Suggest changing to "Antarctica's coastal fast ice may become more susceptible to breakup due to increasing exposure to ocean swells via open-ocean corridors through pack ice (Reid and Massom, 2022; Teder et al., 2022)."

REF: Reid, P.A., and R.A. Massom. 2022. Change and variability in Antarctic coastal exposure, 1979–2020. *Nature Communications,* **13,** 1164, https://doi.org/10.1038/s41467-022-28676-z

Line 580 – change "are" to "will likely be". Also, what is meant by "similar to ice shelf tributary glaciers"? – suggest removing this.

Lines 584-586 – change to: "It is important to continue monitoring not only the glaciers feeding into the Larsen B embayment in terms of their response to changing fast-ice conditions, but also other key glacier-/ice shelf-fast ice interactive systems around Antarctica and their response to increased coastal exposure (Massom et al., 2022; Teder et al., 2022)". I added these references as this is what they propose.

Line 602 Data Availability – change "data is" to "data are", and in line 607 add "data".

---

## Community Comment (CC1)

**Short comments on tc-2023-88 "Triggers of the 2022 Larsen B multi-year landfast sea ice break-out and initial glacier response" by Naomi Ochwat et al.**

This is an interesting manuscript which examines the leading drivers and glaciological implications of the recent fast ice break out at Larsen B Embayment, and one which provides further evidence of the importance of sea ice (pack and fast) for the buttressing and fortification of Antarctic glaciers from damaging ocean conditions (namely swell wave-induced damage, debuttressing and subsequent calving). In keeping with the comments of Helmut Rott (Reviewer 1) and the other, anonymous reviewer (Reviewer 2), the manuscript is well written overall, timely with respect to the growing realisation of the importance of sea ice–ice sheet interactions, and for this reason we believe the manuscript's publication will be of interest to the readership of *The Cryosphere*.

Below, we present two short comments motivated by those raised by the reviewers, which we hope the authors will consider when revising their manuscript.

**Comment 1**

To add to Reviewer 2's comments pertaining to the need to "refer back to, and compare the new findings with, other studies from elsewhere around Antarctica" (see their comments #7, #10 and those re: Lines 65-68, 67,517-519 and 520 onwards, especially), the authors should also consider including in their revised manuscript reference to/discussion of the following recently published papers, which we were somewhat surprised to see omitted in the initial version of the text given their direct relevance to the study:

- Francis, D., Mattingly, K. S., Lhermitte, S., Temimi, M., and Heil, P.: Atmospheric extremes caused high oceanward sea surface slope triggering the biggest calving event in more than 50 years at the Amery Ice Shelf, *The Cryosphere*, 15, 2147–2165, https://doi.org/10.5194/tc-15-2147-2021, 2021.

- Christie, F.D.W., Benham, T.J., Batchelor, C.L., Rack, W., Montelli, A., and Dowdeswell, J.A.: Antarctic ice-shelf advance driven by anomalous atmospheric and sea-ice circulation. *Nature Geoscience*, 15, 356–362, https://doi.org/10.1038/s41561-022-00938-x, 2022.

- Francis, D., Fonseca, R., Mattingly, K. S., Marsh, O. J., Lhermitte, S., and Cherif, C.: Atmospheric triggers of the Brunt Ice Shelf calving in February 2021. *Journal of Geophysical Research: Atmospheres*, 127, e2021JD036424. https://doi.org/10.1029/2021JD036424, 2022.

With particular regards to the eastern Antarctic Peninsula (EAP)-wide study of Christie et al. (2022), this paper provides important regional and historical context for Larsen B's 2022 behaviour in terms of the exact types of sea ice–ocean–ice sheet interactions reported here, which we believe serve to complement the findings/interpretation of the present study well. With reference to e.g. Section 5.1 (Line 463+), Christie et al. (2022) similarly attribute the relative period of glacial advance/stability and increased sea ice presence observed along the EAP over the last ~2 decades to the (multi-)decadal influence of the IPO and SAM.

On the importance of ice mélange for ice-shelf/glacier stability (cf. e.g. Line 525), the authors may also be interested in the following recent paper pertaining to the breakaway of iceberg A-68 from the neighbouring Larsen C Ice Shelf:

- Larour, E., Rignot, E., Poinelli, M., and Scheuchl, B.: Physical processes controlling the rifting of Larsen C Ice Shelf, Antarctica, prior to the calving of iceberg A68. PNAS, 118 (40), e2105080118, https://doi.org/10.1073/pnas.2105080118, 2021.

**Comment 2**

We agree with Reviewer 1 that it is essential that the correct grounding line position be used when analysing the velocity acceleration patterns of the various glaciers.

With the above in mind, and as the authors themselves allude to in the paper (Lines 67, 525), the glacier acceleration trends observed following fast ice breakout (in early 2023, especially) appear to contradict the observations of another recent paper by Sun et al. (note: now published at: https://agupubs.onlinelibrary.wiley.com/doi/epdf/10.1029/2023GL104066).

On the assumption that the acceleration trends reported in the present study still hold after the grounding line is updated and velocity uncertainties are quantified (cf. Reviewer 1's comments), then some additional words reconciling (or not …) the findings of the two studies would be useful to include beyond that already stated on Line 525. For example, could the differences in velocity signal be due to differing processing and/or smoothing techniques used? (probably unlikely unless errors in this study are high). Or perhaps due to the fact that for such a confined embayment, the glacier response was simply lagged until 2023 and hence not captured fully by Sun et al.'s analysis (whose observations ended late 2022)? Whatever the case, both studies ultimately motivate future research into the oft-overlooked nature of sea ice-ice sheet interactions at Larsen B and beyond.

Finally, on the subject of grounding lines, for ease of reference please also show the updated locations on each of the insets shown in Figs 1, 8, 9, 10 and 11.

--

END

**Frazer Christie (fc475@cam.ac.uk), Christine Batchelor, Wolfgang Rack & Julian Dowdeswell**

---

## Author Comment (AC1)

Dear Dr. Frazer Christie et al.,
We appreciate your insightful comments and for your constructive and helpful review of our manuscript. We found the additional references to be valuable in reconstructing our discussion and framing of the results.

Below, you will find our responses to your comments in blue. Thank you for your review.
Naomi Ochwat, on the behalf of the coauthors

**Short comments on tc-2023-88 "Triggers of the 2022 Larsen B multi-year landfast sea ice break-out and initial glacier response" by Naomi Ochwat et al.**

This is an interesting manuscript which examines the leading drivers and glaciological implications of the recent fast ice break out at Larsen B Embayment, and one which provides further evidence of the importance of sea ice (pack and fast) for the buttressing and fortification of Antarctic glaciers from damaging ocean conditions (namely swell wave-induced damage, debuttressing and subsequent calving). In keeping with the comments of Helmut Rott (Reviewer 1) and the other, anonymous reviewer (Reviewer 2), the manuscript is well written overall, timely with respect to the growing realisation of the importance of sea ice–ice sheet interactions, and for this reason we believe the manuscript's publication will be of interest to the readership of The Cryosphere.

Below, we present two short comments motivated by those raised by the reviewers, which we hope the authors will consider when revising their manuscript.

**Comment 1**
To add to Reviewer 2's comments pertaining to the need to "refer back to, and compare the new findings with, other studies from elsewhere around Antarctica" (see their comments #7, #10 and those re: Lines 65-68, 67,517-519 and 520 onwards, especially), the authors should also consider including in their revised manuscript reference to/discussion of the following recently published papers, which we were somewhat surprised to see omitted in the initial version of the text given their direct relevance to the study:

• Francis, D., Mattingly, K. S., Lhermitte, S., Temimi, M., and Heil, P.: Atmospheric extremes caused high oceanward sea surface slope triggering the biggest calving event in more than 50 years at the Amery Ice Shelf, The Cryosphere, 15, 2147–2165, https://doi.org/10.5194/tc-15-2147-2021, 2021.

• Christie, F.D.W., Benham, T.J., Batchelor, C.L., Rack, W., Montelli, A., and Dowdeswell, J.A.: Antarctic ice-shelf advance driven by anomalous atmospheric and sea-ice circulation. Nature Geoscience, 15, 356–362, https://doi.org/10.1038/s41561-022-00938-x, 2022.

• Francis, D., Fonseca, R., Mattingly, K. S., Marsh, O. J., Lhermitte, S., and Cherif, C.: Atmospheric triggers of the Brunt Ice Shelf calving in February 2021. Journal of Geophysical Research: Atmospheres, 127, e2021JD036424. https://doi.org/10.1029/2021JD036424, 2022.

With particular regards to the eastern Antarctic Peninsula (EAP)-wide study of Christie et al. (2022), this paper provides important regional and historical context for Larsen B's 2022 behaviour in terms of the exact types of sea ice–ocean–ice sheet interactions reported here, which we believe serve to complement the findings/interpretation of the present study well. With reference to e.g. Section 5.1 (Line 463+), Christie et al. (2022) similarly attribute the relative period of glacial advance/stability and increased sea ice presence observed along the EAP over the last ~2 decades to the (multi-)decadal influence of the IPO and SAM.

On the importance of ice mélange for ice-shelf/glacier stability (cf. e.g. Line 525), the authors may also be interested in the following recent paper pertaining to the breakaway of iceberg A-68 from the neighbouring Larsen C Ice Shelf:

• Larour, E., Rignot, E., Poinelli, M., and Scheuchl, B.: Physical processes controlling the rifting of Larsen C Ice Shelf, Antarctica, prior to the calving of iceberg A68. PNAS, 118 (40), e2105080118, https://doi.org/10.1073/pnas.2105080118, 2021.

We thank you for suggesting literature that is complementary to our study and should be discussed within our text. You will find that we have incorporated the suggestions you pose here as well as those from Reviewer 1 and 2. Please see the revised discussion. We also note that it was not intentional to leave the relevant studies out of the original manuscript.

Comment 2
We agree with Reviewer 1 that it is essential that the correct grounding line position be used when analysing the velocity acceleration patterns of the various glaciers. With the above in mind, and as the authors themselves allude to in the paper (Lines 67, 525), the glacier acceleration trends observed following fast ice breakout (in early 2023, especially) appear to contradict the observations of another recent paper by Sun et al. (note: now published at:
https://agupubs.onlinelibrary.wiley.com/doi/epdf/10.1029/2023GL104066).

On the assumption that the acceleration trends reported in the present study still hold after the grounding line is updated and velocity uncertainties are quantified (cf. Reviewer 1's comments), then some additional words reconciling (or not …) the findings of the two studies would be useful to include beyond that already stated on Line 525. For example, could the differences in velocity signal be due to differing processing and/or smoothing techniques used? (probably unlikely unless errors in this study are high). Or perhaps due to the fact that for such a confined embayment, the glacier response was simply

**lagged until 2023 and hence not captured fully by Sun et al.'s analysis (whose observations ended late 2022)? Whatever the case, both studies ultimately motivate future research into the oft- overlooked nature of sea ice-ice sheet interactions at Larsen B and beyond.**

We have included the grounding line that Reviewer 1 suggests we use. We have also included the grounding line that we determined using somewhat similar but different methods. We discuss the results in context of both potential grounding lines. We are currently preparing another manuscript that details the evolution of the rapid retreat of Hektoria Glacier in which we discuss the grounding line determination in greater detail. You will also find that we have included more discussion on Sun et al., 2023 and the recent preprint by Surawy-Stepney et al., 2023 results in our revised discussion.

**Finally, on the subject of grounding lines, for ease of reference please also show the updated locations on each of the insets shown in Figs 1, 8, 9, 10 and 11.**

We have updated figures.

--
END
Frazer Christie (fc475@cam.ac.uk), Christine Batchelor, Wolfgang Rack & Julian Dowdeswell

---

## Author Comment (AC2)

Dear Reviewer 2,
We appreciate these insightful comments and for the constructive and helpful review of our manuscript. The notes in the text caused us to review our analysis carefully, and we have adopted or addressed nearly all of the comments.

Below, you will find our responses in blue. Thank you for your review.

Naomi Ochwat, on the behalf of the coauthors

**Comments on "Triggers of the 2022 Larsen B multi-year landfast sea ice break-out and initial glacier response" by N.E. Ochwat et al. , tc-2023-88**

**This paper presents an important case-study analysis of the region of perennial fast ice that formed in the embayment previously occupied by the former Larsen B Ice Shelf following its disintegration in 2002. Specifically it uses a combination of observational, reanalysis and remote-sensing data to examine the causes ("triggers") of a major fast-ice break-out event in 2022, and shows the effects of the latter on the speed, elevation and calving behaviour of various outlet glaciers feeding into the embayment. In so doing, it confirms the findings of a number of recent studies that highlight the important role of fast ice in mechanically buttressing adjacent glaciers and ice shelves, and the linkage between fast ice loss and changes glacier/ice shelf dynamics and calving behaviour - with this effect being modulated by surrounding pack ice that damps the impact of waves in breaking up the fast ice. An additional important finding of this study is that substantial decreases in glacier elevation occurred in response to loss of the fast ice buttress, in concert with major floe speed increases – in much the same way as occurred following the Larsen B disintegration.**

**In summary, this paper makes a valuable contribution to a growing corpus of work that highlights previously overlooked and poorly-quantified though crucially-important linkages between sea ice (change) – both in the form of stationary coastal fast ice and moving pack ice – and (change in) glacier and ice-shelf dynamics. This is particularly timely, given the current state of Antarctic sea ice and the increasing concern over Antarctica's contribution to sea-level rise.**

**Having said this, there are a number of issues that I feel need to be addressed in order to improve the paper. The science questions addressed are well within the scope of TC, and I recommend publication subject to substantial revisions, as laid out below.**

**Please find below my overall comments, followed by a more detailed listing of suggestions. I hope these are useful and help improve the paper.**

**Overview Comments**

**1. The paper itself is generally well written, apart from minor issues relating to inconsistent use of tense and minor grammatical errors. However, the text seems overly long, and could benefit from being substantially shorter and more concise. This would make it more readable while telling the story more clearly – leading to greater impact in this journal.**

We have attempted to do that, but in addition to trying to be more concise, we are addressing numerous detailed comments. Moreover, we have also incorporated ideas from the most recent publications related to the event. We shortened the introduction and description of the study area as well as parts of the discussion, but we have had to add to the results and discussion as well to address reviews and acknowledge the newly published works.

**2. The terms "sea ice" and "fast ice" are used interchangeably. Explain the difference between fast ice and pack ice upfront in the introduction, then refer specifically to fast ice and pack ice as appropriate (or overall sea ice – which is what the passive microwave dataset gives). See my Comment against Line 41 below (in Specific Comments).**

Thank you for this suggestion, we have added a definition of "pack ice" in the introduction and clarified in the text which sea ice type we are discussing.

**3. May I suggest that the Data and Methods Section be shortened and restructured around the variables and phenomena being investigated – with these being grouped accordingly – rather than listing (working through) the individual datasets themselves. This would also help focus the paper more fully on the story being presented, which is certainly a good one, while also minimising repetition and improving the "readability" of the paper. For example, an introductory sentence or two/three (preface) could be added immediately after the Section 3 heading (between lines 99 and 100), along the lines of – "The linkages between fast ice.................. and glacier events were detected and analysed using a combination of observational, reanalysis and remote-sensing data. (NB then briefly adding the high-level information about the individual datasets i.e., what they are/names, where they are obtained from)." Then, subsequent sub- sections could consolidate information currently scattered across the sub-sections by focussing on the different sub-topics - in a more logical sequence than is currently the case, and with emphasis on the techniques used. Section 3.1. could/would then become "Sea Ice Change and Variability" – pulling together information relating to how the fast ice breakout event was detected and monitored; detection of open-ocean corridors in the adjacent pack ice; and fast ice surface melt was determined and mapped. Then Section 3.2 could be "Glaciological Characteristics"; and Section 3 "Atmospheric and Oceanic Factors".**

Thank you for suggesting this, we did consider it. We found that due to the individual datasets being used in several different sub-topics structuring the methods section by dataset avoids unnecessary repetition in the Results section (e.g. for ERA-5 model data, optical imagery,

AMSR/E data, etc.). However, we have added a summary paragraph to address how the datasets were combined to evaluate the different key topics of the paper.

**4. The Results (Section 4) is very long and contains detailed information about the timings of the different events (corridor formation, fast ice breakup, calving, glacier acceleration, elevation change etc. for each glacier) that is difficult to follow. This section could be shortened substantially by (1) condensing the results into a Table (or two), and by (2) adding a timeline schematic along the lines of Figure 11 in https://agupubs.onlinelibrary.wiley.com/doi/full/10.1002/2014JF003223. This would again substantially improve the readability of the findings and increase their impact. It's hard to follow the different timings and events – and how they line up – in the current text.**

Thank you for this suggestion! We have added a timeline schematic to help aid in the understanding of the chronology of the events discussed in the paper (New Figure 12) and shortened the results section. Please see the modified text and new figure.

New Figure 12:

[Figure]

**5. The Discussion (Section 5) is also very long in the way that it works through all of the topics in sequence. This could be substantially shortened – and repetition of Results avoided – by focussing on synthesising the main findings into a coherent story around why the fast ice breakout did not occur until 2002, and what the effects were then on the glacier systems – referring back to the suggested timeline schematic figure and associated Tables.**

We have rewritten the discussion to incorporate Reviewer 1, yours, and the Frazer Christie et al. short comments.

**6. The claim in Lines 315-316 and elsewhere (e.g., Lines 495-496, 503-504) that in January 2022, a relatively ice-free corridor connected the fast ice front area to the open Southern Ocean for the first time since persistent fast ice formed in 2011 needs backing up with further evidence. This assertion is based on Figure 6b, which gives sea ice area in an offshore box for January 19 only for the years 2010 to 2022 (I'm sorry but I don't have access to Figure S5 which is also referred to). I did a quick search through past satellite sea-ice concentration images and also found the persistent occurrence of a corridor in February 2021 (for example). This begs the question – why did the fast ice breakout only occur in January 2022 and not earlier? It also suggests that the fast ice breakout in 2022 may be due to a combination of factors, and not only exposure to ocean swells (as stated in Lines 27-29 of the Abstract and elsewhere).**

We appreciate you bringing up a good point. During the available AMSR data (from 2010-present) we have not found another incidence of a significant additional open corridor besides the February 2021 example you provide. During February 2021, the corridor is not as open to swell from the northeast as it was in January 2022. We also examined the wave data for February 2021 and found that during the most open-water period, only one significant event occurred, and the waves during that period were coming from the southwest (likely due to a foehn event) i.e., waves that moved and increased in amplitude away from the ice front, not towards it. We have addressed this additional corridor and noted the need for both open corridors and southwest-advancing wave direction in the text.

**7. Section 5.3 should specifically refer back to, and compare the new findings with, other studies from elsewhere around Antarctica by Miles et al. (2018), Arthur et al. (2021), Greene et al. (2018) and Gomez-Fell et al. (2022) regarding relationships between fast ice presence/absence and both (1) glacier calving and (2) speedup (i.e., buttressing).**

Thank you for bringing to our attention several studies that we missed. We have revised Section 5.2 and 5.3 to better incorporate relevant literature that you have listed here as well as from the Short Comment by Frazer Christie et al. (https://doi.org/10.5194/tc-2023-88-CC1), posted on 4 September 2023.

**8. Re the Figures – may I recommend marking key phenomena/events referred to in the text directly on the figures (e.g., X, Y or the like), such that pointers can then be given in the text e.g., "....this event is marked X in Fig. XXa". This will greatly help the reader.**

Both the new graphic that you suggested and existing and added notation on the figures addresses this issue.

**9. General comment re the Figures – the colours are challenging to differentiate (at least for me) – e.g., Figures 8 and 9. Also, Figure 9 is too complicated – too many lines. This could be substantially simplified by reducing the number of lines (while including the results in a table). The satellite image in Figure 10 is indistinct and difficult to interpret – this could be improved by adding boundaries and marking key features.**

Thank you for suggesting changes to improve the figures. We have incorporated additional key events/phenomena as you suggested above, as well as brightening the satellite images so that they are clearer. The satellite images in Figure 10 and 11 display the location of the elevation points, which we have now made clearer and added key features.

In Figure 9 each line is a monthly velocity and the gradual increase followed by the more rapid increase in velocity is an important part of our findings. Additionally, we chose color palettes that are color-blind friendly as well as differentiable as to the changes that occurred pre-fast ice break out ('cooler', blue-er colors) and post-fast ice break out ('warmer' colors).

**10. The paper is generally well referenced, but I've made suggestions regarding adding a few key references that are missing. Also, the order of referencing is neither chronological nor alphabetical e.g., Lines 527-528 Just one other thing – I note that the authors (or rather the lead author and 2 others) have also submitted a shorter version of this topic for publication as a "sidebar" in the annual State of the Climate Report for 2022 (in press in the Bulletin of the American Met Soc) – and with a similar title. It may be best if the authors refer to this other publication upfront in this new paper. May I suggest that this information be added in a sentence at the very end of the Introduction (onto Line 72) – stating that a shortened version is in press in BAMS (and referencing that). However, this will also need to briefly state how this paper differs from that sidebar i.e., why this new paper is necessary.**

Thank you for catching that the citations in the text are not in a specific order. We have put them in chronological order and when there were multiple from one year in alphabetical order.

We have added the BAMS sidebar into the text and briefly describe the difference in the papers, as you suggest. The BAMS paper was not in press during the time of the submission of this paper and not citable as we did not have a preprint available online. As it is now published, we can add it in, thank you for suggesting that!

**Specific Comments and Suggestions**

**Line36 – 2008 and 2009 (add Braun, M., Humbert, A. & Moll, A. Changes of Wilkins Ice Shelf over the past 15 years and inferences on its stability. Cryosphere 3, 41–56, 2009).**

We have added the citation and dates.

**Line 41 – "....and outer-margin calving due to ocean swell-induced flexure (Massom et al., 2018). Massom et al. (2018) further implicate loss of attached landfast sea ice (fast ice) in the Wilkins Ice Shelf breakup events, following loss of a protective pack ice buffer offshore – due to the vulnerability of fast ice to ocean swells (Crocker and Wadhams, 1989). While fast ice is consolidated sea ice that remains stationary attached to the coast (Fraser et al., 2021), pack ice refers to sea ice that is constantly in motion under the influence of winds and ocean currents." REFERENCE - G.B. Crocker, P. Wadhams, Breakup of Antarctic fast ice, Cold Regions Science and Technology, 17 (1), 61-76, https://doi.org/10.1016/S0165-232X(89)80016-3, 1989.**

We added the suggested text you have above as well as a clause on the perennial/annual aspect of the fast ice. We removed the definition we originally included on original lines 56-57.

**Line 45 – what is meant by "increase ocean swell"?**

ARs can increase swell height, as discussed in Wille et al., 2022. We added increase ocean swell "height" to the text to clarify.

**Line 50 – replace "catastrophically" with "substantially".**

We have replaced it.

**Lines 56-58 ("Fast ice......") – remove.**

We removed it.

**Lines 58-60 – inappropriate to have this Result in the Introduction – move this to the appropriate place. It's also not clear how these thicknesses were derived. ALSO – line 60 – "containing both fast ice and glacial ice".**

We moved these lines to the results section.

**Lines 65-68 – this should also refer to other studies relating fast ice to glacier calving and advance/speed e.g., Miles et al. (2018), Arthur et al. (2021), Greene et al. (2018) and Gomez-Fell et al. (2022). There is a need to introduce the concept of the damping of waves by pack ice, with references. This is central to the ocean corridor concept proposed by Massom et al. (2018).**

We have added more in the introduction and included the references you suggest, as well as a few others.

**Lines 68-72 – define buttressing.**

We added "resistive stress" to the definition.

**Line 67 – add Massom et al. (2010) after "collapse".**

Done.

**Line 76 – "south".**

Fixed.

**Line 78-79 – To the east, the northwestern Weddell Sea is generally covered by pack ice.**

Added, thank you.

**Line 83 – NB there's more to the Larsen B breakup than this (hydrofracture) alone – refer back to Lines 37-41, and my Comment on Line 41 above. RE THIS, Lines 80-84 could probably be merged into Lines 33-36.**

We have merged the suggested sentences and rewrote part of the study area so as to not be repetitive of the introduction.

**Line 102-103 – remove "a climate....ECMWF)".**

Done.

**In Section 3 and in the appropriate place, add – "Following Massom et al. (2018) and Teder et al. (2022), we investigate the occurrence of open-ocean corridors across the sea ice zone, enabling ocean swells to interact in an unobstructed fashion with the Larsen B embayment fast ice".**

Thank you for the suggestion, we have added it.

**Lines 139-141 – not clear what this means.**

We have reworded this to clarify what we meant.

**Line 147 "several...images"**

Fixed, thank you.

**Line 152 – image cross-correlation**

Fixed, thank you.

**Line 161 – images**

We changed it.

**Line 162 "estimated from the location of a break in slope"**

Fixed.

**Line 164 – what is listric faulting?**

A listric fault is a fault with a curved plane decreasing in dip angle with depth, we have included the reference for this.

**Lines 179-180 – different tenses. Be consistent throughout the paper.**

We have striven to address this.

**Line 181-182 – what is meant by "Assuming snow is negligible"? Also, what is this assumption based on?**

According to van Wessem et al., (2016) the landfast ice area receives ~100-200 mm w.e. per year. This implies that the snow is likely less than a ~0.6 m thick layer (assuming snow density ~300 kg/m3) on a 5-10 m thick ice. We assume negligible because the snow is a small proportion of total ice thickness. Additionally, we state a wide thickness range (5-10 m), which includes any variability that snow thickness and density might have.

van Wessem, J. M., Ligtenberg, S. R. M., Reijmer, C. H., van de Berg, W. J., van den Broeke, M. R., Barrand, N. E., Thomas, E. R., Turner, J., Wuite, J., Scambos, T. A., and van Meijgaard, E.: The modelled surface mass balance of the Antarctic Peninsula at 5.5 km horizontal resolution, The Cryosphere, 10, 271–285, https://doi.org/10.5194/tc-10-271-2016, 2016

**Line 183 – why were these density values chosen (based upon what)?**

These values are based on the estimates in the literature. For example Zwally et al., (2008) use 1023.9 kg m3 and 915 kg m3 for sea water and sea ice, respectively. Due to the variability in sea ice density and the possibility of thin snow cover, we chose 900 kg m3. Without in situ data there is not a way to estimate the density accurately, however the differences in density estimates would result in incremental changes in the estimated thickness, and we are citing a thickness range of 5-10 m for interior ice, so these adjustments would be minor.

Zwally, H. J., D. Yi, R. Kwok, and Y. Zhao (2008), ICESat measurements of sea ice freeboard and estimates of sea ice thickness in the Weddell Sea, J. Geophys. Res., 113, C02S15, doi:10.1029/2007JC004284

**Lines 189-194 – Did AMIGOS also provide meteorological information?**

The AMIGOS station had several sensors on it. Here we focus on the GPS data as there is a paper in prep (Pettit et al.) that is utilizing the rest of the data to discuss the Scar Inlet Ice Shelf and relationship with landfast ice in greater detail.

**Line 202-210 – how was fast ice area determined. Were there any difficulties in distinguishing the boundaries?**

The fast ice was clearly determined along the rocky coasts. Where it met the glacier tongues, the differentiation became more ambiguous, but was usually distinguishable by how far the spacing was between the large (size) icebergs. It is, of course, a spectrum as the floating termini of the glaciers slowly transition into a melange mixture and then fast ice and it is not an abrupt change. The quantitative area was determined by measuring the area 3x and averaging those three estimates using the same qualifiers as what constituted fast ice. Additionally, we compared our estimate of the terminus to Sun et al., 2023's terminus location and found they were quite similar. The "outer portion" of the fast ice was the area that would break-out and reform regularly, as discussed in Section 4.1.1.

**Line 205 unclear – how does an edge reform?**

We have changed the wording to "outer portion".

**Line 209 – not clear what "the edge broke out" means.**

We have changed the wording to "outer portion".

**Lines 2002-210 – this needs a figure to show the sequence of events discussed, as a series of outer margin lines.**

We have made the figure that you suggested in the general comments. We have also made a gif of MODIS extents prior to the break-out for the supplemental information (also see below of a snapshot). We did not include the outer margin lines as they are variable throughout the season and may cause confusion since there would be so many.

[Figure]

**30-04-2011**

**Line 212 – "occupation of"**

Fixed.

**Line 218 – "reformed into"**

Fixed.

**Line 219 – "advanced 16 km from February 2011 to XXXX"**

Advanced 20 km from February 2011 to January 2022.

**Line 220 – "while Punchbowl"**

Corrected.

**Lines 223-224 – why is there a seasonal cycle in the Scar Inlet Ice Shelf flow speed? And is this a feature of all of the glaciers investigated? Please add this information.**

The seasonal signal in the Scar Inlet Ice Shelf flow speed is interpreted to be a result of seasonal warming and weakening of fast ice. The full analysis of the GPS and other data is not included in the current text because it is being examined in a paper currently in prep (Pettit et al., 2023). If the paper is submitted prior to our publication, we will be sure to discuss it.

**Line 232 – it's hard to see the fractures in Figure 1b and 1b – the images are very small. Also – it's not clear what Figures 1d-f show – maybe consider leaving these out.**

We have edited Figure 1 so that more features are evident. However, due to the location of the clouds and the need to show the entire embayment we can only zoom in so much. The main objective of the figure is to show the study area locations and key features during the break-up. It is unfortunate the clouds obscure the landfast ice plates drifting out in Panel C (Fig. 1) however, we prefer to keep that image in as it is the first image that is remotely clear after the fast ice began to break-out. Figures 1d-f show the immediate response that tributary glaciers had after the fast ice break-out, as the photographs were taken on 31 January. These images are important to show how quickly the glacier tongues disaggregated (Crane and Jorum) and calving at the grounding line began (Punchbowl). This is one of the key pieces of evidence that show that somehow the fast ice was buttressing the glacier tongues. To make this more clear, we have referenced the panels in additional text lines.

**Line 238 – not clear what re-enter means here - is this floes from outside moving into, or the formation of new ice within?**

Changed to "reappear", it is likely a combination of floes moving in, and formation of ice within the embayment.

**Line 238 – "sea ice coverage"**

Fixed, thank you.

**Line 239 – "winter 2022"**

Fixed.

**Line 240 – what is meant by "apparent coherency"?**

As we are using MODIS imagery, we do not know exactly how well connected the sea ice was however, it appeared coherent - as in one solid piece, this was also obvious by the movement of specific icebergs throughout the decade.

**Lines 242-243 – this sentence needs rewriting. Also, change plates to floes. Regarding "sea ice concentration varied" – over what area, and does this refer to pack ice or fast ice (noting that a feature of fast ice is its consolidated nature i.e., 100% concentration)?**

Text changed to: In October 2022 the sea ice in the embayment varied in spatial extent, and began to decrease significantly in November 2022, and by December 2022 there were minimal floating bergs or pack ice floes.

**Lines 245 and 246 – should "climate" be "meteorological" here?**

Good catch. We have fixed it.

**Line 245 onwards – need to refer to Crocker and Wadhams (1989) and Langhorne et al. (2001) here, regarding the fact that fast ice is particularly vulnerable to breakup by ocean waves. REFS:**
**Langhorne, P., Squire, V., Fox, C., and Haskell, T. (2001). Lifetime estimation for a land-fast ice sheet subjected to ocean swell. Annals of Glaciology, 33, 333-338. doi:10.3189/172756401781818419**
**G.B. Crocker, and P. Wadhams (1989). Breakup of Antarctic fast ice. Cold Regions Science and Technology, 17(1), 61-76, https://doi.org/10.1016/S0165-232X(89)80016-3.**

We have added these references, but in the discussion where we discuss the causes of the fast ice break-out.

**Figure 3 and Line 261 – make the blue box more prominent. Also, why was this location chosen, and why is 4 grid cells the size chosen?**

We have made the frame of the blue box slightly larger. We chose the location due to the proximity of the outer portion of the fast ice and open water. We experimented with the number of grid cells used and found that 4 was the optimal amount because it covered a large spatial area yet did not smooth out the important results.

**ALSO – is "surface temperature" surface air temperature?**

Yes, we have fixed that.

**Line 262-264 – unclear. Occurring when?**

We added the relationship to the event occurrence.

**Lines 271-273 – not clear as written.**

We fixed the tenses (as suggested previously).

**Lines 292-295 – ungrammatical – rewrite as 2 sentences.**

We fixed these sentences.

**Figure 5 – make the solid and dashed lines thicker.**

We have made the solid and dashed lines thicker.

**Line 307 etc. – is sea ice extent based on the 15% ice concentration threshold? (add this information to the appropriate Data and Methods sub-section).**

Yes, we use 15% or greater concentration for sea ice extent. We clarify this now in the methods section.

**Line 307 – why is January 19 chosen? This is the date of initial fast ice breakout, but what were sea- ice conditions like in the previous and subsequent days?**

Sea ice conditions did not vary significantly for the two-week period prior to the break-up event. The corridor that we infer as the path of wave access from the northeast opened in early January and remained open to January 19th - and beyond, except for the floes released by the break-up event.

**Figure 6b – the text above and y axis state "sea ice area", but the text talks about sea ice extent only.**

We meant to say sea ice area and not extent, using the NSIDC definition of area (https://nsidc.org/arcticseaicenews/faq/). This has been fixed throughout the text. Good catch!

**In Line 308 – should "time series of sea ice extent (concentration multiplied by area of pixel)" be"time series of sea ice area (concentration in each pixel multiplied by the number of ice-covered pixels)"?**

Yes, thank you. We fixed it and the rest of the text in this paragraph.

**Figure 6b – also, why is the value given only for January 19 in all of the years? This could be misleading to interpretation of when and how long open-ocean corridors occurred. Also in lines 315-316 – Figure 6b does not back up the statement that no other corridors occurred over the period from 2011, as it shows January 19 only. For example, I had a quick look at the satellite data and this shows the persistent occurrence of a corridor in February 2021 (for example). This leads to the question – why did the fast ice breakout**

only occur in January 2022 and not earlier? Therefore, the claim in Lines 315-316 that "This pathway, which allows for wave action to access the front of the Larsen B fast ice, had not been present since the fast ice's formation in 2011" needs backing up with further evidence. This comment also applies to Lines 495-496 – "Therefore, for the first time since the formation of the persistent fast ice cover in 2011, a relatively ice-free corridor connected the fast ice front area to the open Southern Ocean." (I'm sorry but I don't have access to Figure S5 which is also referred to). Also Lines 503-504 etc.

Please see General Comment #6, as well as our response to your comment on Line 307 above. We have modified the text accordingly. The February 2021 corridor was never completely open for the Larsen B ice front, and the one significant wave event in our model wave reanalysis data propagated out from the ice front outward (due to foehn winds). There were no other significant openings that we identified in the 11 year fast ice period.

**Line 325 – open-ocean (sea ice-free) corridor**

Corrected, thanks.

**Line 329 – equivalent to a wavelength**

Corrected.

**Line 343 – change "4.3.1 Initial retreats of landfast ice and glacier fronts" to "4.3.1 Retreat of glacier fronts"**

Corrected.

**Line 347 evidenced by**

Done.

**Line 356-358 – what does this mean, and why is it important?**

We include this sentence to draw comparisons to other known calving styles to put the calving style of the Larsen B Glaciers in context. We added a clarification that the calving was specifically buoyant calving.

**Line 372 – what is meant by "Hektoria and Green Glacier responded to the collapse in later months following the fast ice break-out"? What collapse?**

Changed to "fast ice break-out".

**Lines 374-376 – unclear as written.**

We reworded the sentences for clarity.

**Figure 8a and 8c – length scales are missing.**

We chose not to include the length scales on all of the panels because a and b are the same and c and d are the same. We have modified the caption of the figure to note that they are the same.

**Section 5 Discussion. This section seems overly long, and may repeats much of what has been stated before. Much of this information could be captured more concisely in a well-formulated schematic along with Tables – see my General Comment 6 above.**

We have modified the discussion to take into account the various comments by Reviewer 1, yourself, and the short comment from Fraser Christie et al. Please see the revised section.

**Lines 390-391 and Section 4.3.2 – Did Evans Glacier also show a speed change?**

We did not include Evans in the analysis of the tributary glaciers. For information on other tributary glaciers in the embayment we show Punchbowl and Jorum in the supplemental information. We intend on evaluating Hektoria, Green, and Evans in greater detail in our follow-up paper on the HGE system's retreat.

**Line 396 – what is meant by noise levels in the data? What are they?**

We changed this sentence to the following:
"Additionally, the observed speed profiles in the 26-month period (January 2021 to March 2023) show far less local variability upstream of our inferred grounding line."

**Figure 10 – the satellite images are indistinct and difficult to distinguish. Please mark of features and important boundaries.**

We have modified Figure 10 and 11 to highlight the locations of the elevation data.

**Line 462 onwards - As stated above in General Comments, the Discussion (Section 5) is also very long in the way that it works through all of the topics in sequence. This could be substantially shortened – and repetition of Results avoided/minimised – by synthesising the main findings into a coherent story around why the fast ice breakout did not occur until 2002, and what the effects were then on the glacier systems. This would then naturally refer back to the suggested new timeline schematic figure and associated Tables.**

Thank you for the suggestion. We have modified the discussion to incorporate your suggestions, as well as Reviewer 1, and the short comment by Frazer Christie et al..

**Line 463 – "Synoptic scale climate patterns" may be confusing, as synoptic is a meteorological term referring to the approximate horizontal scale of cyclones. Maybe replace with "Meteorological conditions. This comments also applies to other places where "climate" is used e.g., Line 464.**

We have changed it to "Meteorological Conditions and Modes of Atmospheric Variability" and discussed it in the context of large-scale modes of variability instead of synoptic scale climate patterns.

**Line 477 – "eastern (lee) side"**

We have corrected this.

**Line 480 – should low concentration be zero concentration?**

We have fixed this to say "lack of sea ice".

**Line 496 - damping**

Good catch.

**Lines 498 and 507-509 – need to add the references to Langhorne et al. (2001) and Crocker and Wadhams (1989) here.**

Done, thanks.

**Lines 507-509 – not clear whether this is referring to fast ice or glacier ice.**

It can occur in both types of ice, we clarified that now.

**Line 511 – 'the broken-out fast ice had drifted 9-16 km"**

We fixed it, thank you.

**Lines 515-517 – again, it is not clear whether this is referring to fast ice or glacier ice, or both. Hydrofracturing a process associated with crevasses on ice shelves/glacier, and has yet to be observed on fast ice.**

We have clarified which ice we are referring to here:
"Here, we found that foehn events happened prior to, during, and after the January 2022 Larsen B wave event, potentially causing interior hydrofracturing in the ice tongues, after the ocean swell fractured the outer margins of the fast ice, thereby redistributing the stress within the ice that included fast ice and glacier tongue."

We do not think the fast ice hydrofractured because it is not thick enough, however, we cannot rule it out for the interior thick glacier tongue ice. Though the break-out style does not resemble that of other events interpreted as hydrofracture calvings, the HGE tongue has some similarities so we cannot say for certain that it did not happen on the HGE tongue.

**Lines 517-519 – This is similar to the finding of Massom et al. (2018). They found that strong and persistent offshore westerly/northwesterly winds in late 2001 through early 2002 both (1) created a persistent sea-ice free corridor offshore from the Larsen B Ice Shelf to enable swell penetration that contributed to the ice-shelf breakup, then (2) blew the resultant icebergs and melange out of the Larsen B embayment. Please refer to this parallel here.**

We have added this in, as well as other literature as suggested by the short comment.

**Line 520 onwards (Section 5.3) – please specifically refer back to, and compare the new findings with, other studies from elsewhere around Antarctica by Miles et al. (2018), Arthur et al. (2021), Greene et al. (2018) and Gomez-Fell et al. (2022) regarding relationships between fast ice presence/absence and both (1) glacier calving and (2) speedup (i.e., buttressing).**

We have rewritten the discussion to incorporate these discussions as well as those suggested by Reviewer 1 and the short comment by Frazer Christie et al..

**Line 524 – replace "despite" with "contrary to".**

We have fixed this, as well as added more information

**Lines 525-529 – I didn't quite understand these 2 sentences, and how these factors relate to the findings of this paper.**

Please see the revised section as we have rewritten a large portion of it.

**Line 569 – effects on glacier flow and decreased surface elevation. Also reference Rignot et al. (2004) and Scambos et al. (2004) here.**

We have incorporated this suggestion, thank you.

**Line 560 – that the sea ice concentration in the Weddell Sea in 2022 was the lowest recorded is somewhat ambiguous. When did this occur (in the year)? Also, does this refer to the entire Weddell Sea?**

**ALSO – there was a large sea-ice free corridor prior to and during the Larsen B disintegration event in 2002 (see Massom et al., 2018)**

We have fixed this. We have incorporated more discussion of Massom et al., 2018 and the similarities in the discussion portion of the text.

**Line 555 onwards (Conclusions) – again, please place the current findings more in the context of previous studies.**

Please see the revised conclusions.

**Lines 557-559 – where is it shown that high temperatures (alone) caused the ice-free corridor? Is it more likely to be wind-driven?**

It is likely wind-driven as you suggest, as well as linked to large scale climate patterns such as SAM and the position of the ASL. We have changed the conclusion and revised accordingly.

**Lines 559-561 – this may not be the case – refer to Massom et al. (2018) regarding the extraordinary opening in late 2001 through early 2002. Also see my comments above and in the Overall Comments.**

We have revised the conclusions. Additionally, we added the Turner et al., 2022 reference as they state Antarctic sea ice was at an all time low in the satellite record. We specified that is what we were referring to and added more detail on the Weddell Sea ice extent specifically. We also acknowledge that the spatial configuration of the sea ice determines the access to the ice shelves that it is not only about the low extent but also where the low sea ice concentration resides spatially.

Turner, J., Holmes, C., Caton Harrison, T., Phillips, T., Jena, B., Reeves-Francois, T., et al. (2022). Record low Antarctic sea ice cover in February 2022. *Geophysical Research Letters*, 49, e2022GL098904. https://doi.org/10.1029/2022GL098904

**Lines 563-564 "pack ice-free corridor"**

Fixed.

**Lines 564-565 – fast ice flexure would not be confined to the outer margins – see Langhorne and Crocker and Wadhams papers.**

We have revised the conclusions, please see the new paragraphs.

**Line 565 – hydrofracture is not a process that has been associated with fast ice.**

Changed to glacier tongues.

**Line 571 – replace "The fast ice was clearly buttressing..." to "This suggests that the fast ice slab was acting to buttress..."**

We have revised the conclusions, please see the new paragraphs.

**Line 571-572 – this is a place to reference previous studies i.e., "confirming the findings of previous studies e.g., Massom et al. (2018), Miles et al. (2018), Arthur et al. (2021), Greene et al. (2018) and Gomez-Fell et al. (2022).**

We have referenced the studies in the discussion and revised the conclusion section.

**Lines 576-586 – Suggest combining the 2 paragraphs into one coherent paragraph.**

We have revised the conclusions, please see the new paragraphs.

**Line 576 – move the Fraser reference to "...fringed with multi-year fast ice (Fraser et al., 2021)..."**

We have revised the conclusions, please see the new paragraphs.

**Line 577-578 – this is not a new trigger mechanism. Suggest changing to "Antarctica's coastal fast ice may become more susceptible to breakup due to increasing exposure to ocean swells via open- ocean corridors through pack ice (Reid and Massom, 2022; Teder et al., 2022)."**

**REF: Reid, P.A., and R.A. Massom. 2022. Change and variability in Antarctic coastal exposure, 1979– 2020. Nature Communications, 13, 1164, https://doi.org/10.1038/s41467-022-28676-z**

We have added this reference to our revised conclusions.

**Line 580 – change "are" to "will likely be". Also, what is meant by "similar to ice shelf tributary glaciers"? – suggest removing this.**

Thank you for the suggestion, we have incorporated it.

**Lines 584-586 – change to: "It is important to continue monitoring not only the glaciers feeding into the Larsen B embayment in terms of their response to changing fast-ice conditions, but also other key glacier-/ice shelf-fast ice interactive systems around Antarctica and their response to increased coastal exposure (Massom et al., 2022; Teder et al., 2022)".**

Great, thank you.

**Line 602 Data Availability – change "data is" to "data are", and in line 607 add "data".**

Fixed.

---

## Author Comment (AC3)

**Response to Dr. Helmut Rott**

We thank the reviewer sincerely for a very careful and thorough review. We appreciate these insightful comments and for the constructive and helpful review of our manuscript. The notes on the text caused us to review our analysis carefully, and we have adopted or addressed nearly all of the comments.

Below, you will find our responses in blue.
Naomi Ochwat, on the behalf of the coauthors

Comments on "Triggers of the 2022 Larsen B multi-year landfast sea ice break-out and initial glacier response"
by N.E. Ochwat et al.

This paper reports on the break-up of land-fast sea ice in the Larsen B embayment and the initial response of glaciers after the buttressing sea ice had drifted away. Furthermore, the authors studied various potential triggers leading to this process, including changes in atmospheric and circulation and sea ice cover during preceding and concurrent periods, checking numerical meteorological re-analysis data and changes in satellite-based sea ice concentration over an extended area. Patterns of fast ice break-up and the retreat of glacier fronts are documented, using optical satellite imagery. The analysis of glacier response is based on satellite observations, focusing on several Antarctic Peninsula outlet glaciers draining into the Larsen B embayment. Repeat observations of surface elevation on specific points and flow velocities along central flowlines are shown for a period spanning the break-up event.

By and large, this is a well written article, presenting interesting material on the land-fast sea ice break-up event, its potential causes and the response of glaciers. The work confirms the influence of land-fast sea ice on glacier flow dynamics, reflecting - with opposite sign - the slowdown, reduced downwasting and frontal advance of Larsen B glaciers during the previous period when fast ice was built-up. Still, there is need for major checks and improvement, as there are various issues lacking traceability or being inconsistent with previously published data.

Main issues:
This section includes a short summary on main issues to be clarified and improved. Details on the concerns and suggestions for improvements are put forth in the next section.

1) **Location of grounding zone: The grounding zone positions are outdated. The 2016 grounding zone position of Hektoria Glacier was 12 km inland of the position shown in Fig. 9, and of Crane Glacier 5 km inland. This mismatch has major implications on various issues such as the interpretation of glacier flow dynamics, the estimation of the thickness of floating ice and the assessment of driving mechanisms for frontal retreat.**

We agree that the mismatch in grounding zone location has widespread implications on various glacier dynamics. In our analysis, we have found that there is a distinct break in slope and transition in calving style at the location of our estimated grounding zone, as well as the absence of surface depressions that are usually indicative of basal crevasses. Though we have reason to suggest the grounding zone is further downstream than the 2016 estimates, we agree that in this paper we do not present enough conclusive evidence to establish the 2022 location of the grounding zone. Our focus of the paper here is to show that the glaciers underwent rapid retreat and speed-up after the removal of the fast ice. Further details on the dynamics and details of Hektoria's retreat are outside the scope of this paper (but are planned for later work). Therefore, we have added the Rott et al. 2016 grounding zone to the figure and the text.

2) **Extent and thickness of floating sections of glacier tongues: The floating section of the HGE terminus, extending between the glacier fronts of 2011 and of 21 November 2021 (shown in Fig. 8), covers more than 200 km2 in area. In the manuscript ice thicknesses of the floating terminus up to some hundred metres are mentioned. This is not an agreement with mass continuity, as it would exceed the ice volume delivered across the flux gate close the 2011 ice front.**

Below, we present ICESat-2 data that shows that the freeboard elevation of the outer edge of the HGE floating tongue on 4 February 2022 was ~10-15m above sea level (new supplemental figure 1). This is approximately 16km down flow of the 2011 grounding line (yellow outline, panel A). Panel A in the figure below shows the 28 January 2022 ice edge during a period of open water at the front. Panel B shows a MODIS image from 4 February 2022, the same day that the ICESat-2 data was acquired, showing that sea ice was in the embayment during the time the elevation data was collected. Panel C is a Worldview image from 9 February 2022 showing a detailed view (blue box on Panel A) of the ICESat 2 Spot 3 track. Panel D shows the elevation data collected on 4 February of one of the ICESat-2 tracks (Spot 1 track, in green), showing the ice edge near the center of the HGE floating front (red circle, red arrow). The Hektoria ice tongue freeboard was ~10-15 m, consistent with an ice thickness of 80-120 m; but within 3-4 km upstream it thickens to an elevation of 25-30 m that is consistent with floating ice of 200-240 m thickness. Panel E shows a plot of ICESat-2 Spot 3 track (purple track), where an ice edge is present for a small portion of the track (Panel C, pink circles). This portion of the floating tongue has a freeboard of 30-35 m, consistent with an ice thickness of ~240-280m; again, within 2-3 km upstream the ice thickens to 40-45 m freeboard, i.e. ice thickness of 320-360 m.

[Figure]

In addition to the ICESat-2 data, Worldview DEMs from later in 2022 show a similar freeboard for tabular icebergs near the Hektoria front, over 40 m at its center, see below (note North is grid north in ESPG: 3031; new supplemental figure 2). This is consistent with a thickness of >300 m.

[Figure]

In combination, these data show that the ~250 km$^2$ HGE floating tongue had ice thicknesses of at least a few hundred meters, even along the outer edges. This suggests our estimates are closer to the 70-100 km$^3$ volume range from 2011-2021. It is unclear how to reconcile the mass continuity question at hand given the multiple observations of the HGE floating tongue ice thicknesses.

3) **Processes of fast ice break-up and drift: The discussion on these processes refers to changes in the backstress of the fast sea ice as main factor, but estimates on the stress magnitude are not provided. It is questionable if changes in the backstress due to depletion of fast ice in a widening bay (as in front of HGE glacier) can be the main trigger or if other factors play an important role as well.**

In reconsidering our evidence, the comments of Reviewer 2, and the recent publications (Sun et al., 2023 and the preprint by Surawy-Stepney et al., 2023), we have modified our text (and our thinking) somewhat. The majority of the backstress on the glaciers may have come from the ice tongues' interaction with the fjord walls; loss of the fast ice re-activated calving and rifting of the ice tongues, which in turn reduced backstress on the glaciers, leading to their acceleration. When examining Figure 4 in Sun et al. (see below) there are areas of melange and fast ice compression that indicate that the downstream fast ice was indeed providing resistive stress. Moreover, the GPS results from the Scar Inlet station cannot be easily reconciled without some fast ice buttressing (that apparently varies seasonally), which will be discussed by Erin Pettit et al., (in prep). When looking at ITS_Live data (see below) it is apparent that there was a significant slow down during the fast ice occupation of the embayment as well (explored more in our follow-up paper). The rapid disaggregation of the multiple several hundred meter thick floating glacier tongues that occurred immediately after the loss of the fast ice, indicates that the fast ice provided enough backstress to keep them stable prior to the fast ice break-out. The sequence of events before and after fast ice loss, and the response of the ice tongues and glaciers in the days and months following the loss, seems hard to reconcile with a system in which the fast ice played no resistive role.

As an aside, the fast ice also eliminated wind traction on the sea surface in the area just in front of the glacier tongues. The loss of fast ice in the Larsen B embayment could have led to a significant sea surface slope in the embayment in late January 2022 and beyond. This has been shown to be capable of gravitationally 'pulling away' chunks of the floating ice (Francis et al., 2021, TCryo). This would be an interesting feature to examine in a different paper but is currently out of the scope of our study, though we now mention it in the discussion.

[Figure]

Figure 4 from Sun et al., 2023.

Hektoria Glacier slowed down during the period of fast ice occupation. ([https://mappin.itsliveiceflow.science/chart?lat=-64.93231&lon=-61.65527&c=b](https://mappin.itsliveiceflow.science/chart?lat=-64.93231&lon=-61.65527&c=b))

[Figure]

Velocity data generated using auto-RIFT (Gardner et al., 2018) and provided by the NASA MEaSUREs ITS_LIVE project (Gardner et al., 20XX).

4) **Processes of glacier flow acceleration and frontal retreat: Similarity of the glacier response in the wake of fast ice removal to the response after the 2002 ice shelf disintegration is claimed. However, the prerequisites and course of the events are quite different. The 2022/2023 case refers to recently formed, thin floating glacier tongues of structural weakness, in contrast to the compact, grounded ice bodies of the pre-2002 period. The patterns of frontal retreat and the ice fracture processes are also rather different.**

We agree that the two main break-up events that have affected the Larsen B tributary glaciers are different. The Larsen B Ice Shelf was much thicker and longer-lived, and provided significant backstress to the glaciers. However, despite being thinner, as we have shown the fast ice must also have provided backstress, enough such that when removed the glaciers tongues immediately broke apart and drifted away, and within months the outlet glaciers retreated and accelerated. The central portion of the fast ice was thin, 5-15 m thick, yet the floating glacier tongues were several hundred meters thick. When they were removed, Crane and Hektoria clearly reversed their decade-long trend of deceleration after the fast ice broke-up. Naturally, as the glaciers were several hundred meters thicker prior to 2002, the magnitude of their response to the two events differs, yet the critical importance is that sea ice and fast ice can affect glacier dynamics. As the climate continues to change, there will be continued losses of sea ice and fast ice that will likely cause glacier acceleration, as we see here. We have added more clarification in the discussion on the differences of these two events and emphasize that the importance of the fast ice is not proportional to its thickness (See section 5.3).

**Details:**
**L53-L56: It would be of interest mentioning in this context also the 10-fold ice speed increase in 2009 at the Hektoria Glacier front compared to the years before 2002 (Wuite et al., 2015) and the declining mass losses during the period when fast ice was present. The total HGE losses of grounded ice amounted to 4.26 Gt/yr in 2011 to 2013 and 1.75 Gt/yr in 2013 to 2016 (Rott et al., 2018).**

We have added more to this paragraph describing the changes in HGE, as you suggest.

**L59-L60: Taking into account the ice volume discharged from grounded ice to floating sections of the glacier termini and assuming mass continuity, an ice thickness of hundreds of metres can refer only to a very small portion of the floating glacier tongue at its full extent. Rott et al. (2018) derived the mass fluxes across gates close to the 2011 glacier fronts of Larsen B glaciers for the period 2011 to 2016. Based on this number and accounting for further slowdown after 2016, the estimate of the total ice volume supplied to the newly formed section of the HGE tongue amounts to 39 km3 between 2011 and 2021.**

Please see our explanation at the beginning of the response (#2).

**L93-L98; Fig 1: The scale of the figures is too small for providing clear information on features of interest in the glacier bays and Larsen B embayment (e.g. structural properties, deformation patterns, rifts, glacier fronts). The displayed ICESat freeboard map shows blocks and streaks, with discontinuities along straight lines. Furthermore, the grounding zone positions (though not well traceable at that scale) seem to be outdated (see comment on L343ff).**

We have edited this figure to show greater detail of the glacier fronts. We adjusted the area of the interpolation, and re-interpolated with additional data, to emphasize the glacier fronts and fast ice thickness. Some of the structural properties, deformation patterns, rifts, and glacier fronts are not the primary focus of this figure and are shown in Figures 8, 10, and 11. The thicknesses of various points along the floating glacier tongues and (potentially) grounded ice are shown in Figures 10 and 11. We have also added the 2016 grounding zone from Rott et al. (2018) to nearly all of the figures.

**L108-109: Relative humidity less than 79% is not a suitable criterion for determining surface melt conditions. If the vapor pressure on the snow surface is higher than in the atmosphere sublimation may take place also at air temperatures above 0°C, depending on the net incoming energy.**

We follow the method that Laffin et al. (2022) uses that requires a combination of parameters to determine foehn conditions - not only humidity of the descending air. We agree that surface melt can occur at various humidity levels, but in our case we are interested in the presence foehn wind conditions. According to Cape et al. (2015), the criteria for foehn wind conditions include

an increase in temperature of 1°C per hour, an increase in wind speed above 5 m s⁻¹ from a westerly direction, and a decrease in relative humidity of at least 5% per hour. Using the criteria in both Laffin et al. (2022) and Cape et al. (2015), the events that we have shaded in S2 (Now S4) refer to foehn conditions. The first event in S2 (16 January) is the only one that does not match both methods here we refer to Laffin et al. (2022) methods to include it because it is the most recent analysis of foehn conditions in this area. We include more information on this in the manuscript.

**L167-L171: Please provide information on the procedure for deriving flow velocities (matching window size, sampling steps, cross-correlation threshold) and on the uncertainty of the velocity products for the different sensors and time spans.**

We derived the flow velocities using two different methods. Using the Alaska Satellite Facility Vertex Tool we were able to derive velocities with Sentinel 1 Synthetic Aperture Radar imagery. The Vertex Tool utilizes autoRIFT, an automatic image feature tracking tool. autoRIFT includes an iterative process for determining the flow velocity. At first, with a given chip and source size, a sparse search is performed to determine areas of low coherence, then a dense search is performed, where each source and chip is matched by identifying the peak normalized cross-correlation (NCC) value. Next, whenever a specific chip size does not estimate displacements, then the program increases the size of the search chip, resulting in nested grid design with various chip sizes. The larger chip size correlation is performed on a resampled coarser grid. In other words, in areas of high flow speeds, a larger window and chip size will be needed and the software accounts for that. The final displacements are posted on the smallest chip size grid. Therefore, the final downloaded raster product has varying window size, chip size, and correlation threshold. Subsequently, the errors vary, however on average Sentinel-1 has a relative error of 4% for both X and Y-direction velocity (Lei et al., 2021). This is likely to be an underestimate because the only way to get the absolute error is by using in-situ measurements, like GPS/GNSS instruments deployed on the ground and comparing it to the satellite data.

To derive velocities using Landsat optical imagery we used PyCorr. PyCorr measures ice displacement between two images by finding the peaks in normalized cross-correlation surfaces between image chips extracted in a grid pattern over both images. Each image pair had several window and chip sizes tested and the optimal combination was chosen for the analysis. Ice flow speed errors are a combination of geolocation errors for the image pair (typically ~5m) and measurement errors on the individual correlations (0.1 pixel or 1.5 m; Fahnestock et al., 2016). PyCorr returns a limited set of metrics that help document the uniqueness and strength of a peak that can be used to filter the output, but it does provide an error estimate for each match. With that said, for images separated by one year, error is ~±7.5m/yr, however shorter time intervals result in higher errors (Fahnestock et al., 2016). The PyCorr results we have agree with the available Sentinel 1 autoRIFT results.

Given the complexity of both of the algorithms and how they process image pairs we have not added information on the window and chip sizes. However, we have added information

elaborating on how these processes work and errors associated with the different methods. Please see the revised methods section.

**L215- L216: Considering the ice export across the 2011 glacier front, ice with thickness on the order of 300 m can stretch out only over rather small areas. Please provide details on the ice thickness information (dates, extent of area with freeboard > 40 m). Due to the coarse resolution the Fig.1 is not a suitable source for this information.**

Please refer back to response #2 at the beginning of the response.

**L217-L218: The maps of surface elevation change (SEC) 2013 to 2016 by Rott et al. (2018) show increase in surface elevation on the lower 10 km of Crane Glacier and frontal advance, coinciding with major surface lowering in the upper reaches of the main glacier trunk. This contradicts the statement on dominating ice input from tributary glaciers in the paper of Needell and Holschuh (2022) who do not cite the observations of Rott et al.**

Both Rott et al., (2018) and Needell and Holschuh (2022) describe the main trunk of Crane glacier as increasing in surface elevation during the period of fast ice occupation. Needell and Holschuh (2023) refer to the tributary glaciers as thinning, which are actually the upper reaches of the main glacier (see screenshot of their Supplemental Figure 7 below). Given this we see the statements as not contradictory but in agreement. We have added the Rott et al., (2018) citation to this sentence in the manuscript.

[Figure]

Figure S7. Extended surface elevation profile, capturing evidence of upstream elevation change. Local thickening upstream of the 2021 glacier terminus occurred ~12 km along the flowline from 2008-2021, and a period of thinning 12-30 km along the flowline occurred from 2002-2008. Data represent a combination of NASA airborne altimetry data (Airborne Topographic Mapper data and Land, Vegetation, and Ice Sensor data), and ICESat-2 calibrated stereophotogrammetric DEMs.

**L218-L219: Referring to the glacier fronts on 25 June 2011 (TanDEM-X image, shown in Fig. 2 in Supplement of Rott et al., 2020) and on 21 November 2021 (shown in Fig. 8 of**

**Ochwat et al.) the Hektoria Glacier front advanced during this period by about 20 km and the new floating area covered more than 200 km2.**

We edited the text accordingly. Calculating the area from the 2011 coastlines in Rott et al.., 2020, the area covers approximately 250 km$^2$.

**L232: Fig. 1b and 1c show MODIS images of 16 and 21 January 2002, not 18, 19 and 20 January.**

We have fixed this mismatch and added the correct dates.

**L261-L264: Please explain the decision rule for identifying and for shutting on and off foehn events. According to Fig. S1 a threshold of 3 m/sec is used for Foehn detection. This number corresponds to a light breeze, not suitable for facilitating the breakthrough of the flow at the leeside of a mountain range.**

As noted earlier, we use the empirically derived thresholds from Laffin et al., 2022 in section 2.2. The wind and humidity thresholds are only used in concert with other parameters. We also refer to Cape et al. (2015) that uses slightly different criteria, yet it agrees with our identification of foehn events for all but one event, in which we include because it matches Laffin et al. (2022). In Cape et al. (2015) the criterion for the wind is 5 m/s, which is the case for all but one of our identified events. Additionally, in Cape et al. (2015), the foehn conditions are split into "foehn days" where the conditions are met for longer than 6 hrs, when those conditions are no longer met then the foehn day ends. We have adjusted the text to include Cape et al. (2015)'s criterion of the "foehn day". We note foehn events only to describe weather conditions that might have induced melt or could move the fast ice.

**L274, Fig. 4: The label of Fig 4a abscissa is date (not time).**

Good catch, thank you. We have fixed it.

**L295: The surfaces may have been frozen on the date of the Landsat image (16 January 2022), but the temperature record (Fig. S2) shows several days of high temperature in January 2022, an indication for with surface melt that is also apparent in SAR images.**

We say with "reduced melt pond coverage" not that there is not any melt at all. There very well may have been some melt, but not to the extent that was evident in the November and December satellite images.

**L343ff, Section 4.3.1: In the context of various issues addressed in this section, an outdated version of the grounding zone positions is used. Estimates for grounding line locations of Larsen A and Larsen B glaciers in 2013 and 2016 were obtained by Rott et al. (2018), based on changes in surface elevation of TanDEM-X data. The transition from grounded to floating ice is associated with a strong drop in surface elevation change**

**(dh/dt). The 2016 grounding lines of Hektoria Glacier and Crane Glacier (see zoom images in Rott et al., 2020) are about 12 km (Hektoria) and 5 km (Crane) inland of the position shown in Fig. 9 of Ochwat et al. The 2016 Crane location refers to the centre of the canyon. Further retreat during recent years has to be expected.**

We have added the 2016 grounding zone that was referenced in Rott et al. (2018) to various figures, and in the text. (Thank you for having that data accessible and easy to add!). We have kept the grounding zone as we inferred as well, which were determined using the change in slope from the 2020-2022 Worldview DEMs, the onset location of surface depressions indicative of basal crevasses, and the location of a transition in calving style.

Due to the buttressing effect of the landfast sea ice (evident in the ITS_LIVE velocity data of Hektoria above, as well as the GPS data from Scar Inlet, Fig. 2), the grounding line may have not continued to retreat during these years. Due to the complexity of our assessed grounding zone migration, we are saving a deeper analysis of these dynamics and rapid retreat of Hektoria Glacier for a future study (now in prep).

**L346-L350: Please provide evidence on the iceberg properties and calving features described, as well as on the dates of the images to which the described features refer. Checking the pre-frontal embayment area in the Sentinel-1 time series of 2022 and 2023, there are no features that can be uniquely allocated to a toppled iceberg. In the SAR images of January and February 2022 and December 2023 to February 2023 icebergs in the pre-frontal areas show low reflectivity, evidence for the presence of melting firn. In subsequent colder periods the firn freezes, causing a transition to high backscatter intensity.**

Several characteristics indicate the type of calving style that occurred. Tabular icebergs are evident in optical imagery by the coherent structure that resembles the surface of the glacier and has simply detached from it, usually significantly crevassed and darker due to shadowing. Rotated or toppled icebergs, a product of buoyancy-driven calving (or in some cases, forward tipping due to high basal stress at a tidewater ice front), are usually lower to the ocean/sea ice surface, smooth with minimal crevasses, and have a gently undulating upper surface. In some instances, we used SAR backscatter intensity in conjunction with the optical imagery. We agree that the refreezing of firn can cause a transition to high backscatter intensity, which is why we did not only use SAR for identifying toppled bergs.

Below you can see an example of the tabular berg next to a toppled berg in front of Crane Glacier in March 2022. At this time, Crane was still calving from its floating tongue and not yet at the grounding zone, hence having both types of icebergs.

[Figure]

In the figure below you can see the toppled berg at the front of Hektoria Glacier. The elevation profile (from Worldview DEM from the same image, orange line) shows the backward rotation of the berg. The red dot shows the crest of the iceberg in both the image and the profile. The features of this iceberg, including the elevation profile, morphology of the iceberg, and surface texture, clearly show it is a buoyantly calved iceberg.

[Figure]

**L373: An extended thick (>300 m) floating tongue in front of HGE glaciers does not match the mass continuity for input across the 2011 glacier front (see related comments above).**

Please refer back to response #2 at the beginning of the response.

**L389-L428, Section 4.3.2: Also for this section an update on the grounding zone positions is needed (see comment L343ff). The main parts of the velocity profiles shown in Fig. 9 are located either on floating ice and or on pre-frontal ice mélange. This questions the argumentation of issues referring to the grounding zone and to velocities on grounded ice.**

Please refer to response #1. We have added the location of the 2016 GL's to the figure.

**L420, Figure 9: Please mark the updated location of the grounding line and the location of the ice front on different dates. The profiles of Hektoria and Green glacier deviate from the course of the central flowline. The Landsat-based Hektoria Glacier velocities of Nov. 2022 to Jan. 2023 deviate significantly from the Sentinel-1 based velocities of June to Oct. 2022. How reliable are the Landsat velocities?**

We follow the central lines of the Icebridge ATM track. The velocities deviate because there is a significant speedup that begins in the autumn, this is evident in the Sentinel 1 velocity data on Crane, as well as the Sentinel and Landsat velocity data on Green and Hektoria. You can see that the January 2023 and December 2022 velocities of Sentinel and Landsat are in agreement on the Green Glacier profile. The Landsat velocities have been used in numerous publications and have frequently been compared to and merged with GPS and SAR-derived ice flow measurements.

**L446: The statement "Hektoria Glacier lacks long-term elevation change data points" is not valid, at least for TanDEM-X data 2011 to 2016 (Rott et al., 2018), but also regarding further data from this mission and from other satellite sensors (e.g. CryoSat).**

We have altered the sentence, because in this context we were referring to elevation changes beginning with the collapse, not "long-term": in the sense of many years or decades. We see how this was confusing. We will explore longer-term changes with more satellite data when we investigate Hektoria's grounding line migration and unusual retreat in a follow-on paper.

**L523-L525: Quantitative estimates on the fast ice backstress magnitude and its spatial pattern are needed for affirming this conclusion, as well as considerations regarding possible changes of other driving factors, such as winds and ocean currents.**

Please see the response to General Comment #3 and the revised discussion.

**L531-L554: The argumentation, referring to proposed governing processes for glacier response and claiming similarity to the 2002 break-up event, are not well founded. There are various clues indicating major differences regarding the calving regimes and dynamic response of the Larsen B tributary glaciers in 2002 versus 2022/2023. Up to the 2002 event the glaciers were close to a balanced state, the tongues were several hundred**

metres thick and grounded. In 2022 the newly formed sections of the glacier tongues were composed of comparatively thin floating ice. Furthermore, there is evidence of structural weakness. On the floating sections high resolution elevation data show rugged surface structure and wave-like surface features of different wavelengths. There are also indications for major strain-rate weakening, as velocity data show. For example, on the central flowline of Hektoria Glacier a threefold increase of velocity between the grounding line and the glacier front is evident in 2017 data, and on Crane Glacier a twofold increase (Rott et al., 2020).

We have added details about the state of the glaciers and ice shelf/fast ice prior to the two events into this section (5.3).

L537-L538: On Crane Glacier close to the calving front a velocity of 9.6 m/day was observed in June 2007 data (Wuite et al., 2015), 2.3 times the 2003 value cited in L538.

Good suggestion, we have now elaborated on the velocity history of Crane Glacier.

L542: Based on the updated grounding line position, the number for retreat of grounded ice should be corrected.

We have added retreat estimates that reference the 2016 grounding line.

L543-L544: The floating ice of 15 km length in front of Hektoria Glacier was in fact a remnant section of the Larsen B ice shelf that remained in the pro-glacial bay for several month after disintegration of the main ice shelf. The boundary of the ice shelf is clearly evident in the tidal deformation pattern of ERS-tandem (1-day repeat) interferometric data. The retreat of grounded ice started in March 2003. The changes between February 2002 and April 2003 are documented by Rack et al. (2004) by means of several SAR images.

We see your point that what we called the Hektoria Glacier Tongue was still the ice shelf, we suppose here it is a point of semantics, as this part of the ice shelf was located in the Hektoria Fjord and therefore originated from HGE and not a combination of many outlet glaciers in the main Larsen B Embayment. For clarity and to distinguish it from the rest of the shelf, we retained the term, but we note our informal designation. We corrected the date at which Hektoria began calving at its grounded front and referenced Rack and Rott (2004).

L555ff, Conclusions: Major revisions are needed, taking into account the comments above.

We have changed the conclusion to incorporate some of the suggestions that were described above.

References cited in the manuscript, but not included in the list:

**Crawford et al., 2022 (cited in L159)**
**Hersbach et al., 2020 (cited in L104)**
**Kwon et al., 2020; (cited in L473)**
**Ochwat et al., 2022 (cited in L159)**
**Robel et al., 2017 (cited in L527)**
**Smith et al., 2020 (cited in L38 and L 501)**

We have added these, thank you.

**References**
Needell, C., and Holschuh, N.: Evaluating the retreat, arrest, and regrowth of Crane Glacier against marine ice cliff process models. Geophys. Res. Lett., 50, e2022GL102400. https://doi.org/10.1029/2022GL102400, 2023.

Rack, W., and Rott, H.: Pattern of retreat and disintegration of Larsen B ice shelf, Antarctic Peninsula, Ann. Glaciol., 39, 505-510, 2004.

Rott, H. Abdel Jaber, W., Wuite, J., Scheiblauer, S., Floricioiu, D., van Wessem, J.M., Nagler, T., Miranda, N., and van den Broeke, M.R.: Changing pattern of ice flow and mass balance for glaciers discharging into the Larsen A and B embayments, Antarctic Peninsula, 2011 to 2016, Cryosphere, 12, 1273–1291, https://doi.org/10.5194/tc-12-1273-2018, 2018.

Rott, H., Waite, J., De Rydt, J., Gudmundsson, G.H., Floricioiu, D., and Rack, W.; Impact of marine processes on flow dynamics of northern Antarctic Peninsula outlet glaciers, Nature Communications, 11:2969,| https://doi.org/10.1038/s41467-020-16658-y, 2020.

Wuite, J., Rott, H., Hetzenecker, M., Floricioiu, D., De Rydt, J., Gudmundsson, G. H., Nagler, T., and Kern, M.: Evolution of surface velocities and ice discharge of Larsen B outlet glaciers from 1995 to 2013, Cryosphere, 9, 957–969, https://doi.org/10.5194/tc-9-957-2015, 2015.

**References in our response:**
Cape, M. R., Vernet, M., Skvarca, P., Marinsek, S., Scambos, T., and Domack, E. (2015), Foehn winds link climate-driven warming to ice shelf evolution in Antarctica, *J. Geophys. Res. Atmos.*, 120, 11,037–11,057, doi:10.1002/2015JD023465.

Francis, D., Mattingly, K. S., Lhermitte, S., Temimi, M., and Heil, P.: Atmospheric extremes caused high oceanward sea surface slope triggering the biggest calving event in more than 50 years at the Amery Ice Shelf, The Cryosphere, 15, 2147–2165, https://doi.org/10.5194/tc-15-2147-2021, 2021

Laffin, M. K., Zender, C. S., van Wessem, M., and Marinsek, S.: The role of föhn winds in eastern Antarctic Peninsula rapid ice shelf collapse, The Cryosphere, 16, 1369–1381, https://doi.org/10.5194/tc-16-1369-2022, 2022.

Needell, C., & Holschuh, N. (2023). Evaluating the retreat, arrest, and regrowth of Crane Glacier against marine ice cliff process models. *Geophysical Research Letters*, 50, e2022GL102400. https://doi.org/10.1029/2022GL102400

Sun, Y., Riel, B., & Minchew, B. (2023). Disintegration and buttressing effect of the landfast sea ice in the Larsen B embayment, Antarctic Peninsula. *Geophysical Research Letters*, 50, e2023GL104066. https://doi.org/10.1029/2023GL104066

Surawy-Stepney, T., Hogg, A. E., Cornford, S. L., Wallis, B. J., Davison, B. J., Selley, H. L., Slater, R. A. W., Lie, E. K., Jakob, L., Ridout, A. L., Gourmelen, N., Freer, B. I. D., Wilson, S. F., and Shepherd, A.: The impact of landfast sea ice buttressing on ice dynamic speedup in the Larsen-B Embayment, Antarctica, The Cryosphere Discuss. [preprint], https://doi.org/10.5194/tc-2023-128, in review, 2023.

---

## Referee Report (RR1)

**Comments on "Triggers of the 2022 Larsen B multi-year landfast sea ice break-out and initial glacier response" (revised version)**

by N.E. Ochwat et al.

I wish to thank the authors for the careful considerations of the comments on the first version of the paper and the detailed response. The revisions address a main part of the issues raised in the review. However, there are still some items to be revisited, taking into account the issues explained below.

*Main issues:*

*Volume and mass of the floating terminus of HGE glaciers:*

The volume of the floating terminus of HGE glaciers is largely overestimated, contradicting mass continuity in view of the available mass for frontal advance. In L392/393 it is stated: "Hektoria Glacier had an extended thick (> 300 m) floating tongue that persisted until 12 to 17 March 2022". The joint HGE terminus area (formed by frontal advance after 2011) covered in January 2022 an area of 250 km$^2$ (L243). Assuming a thickness of 300 m and density of 900 kg m$^{-3}$ adds up to a total volume of 75 km$^3$ and mass of 67.5 Gt. This is about two times the mass flux (MFL) supplied through the HGE gates located close to the 2011 glacier fronts. See numbers for 2011 to 2016 in Rott et al., 2018, and 2016 to 2021 computed for the same gates accounting for reduced velocities:

- MFL 2011-2013: HG 5.73 Gt a$^{-1}$, Evans 0.39 Gt a$^{-1}$
- MFL 2013-2016: HG 3.39 Gt a$^{-1}$, Evans 0.30 Gt a$^{-1}$.
- MFL for July 2016 to Jan 2022 HG 2.24 Gt a$^{-1}$, Evans 0.23 Gt a$^{-1}$.
- Total HGE mass flux July 2011 to Jan 2022 to the frontal advance area: 36.9 Gt a$^{-1}$.

This implies a mean ice thickness of 148 m for an area of 250 km$^2$ if no frontal calving at all would have taken place between 2011 and Jan 2022. In fact, there were many small calving events, in particular during the first years of frontal advance. In view of these numbers, the statements referring to ice thickness on the order of 300 m and more need to be corrected.

*Estimate of the grounding zone location:*

The assumption of a partial grounding zone (GZ) within the advancing glacier terminus area and the inferred location of the grounding zones shown in Fig. 8 for HG and Crane glaciers are lacking traceability and are not in agreement with mass continuity (see point above). A much larger supply of ice mass to the frontal advance area would be needed than actually available. Fig. 8 (page 17) shows the inferred grounding zone (GZ) positions downstream (seaward) of the 2011 glacier fronts. By contrast, intensive thinning of the ice inland of the HG 2011 front, going on in subsequent years, implies further upstream shifts of the GZ after2011 (see the figure on HG elevation change below). Furthermore, the inferred GZ of Crane glacier in Fig. 8 shows a GZ seaward protrusion in the centre of the glacier which is located in a deep narrow canyon. In such a setting seaward extent of grounded ice has to be expected along the lateral slopes rather than in the centre.

*Further issues:*

L32: The calving of grounded ice was delayed by many months (not rapid).

L236/236: "During the 2011-2022 period of fast ice presence in the embayment, changes in the glacier extents suggests that the fast ice stabilized the Larsen B tributary glaciers" During 2011 to 2013 the losses of glacier mass were very high, and also in 2013 to 2016 the mass deficit was significant (Rott et al., 2018). This means there was no distinct stabilization signal during the first years of the fast ice period, but a rather gradual transition.

Figure 9c: Please check the Hektoria LS velocities for Nov. 2022, Dec 2022, Jan 2023. These numbers exceed the numbers derived from Sentinel-1 data and show a different trend along the flowline.

Figure 12, chronology: In the context of processes leading to frontal advance after 2011 detailed data on glacier mass balance, mass fluxes and ice flow velocities (as reported by Rott et al., 2018) are of relevance. (not mentioned in the chronology)

L592/593: "The calving regimes and dynamical changes of the Larsen B tributary glaciers are similar to their response after the 2002 Larsen B ice shelf." This is statement is not well founded, considering that the ice shelf in 2002 was more than 200 m thick, the tributary glaciers were in balance, at least up to 1999, and the glacier tongues were several hundred metres thick, of compact ice and grounded. In contrast, in 2022 the glacier had newly formed floating glacier tongues of less than 150 m mean thickness with rugged surface topography (evident in ICESat-2 transects of Fig. S1), implying significant variations in ice thickness at small spatial scale and rheologically weak ice on account of this.

L623: High thinning rates were observed on Hektoria Glacier also in 2011 to 2013, amounting to 20 m a$^{-1}$ on extended sections of grounded ice.

[Figure]

Figure 1: Map of rate of surface elevation change (dh/dt, m/year) from June 2011 to June 2013 based on the elevation difference in TanDEM-X DEMs. Background: TanDEM-X amplitude image of Hektoria and Green glacier terminus, 2011-06-25. Colour code dh/dt from ≤-22 m/yr to ≥+6 m/yr. According to mass continuity the transition from dh/dt on the order of -20m/year to much smaller numbers is a clear indication for the transition from grounded to floating ice. Further details in Rott et al., 2018 and 2020.

---

## Author Response (AR2)

Response to Dr. Helmut Rott
We thank the reviewer sincerely for a very careful and thorough review. We appreciate the meticulous comments and the perspective of the reviewer as well. We have modified the text to incorporate some of the changes suggested. Below, you will find our responses in blue.

Naomi Ochwat, on the behalf of the coauthors

**Comments on "Triggers of the 2022 Larsen B multi-year landfast sea ice break-out and initial glacier response" (revised version)**
**by N.E. Ochwat et al.**

**I wish to thank the authors for the careful considerations of the comments on the first version of the paper and the detailed response. The revisions address a main part of the issues raised in the review. However, there are still some items to be revisited, taking into account the issues explained below.**

**Main issues:**
**Volume and mass of the floating terminus of HGE glaciers:**
**The volume of the floating terminus of HGE glaciers is largely overestimated, contradicting mass continuity in view of the available mass for frontal advance. In L392/393 it is stated: "Hektoria Glacier had an extended thick (> 300 m) floating tongue that persisted until 12 to 17 March 2022". The joint HGE terminus area (formed by frontal advance after 2011) covered in January 2022 an area of 250 km2 (L243). Assuming a thickness of 300 m and density of 900 kg m-3 adds up to a total volume of 75 km3 and mass of 67.5 Gt. This is about two times the mass flux (MFL) supplied through the HGE gates located close to the 2011 glacier fronts. See numbers for 2011 to 2016 in Rott et al., 2018, and 2016 to 2021 computed for the same gates accounting for reduced velocities:**
- **MFL 2011-2013: HG 5.73 Gt a-1, Evans 0.39 Gt a-1**
- **MFL 2013-2016: HG 3.39 Gt a-1, Evans 0.30 Gt a-1.**
- **MFL for July 2016 to Jan 2022 HG 2.24 Gt a-1, Evans 0.23 Gt a-1.**
- **Total HGE mass flux July 2011 to Jan 2022 to the frontal advance area: 36.9 Gt a-1.**

**This implies a mean ice thickness of 148 m for an area of 250 km2 if no frontal calving at all would have taken place between 2011 and Jan 2022. In fact, there were many small calving events, in particular during the first years of frontal advance. In view of these numbers, the statements referring to ice thickness on the order of 300 m and more need to be corrected.**

We appreciate your detailed analysis and calculations of the mass flux; however, the altimetry and DEM data indicate that the thickest part of the HGE tongue is at least 300-400 m thick (Figure S1 and S2). The shear margins are slightly thinner (Figure S1D) so not all of the 250 km$^2$ of the ice tongue is that thick, but the majority of it is (Figure S1E). There is some variation

in the ice thickness, but not enough to constitute a deep discussion in this paper as the main focus of the paper is on the triggers of the loss of the fast ice and how the glaciers responded in its immediate aftermath.

**Estimate of the grounding zone location:**
**The assumption of a partial grounding zone (GZ) within the advancing glacier terminus area and the inferred location of the grounding zones shown in Fig. 8 for HG and Crane glaciers are lacking traceability and are not in agreement with mass continuity (see point above). A much larger supply of ice mass to the frontal advance area would be needed than actually available. Fig. 8 (page 17) shows the inferred grounding zone (GZ) positions downstream (seaward) of the 2011 glacier fronts. By contrast, intensive thinning of the ice inland of the HG 2011 front, going on in subsequent years, implies further upstream shifts of the GZ after2011 (see the figure on HG elevation change below). Furthermore, the inferred GZ of Crane glacier in Fig. 8 shows a GZ seaward protrusion in the centre of the glacier which is located in a deep narrow canyon. In such a setting seaward extent of grounded ice has to be expected along the lateral slopes rather than in the centre.**

Regarding the grounding line position, our determination of the grounding line was from a distinct change in calving style, as well as altimetry profile data, DEMs, and optical imagery glacier characteristics. We suggest that the discrepancy in our two grounding lines is due to the presence of an ice plain spanning the region between the two G.L. assessments. We infer that the ice plain may be ephemerally grounded with the tides (see Tuckett et al., 2020) or may have a tidally-paced oscillation in ice flow speed. In addition to your grounding line and our grounding line, we have now included Tuckett's grounding line on Figure 8. In the methods we explain the different possibilities of the grounding lines, the methodologies, uncertainties, and how the presence of an ice plain links it all together. We are also writing a follow-up paper that goes into greater detail on the grounding line.

**Further issues:**
**L32: The calving of grounded ice was delayed by many months (not rapid).**

According to Pfeffer 2007, "rapid" is defined as "retreat rates in excess of 200 m/yr". With this definition, the loss of grounded ice occurred within the year and was much greater than 200 m/yr. Other papers in the literature refer to "rapid" as 1000 m/yr for Columbia Glacier and Upsala Glacier East as well as 15 km over 8 years for Röhss Glacier (Pfeffer 2007; Warren et al., 1995; Glasser et al., 2011). This delay you mention may have existed in previously documented "rapid retreat" cases but due to the low temporal resolution of data at the time the delay may not have been obvious. To maintain a consistent terminology, we will continue to use "rapid".

**L236/236: "During the 2011-2022 period of fast ice presence in the embayment, changes in the glacier extents suggests that the fast ice stabilized the Larsen B tributary glaciers"**
**During 2011 to 2013 the losses of glacier mass were very high, and also in 2013 to 2016 the mass deficit was significant (Rott et al., 2018). This means there was no distinct**

**stabilization signal during the first years of the fast ice period, but a rather gradual transition.**

We have altered the sentence to say "During the 2011-2022 period of fast ice presence in the embayment, changes in the glacier extents and GNSS data suggests that the fast ice stabilized the Larsen B tributary glaciers and buttressed the Scar Inlet Ice Shelf, relative to the state prior to the fast ice occupation."

**Figure 9c: Please check the Hektoria LS velocities for Nov. 2022, Dec 2022, Jan 2023. These numbers exceed the numbers derived from Sentinel-1 data and show a different trend along the flowline.**

Thank you for bringing this to our attention. We noticed that one of the image pairs for November 2022 had only an 8-day separation, making the error range for the reported velocities very large. We have removed that image pair and replaced it with one spanning 22 November to 08 December, 2022. We have also adjusted the scatter plot to show the difference between grounded ice flow speeds (solid dots, as before) and melange flow (open circles) for January 2023. We have adjusted the manuscript text accordingly. This data is consistent with the ITS_LIVE data (see screenshot below), including the higher accelerations in the uppermost region of Hektoria in late 2022.

Our velocity profile is aligned with the altimetry profiles from the IceBridge flights earlier in the 2000's. This is to support later analyses that will look at the relationship between thinning and flow speed. Because the velocities extracted do not follow a central flow line, and because our extracted band to gather vectors for the profile is wide, there are some variations in the along-flow trend in speed relative to a true centerline of flow. Nevertheless, the graph provides a good representation of the general speed-up and loss of ice through the mapped period.

Please see the replacement figure:

[Figure]

And data from the ITS_LIVE interactive mapping tool:

[Figure]

Velocity data generated using auto-RIFT (Gardner et al., 2018) and provided by the NASA MEaSUREs ITS_LIVE project (Gardner et al., 20XX).

https://mappin.itsliveiceflow.science/chart?lat=-64.89619&lon=-61.64429&c=ll&lat=-64.86528&lon=-61.65939&c=cpg&lat=-64.92356&lon=-61.63193&c=reb&lat=-64.96776&lon=-61.62231&c=c

**Figure 12, chronology: In the context of processes leading to frontal advance after 2011 detailed data on glacier mass balance, mass fluxes and ice flow velocities (as reported by Rott et al., 2018) are of relevance. (not mentioned in the chronology)**

We have now included reference to your 2018 paper in the chronology figure.

**L592/593: "The calving regimes and dynamical changes of the Larsen B tributary glaciers are similar to their response after the 2002 Larsen B ice shelf." This is statement is not well founded, considering that the ice shelf in 2002 was more than 200 m thick, the tributary glaciers were in balance, at least up to 1999, and the glacier tongues were several hundred metres thick, of compact ice and grounded. In contrast, in 2022 the glacier had newly formed floating glacier tongues of less than 150 m mean thickness with rugged surface topography (evident in ICESat-2 transects of Fig. S1), implying significant variations in ice thickness at small spatial scale and rheologically weak ice on account of this.**

There are several aspects of the two events that are similar, as well as important differences to identify. The rest of the paragraph discusses this:

"The calving regimes and dynamical changes of the Larsen B tributary glaciers are similar to their response after the 2002 Larsen B ice shelf disintegration, suggesting that calving is an immediate response to stress perturbations (Hulbe et al. 2008). At first glance, the two events were quite different; for example, the tributary glaciers were stable prior to the 2002 event and though they were readvancing and stabilizing prior to the 2022 event they were still in an imbalanced state (Seehaus et al., 2023), additionally the ice shelf was old and thick whereas the fast ice was much younger and an order of magnitude thinner. However, despite these differences, the similarities in the tributary glacier response to the two events are important to identify."

**L623: High thinning rates were observed on Hektoria Glacier also in 2011 to 2013, amounting to 20 m a-1 on extended sections of grounded ice.**

We acknowledge that Hektoria was still in a state of thinning from 2011-2013, but this section of the paper is discussing the immediate response of the glacier to the loss of the ice shelf in 2002-2003 and fast ice in 2022-2023; hence, we left out a more detailed examination of thinning rates over the last 20 years.

[Figure]

**Figure 1: Map of rate of surface elevation change (dh/dt, m/year) from June 2011 to June 2013 based on the elevation difference in TanDEM-X DEMs. Background: TanDEM-X amplitude image of Hektoria and Green glacier terminus, 2011-06-25. Colour code dh/dt from -22 m/yr to +6 m/yr. According to mass continuity the transition from dh/dt on the order of -20m/year to much smaller numbers is a clear indication for the transition from grounded to floating ice. Further details in Rott et al., 2018 and 2020.**

Given this graphic, it is evident the area of the hypothesized ice plain thinned up to ~8 m/yr in some areas during this time. If that was floating ice in hydrostatic equilibrium that would be more than 60 m of thinning in one year, which is clearly not likely given the ocean conditions of this region (Nicholls, Pudsey, and Morris 2004). This kind of transition is possible when going from a steep glacier to a flat ice plain, furthering our interpretation of the presence of a partially grounded ice plain. As mentioned previously, we will be evaluating the grounding lines and ice plain feature in greater detail in our follow-up paper.